# Unlocking Tuning-free Generalization: Minimizing the PAC-Bayes Bound with Trainable Priors

## Abstract

It is widely recognized that the generalization ability of neural networks can be greatly enhanced through carefully tuning the training procedure. The current state-of-the-art training approach involves utilizing stochastic gradient descent (SGD) or Adam optimization algorithms along with a combination of additional regularization techniques such as weight decay, dropout, or noise injection. Optimal generalization can only be achieved by tuning a multitude of hyper-parameters extensively, which can be time-consuming and necessitates the additional validation dataset. To address this issue, we present a nearly tuning-free PAC-Bayes training framework that requires no extra regularization. This framework achieves test performance comparable to that of SGD/Adam, even when the latter are optimized through a complete grid search and supplemented with additional regularization terms.

## 1 Introduction

The prevalent training methodologies for neural networks, utilizing SGD or Adam, often necessitate integrating various tricks/regularization techniques to optimize generalization performance. To understand the underlying benefits of these strategies, numerous studies have focused on studying individual strategies. For instance, it has been shown that larger learning rates (Cohen et al., 2021; Barrett & Dherin, 2020), momentum (Ghosh et al., 2022), smaller batch sizes (Lee & Jang, 2022) and batch normalization (Luo et al., 2018) individually induce higher degrees of *implicit regularization* on the sharpness of the loss function, yielding better generalization. Additionally, the intensity of *explicit regularization* techniques such as weight decay (Loshchilov & Hutter, 2017), dropout (Wei et al., 2020), parameter noise injection (Neelakantan et al., 2015; Orvieto et al., 2022), label noise (Damian et al., 2021) can significantly affect generalization. Despite these observations and explanations, it's unclear why seeking optimal combinations of these regularizations is still crucial in practice. Adjusting the intensity of each regularization based on different scenarios can be a tedious job, especially when previous research has indicated that some techniques can conflict with each other (Li et al., 2019). We summarize this challenge for conventional training in (Q1).

Alternatively, PAC-Bayes generalization bounds provide foundational insights into generalization in the absence of validation and testing data (Shawe-Taylor & Williamson, 1997). Jiang et al. (2019) further suggests that PAC-Bayes bounds are among the best for evaluating generalization capabilities. Although PAC-Bayes bounds were traditionally used only in the post-training stage for quality control (Vapnik, 1998; McAllester, 1999), the recent work (Dziugaite & Roy, 2017) has opened the door to using these bounds during the training phase. They showed that one can directly train a network via optimizing the PAC-Bayes bound, a strategy we refer to as PAC-Bayes training, and obtain reasonable performances. Ideally, the generalization performance of deep neural networks should be enhanced by directly minimizing its quantitative measurements, specifically the PAC-Bayes bounds, without any other regularization tricks. However, it is well-known that PAC-Bayes bounds could become vacuous in highly over-parameterized regimes, making the practical use of PAC-Bayes training challenging on highly deep neural networks. Furthermore, choosing the right prior presents another challenge, as the quality of the prior critically affects the tightness of the bound. We summarize the challenges of minimizing PAC-Bayes bound to improve the generalization of deep learning models in (Q2).

**Q1**: *Is it possible to reduce the reliance of the training process on hyper-parameter selection and minimize the use of regularizations/tricks without compromising generalization?*

**Q2**: *Can we design a training framework based on PAC-Bayes bounds minimization with trainable priors for highly over-parameterized deep neural networks as a solution of Q1?*

This paper provides affirmative answers to both questions by proposing a practical PAC-Bayes training framework with learnable priors. Using the framework, we show:

1. PAC-Bayes training can achieve state-of-the-art results for deep neural networks with various architectures.

2. PAC-Bayes training is nearly tuning-free. This eliminates the complexities of hyper-parameter searches and the dependency on validation data, effectively augmenting the training dataset.

3. From PAC-Bayes training, we see that among the different regularization/tricks, only weight decay and noise injections are essential.

## 2 PRELIMINARIES

Throughout the paper, boldface letters denote vectors. We first introduce the basic setting of the PAC-Bayes analysis. For any supervised-learning problem, the goal is to find a suitable model $\mathbf{h}$ from some hypothesis space, $\mathbf{h} \in \mathcal{H} \subseteq \mathbb{R}^d$, with the help of the training data $\mathcal{S} \equiv \{z_i\}_{i=1}^m$, where $z_i$ is the training pair with sample $\mathbf{x}_i$ and its label $y_i$. The usual assumption is that the training and testing data are i.i.d. sampled from the same unknown distribution $\mathcal{D}$. For a given model $\mathbf{h} \in \mathcal{H}$, the empirical and population/generalization errors are defined as:

$$\ell(\mathbf{h}; \mathcal{S}) = \frac{1}{m} \sum_{i=1}^m \ell(\mathbf{h}; z_i), \quad \ell(\mathbf{h}; \mathcal{D}) = \mathbb{E}_{\mathcal{S} \sim \mathcal{D}}(\ell(\mathbf{h}; \mathcal{S})),$$

where the loss function $\ell(\mathbf{h}; z_i) : \mathbf{h} \mapsto \mathbb{R}^+$ measures the misfit between the true label $y_i$ and the predicted label by the model $\mathbf{h}$. PAC-Bayes bounds include a family of upper bounds on the generalization error of the following type.

**Theorem 2.1.** *Maurer (2004)Assume the loss function $\ell$ is **bounded** within the interval $[0, C]$. Given a **preset** prior distribution $\mathcal{P}$ over the model space $\mathcal{H}$, and given a scalar $\delta \in (0, 1)$, for any choice of i.i.d $m$-sized training dataset $\mathcal{S}$ according to $\mathcal{D}$, and all posterior distributions $\mathcal{Q}$ over $\mathcal{H}$, we have*

$$\mathbb{E}_{\mathbf{h} \sim \mathcal{Q}} \ell(\mathbf{h}; \mathcal{D}) \leq \mathbb{E}_{\mathbf{h} \sim \mathcal{Q}} \ell(\mathbf{h}; \mathcal{S}) + C \sqrt{\frac{\ln(\frac{\sqrt{2m}}{\delta}) + \mathrm{KL}(\mathcal{Q}||\mathcal{P})}{2m}},$$

*holds with probability at least $1 - \delta$. Here, KL stands for the Kullback-Leibler divergence.*

A PAC-Bayes bound measures the gap between the expected empirical error and the expected generalization error regarding the KL divergence between the prior $\mathcal{P}$ and the posterior $\mathcal{Q}$. It's worth noting that this bound holds for any data-independent prior $\mathcal{P}$ and any posterior $\mathcal{Q}$, which enables one to further optimize the bound by searching for the best posterior. In practice, the posterior expectation can be the well-trained weights based on data, and the prior expectation can be the initial model or $\mathbf{0}$. However, there are two major obstacles to using existing PAC-Bayes bound in the literature directly to conduct training for classification tasks with deep neural networks:

**Bounded loss.** As the cross-entropy loss used in classification tasks is unbounded, directly applying the bound for bounded loss in Theorem 2.1 would fail. To use Theorem 2.1 appropriately, one can begin by converting the cross-entropy loss to a bounded version and then apply Theorem 2.1. There are many ways to convert it to bounded loss (clipping, log-transforms), but they all tend to decrease the variance of the loss across the inputs, making the training slow. From our experience with deep neural networks, this will even cause the training accuracy to plateau at a very low level.

**Computational elusiveness.** While some PAC-Bayes bounds have been proposed for unbounded loss, their numerical values prove challenging to estimate in practice. In Haddouche et al. (2021), an upper bound was derived for variables that satisfy the so-called hypothesis-dependent range condition, which is stated as $\sup_z \ell(\mathbf{h}; z) \leq K(\mathbf{h})$, $\forall \mathbf{h} \in \mathcal{H}$. However, the cross entropy loss does not satisfy this condition without putting extra assumptions on the input. Kuzborskij & Szepesvári

(2019) proposed PAC-Bayes bound for unbounded variables using Efron-Stein type of inequalities and obtained the following PAC-Bayes bound (adapted to our notations):

$$\mathbb{E}_{\mathbf{h}\sim\mathcal{Q}}\ell(\mathbf{h};\mathcal{D}) \leq \mathbb{E}_{\mathbf{h}\sim\mathcal{Q}}\ell(\mathbf{h};\mathcal{S}) + \sqrt{\frac{1}{m}\mathbb{E}_{\mathbf{h}\sim\mathcal{Q}}\left[\ell_1(\mathbf{h};\mathcal{S}) + \mathbb{E}_{z'\sim\mathcal{D}}\ell(\mathbf{h};z')^2\right]\mathrm{KL}(\mathcal{Q}||\mathcal{P})} + \frac{1}{m},$$

where $z' \sim \mathcal{D}$ is a test sample drawn from the data distribution and $\ell_1(\mathbf{h};\mathcal{S}) := \frac{1}{m}\sum_{i=1}^m \ell(\mathbf{h};z_i)^2$, This bound holds for any unbounded loss with a finite second-order moment. However, the term $\mathbb{E}_{\mathbf{h}\sim\mathcal{Q}}\mathbb{E}_{z'\sim\mathcal{D}}\ell(\mathbf{h};z')^2$ is almost as difficult to estimate as the generalization error itself. Germain et al. (2016) has a PAC-Bayes bound, including a term of the loss variance. However, the variance will be large initially, and the training accuracy will not increase properly if training with this bound.

## 3 RELATED WORK

The PAC-Bayes bound was first used for training of neural networks in Dziugaite & Roy (2017). Specifically, the bound McAllester (1998) is used for training a shallow stochastic neural network on binary MNIST classification with bounded 0-1 loss and has proven to be non-vacuous. Following this work, many recent studies (Letarte et al., 2019; Rivasplata et al., 2019; Pérez-Ortiz et al., 2021; Biggs & Guedj, 2021; Perez-Ortiz et al., 2021; Zhou et al., 2018) expanded the applicability of PAC-Bayes bounds to a wider range of neural network architectures and datasets. However, most studies are limited to training shallow networks with binary labels using bounded loss, which restricts their broader application to deep network training. To better align with practical applications, several PAC-Bayes bounds for unbounded loss have been established (Audibert & Catoni, 2011; Alquier & Guedj, 2018; Holland, 2019; Kuzborskij & Szepesvári, 2019; Haddouche et al., 2021; Rivasplata et al., 2020; Rodríguez-Gálvez et al., 2023). However, from the PAC-Bayes training perspective, it is still unclear if these theoretically tight bounds can lead to better performance in training.

Along another vein, Dziugaite et al. (2021) suggested that a tighter PAC-Bayes bound could be achieved with a data-dependent prior. They divide the data into two sets, using one to train the prior distribution and the other to train the posterior with the optimized prior, thus making the prior independent from the training dataset for the posterior. This, however, reduces the training data available for the posterior. Dziugaite & Roy (2018) and Rivasplata et al. (2020) justified the approach of learning the prior and posterior with the same set of data by utilizing differential privacy. However, their argument only holds for priors provably satisfying the so-called $DP(\epsilon)$-condition in differential privacy, which limits their practical application. As of now, most existing PAC-Bayes training algorithms require hyper-parameter tuning, sometimes even more than vanilla SGD training, making it less feasible in practice. In this work, we make a step forward in the PAC-Bayes training regime, making it more practical and demonstrating its potential to replace the normal training of neural networks in realistic settings.

## 4 PAC-BAYES BOUNDS WITH TRAINABLE PRIORS

We first provide a new extension of the PAC-Bayes bound from bounded to unbounded loss. First, we need to define some sort of "soft" upper bound of the loss to be used in place of the hard upper bound in the traditional PAC-Bayes bound for bounded loss. For example, in Kuzborskij & Szepesvári (2019), the soft upper bound of the loss is set to the second-order moment. However, the second-order moment could be much larger than the variance, leading to a vacuous bound in practice. This matches our numerical observation on deep neural network applications, where the bound is too large to be binding. Therefore, we propose the following definition of the "soft" bound using the exponential moment inequality that will yield a more practical PAC-Bayes bound later.

**Definition 1** (Exponential moment on finite intervals). *Let $X$ be a random variable and $0 \leq \gamma_1 \leq \gamma_2$ be two real numbers. We call any $K > 0$ an exponential moment bound of $X$ over the interval $[\gamma_1, \gamma_2]$, when*

$$\mathbb{E}[\exp{(\gamma X)}] \leq \exp{(\gamma^2 K)} \tag{1}$$

*holds for all $\gamma \in [\gamma_1, \gamma_2]$.*

Condition 1 arises naturally in various proofs [1] of traditional PAC-Bayes bounds (Theorem 2.1) as a key property that determines whether or not the PAC-Bayes bound holds. Due to its similarity to the definition of sub-Gaussian distribution, the traditional PAC-Bayes bound can be easily

---

[1] Such as the proof of Theorem 4.1 in Appendix B.1

extended from bounded loss to unbounded sub-Gaussian losses. However, PAC-Bayes bounds for sub-Gaussian losses are still often vacuous in deep neural network applications. We observed that the issue lies in that the sub-Gaussian bound unnecessarily requires the exponential moment inequality to hold on $[0, \infty)$. In most applications, we only need it to hold on a finite interval, which will yield a much smaller $K$ and in turn a tighter PAC-Bayes bound. This is the motivation of Definition 1.

To use Definition 1 in PAC-Bayes analysis, we first need to extend it to random variables parametrized over a hypothesis space.

**Definition 2** (Exponential moment over hypotheses). *Let $X(\mathbf{h})$ be a random variable parameterized by the hypothesis $\mathbf{h}$ in some space $\mathcal{H}$ (i.e., $\mathbf{h} \in \mathcal{H}$), and fix an interval $[\gamma_1, \gamma_2]$ with $0 \le \gamma_1 < \gamma_2 < \infty$. Let $\{\mathcal{P}_{\boldsymbol{\lambda}}, \boldsymbol{\lambda} \in \Lambda\}$ be a family of distribution over $\mathcal{H}$ parameterized by $\boldsymbol{\lambda} \in \Lambda \subseteq \mathbb{R}^k$. Then, we call any non-negative function $K(\boldsymbol{\lambda})$ a uniform exponential moment bound for $X(\mathbf{h})$ over the priors $\{\mathcal{P}_{\boldsymbol{\lambda}}, \boldsymbol{\lambda} \in \Lambda\}$ and the interval $[\gamma_1, \gamma_2]$ if the following holds*

$$\mathbb{E}_{\mathbf{h} \sim \mathcal{P}_{\boldsymbol{\lambda}}} \mathbb{E}[\exp(\gamma X(\mathbf{h}))] \le \exp(\gamma^2 K(\boldsymbol{\lambda})),$$

*for any $\gamma \in [\gamma_1, \gamma_2]$, and any $\boldsymbol{\lambda} \in \Lambda \subseteq \mathbb{R}^k$. The minimal such $K(\boldsymbol{\lambda})$ is defined to be*

$$K_{\min}(\boldsymbol{\lambda}) = \sup_{\gamma \in [\gamma_1, \gamma_2]} \frac{1}{\gamma^2} \log(\mathbb{E}_{\mathbf{h} \sim \mathcal{P}_{\boldsymbol{\lambda}}} \mathbb{E}[\exp(\gamma X(\mathbf{h}))]). \quad (2)$$

Now, we can establish the PAC-Bayes bound for losses that satisfy Definition 2. Specifically, we will plug into the $X(\mathbf{h})$ in Definition 2 with the loss $\ell(\mathbf{h}, z_i)$, which is a random variable parameterized by $\mathbf{h}$ with randomness coming from the input data $z_i$.

**Theorem 4.1** (PAC-Bayes bound for unbounded losses with **preset** priors). *Given a prior distribution $\mathcal{P}_{\boldsymbol{\lambda}}$ over the hypothesis space $\mathcal{H}$, that is parametrized by a fixed $\boldsymbol{\lambda} \in \Lambda$. Fix some $\delta \in (0, 1)$. For any choice of i.i.d $m$-sized training dataset $\mathcal{S}$ according to $\mathcal{D}$, and all posterior distributions $\mathcal{Q}$ over $\mathcal{H}$, we have*

$$\mathbb{E}_{\mathbf{h} \sim \mathcal{Q}} \ell(\mathbf{h}; \mathcal{D}) \le \mathbb{E}_{\mathbf{h} \sim \mathcal{Q}} \ell(\mathbf{h}; \mathcal{S}) + \frac{1}{\gamma m}(\ln \frac{1}{\delta} + \mathrm{KL}(\mathcal{Q}||\mathcal{P}_{\boldsymbol{\lambda}})) + \gamma K(\boldsymbol{\lambda}) \quad (3)$$

*holds for all $\gamma \in [\gamma_1, \gamma_2]$ with probability at least $1 - \delta$, provided that $\ell(\mathbf{h}, z)$ satisfies Definition 2 with bound $K(\boldsymbol{\lambda})$.*

The proof is available in Appendix B.1. Based on the PAC-Bayes bound in Theorem 4.1, we propose the following PAC-Bayes training procedure. We first parameterize the posterior distribution as $\mathcal{Q}_{\boldsymbol{\sigma}}(\mathbf{h})$, where $h$ is the mean of the posterior and $\boldsymbol{\sigma} \in \mathbb{R}^d$ contains the variance. Then we parametrize the prior as $\mathcal{P}_{\boldsymbol{\lambda}}$ with $\boldsymbol{\lambda} \in \mathbb{R}^k$, $k \ll m, d$. For the PAC-Bayes training, we propose to optimize over all four variables: $\mathbf{h}, \gamma, \boldsymbol{\sigma}$, and $\boldsymbol{\lambda}$.

$$(\hat{\mathbf{h}}, \hat{\gamma}, \hat{\boldsymbol{\sigma}}, \hat{\boldsymbol{\lambda}}) = \arg \min_{\substack{\mathbf{h}, \boldsymbol{\lambda}, \boldsymbol{\sigma}, \\ \gamma \in [\gamma_1, \gamma_2]}} \underbrace{\mathbb{E}_{\tilde{\mathbf{h}} \sim \mathcal{Q}_{\boldsymbol{\sigma}}(\mathbf{h})} \ell(\tilde{\mathbf{h}}; \mathcal{S}) + \frac{1}{\gamma m}(\ln \frac{1}{\delta} + \mathrm{KL}(\mathcal{Q}_{\boldsymbol{\sigma}}(\mathbf{h})||\mathcal{P}_{\boldsymbol{\lambda}})) + \gamma K(\boldsymbol{\lambda})}_{\equiv L_{PAC}(\mathbf{h}, \gamma, \boldsymbol{\sigma}, \boldsymbol{\lambda})}. \quad \text{(P)}$$

Optimizing the parameters in the prior will clearly help reduce the bound. However, whether it is a valid operation is questionable at first glance, as it seemingly contradicts the traditional belief that the prior has to be data-independent. Some previous work, such as Dziugaite & Roy (2017), already attempted a similar idea with bounded losses. Here, we apply it to unbounded loss with refined PAC-bayes bound, and more importantly, we provide a formal justification to validate this approach[2]. To establish the theoretical guarantee for the optimization over the prior, we need the following two assumptions.

**Assumption 4.1.1** (Continuity of the KL divergence). *Let $\mathfrak{Q}$ be a family of posterior distributions, let $\mathfrak{P} = \{P_{\boldsymbol{\lambda}}, \boldsymbol{\lambda} \in \Lambda \subseteq \mathbb{R}^k\}$ be a family of prior distributions parameterized by $\boldsymbol{\lambda}$. We say the KL divergence $\mathrm{KL}(\mathcal{Q}||\mathcal{P}_{\boldsymbol{\lambda}})$ is continuous with respect to $\boldsymbol{\lambda}$ over the posterior family, if there exists some non-decreasing function $\eta_1(x) : \mathbb{R}_+ \mapsto \mathbb{R}_+$ with $\eta_1(0) = 0$, such that $|\mathrm{KL}(\mathcal{Q}||\mathcal{P}_{\boldsymbol{\lambda}}) - \mathrm{KL}(\mathcal{Q}||\mathcal{P}_{\tilde{\boldsymbol{\lambda}}})| \le \eta_1(\|\boldsymbol{\lambda} - \tilde{\boldsymbol{\lambda}}\|)$, for all pairs $\boldsymbol{\lambda}, \tilde{\boldsymbol{\lambda}} \in \Lambda$ and for all $\mathcal{Q} \in \mathfrak{Q}$.*

---

[2]Partial theoretical justification for the use of data-dependent prior was provided by using the differential privacy (Rivasplata et al., 2020; Dziugaite & Roy, 2018). However, the restrictive requirement for the differential private priors seems to make it difficult to be applied to general settings.

**Assumption 4.1.2** (Continuity of the exponential moment bound). *Let $K_{\min}(\boldsymbol{\lambda})$ be as defined in Definition 2. Assume it is continuous with respect to the parameter $\boldsymbol{\lambda}$ of the prior in the sense that there exists a non-decreasing function $\eta_2(x) : \mathbb{R}_+ \mapsto \mathbb{R}_+$ with $\eta_2(0) = 0$ such that $|K(\boldsymbol{\lambda}) - K(\tilde{\boldsymbol{\lambda}})| \leq \eta_2(\|\boldsymbol{\lambda} - \tilde{\boldsymbol{\lambda}}\|)$, for all $\boldsymbol{\lambda}, \tilde{\boldsymbol{\lambda}} \in \Lambda$.*

In actual training, we will use the Gaussian family for the posterior and prior distributions. With Gaussian families, it is sufficient to set $\eta_1(x) = C_1 x$ and $\eta_2(x) = C_2 x$ for the two assumptions to hold, where $C_1, C_2$ are universal constants. Next, we present the main theorem that guarantees the performance of the minimizer of P.

**Theorem 4.2** (PAC-Bayes bound for unbounded losses and **trainable** priors). *Let $\mathfrak{Q}$ be a family of posterior distribution, let $\mathfrak{P} = \{P_{\boldsymbol{\lambda}}, \boldsymbol{\lambda} \in \Lambda \subseteq \mathbb{R}^k\}$ be a family of prior distributions parameterized by $\boldsymbol{\lambda}$. Fix two scalars $\epsilon, \varepsilon > 0$ to arbitrary (small) values. Let $n(\varepsilon) := \mathcal{N}(\Lambda, \|\cdot\|, \varepsilon)$ be the covering number of the set $\Lambda$ of the prior parameters. Under Assumption 4.1.1 and Assumption 4.1.2, the following inequality holds for the minimizer $(\hat{\mathbf{h}}, \hat{\gamma}, \hat{\boldsymbol{\sigma}}, \hat{\boldsymbol{\lambda}})$ of equation P with probability as least $1 - \epsilon$:*

$$
\mathbb{E}_{\mathbf{h} \sim \mathcal{Q}_{\hat{\boldsymbol{\sigma}}}(\hat{\mathbf{h}})} \ell(\mathbf{h}; \mathcal{D}) \leq \mathbb{E}_{\mathbf{h} \sim \mathcal{Q}_{\hat{\boldsymbol{\sigma}}}(\hat{\mathbf{h}})} \ell(\mathbf{h}; \mathcal{S}) + \frac{1}{\hat{\gamma}m} \left[ \log \frac{n(\varepsilon)}{\epsilon} + \mathrm{KL}(\mathcal{Q}_{\hat{\boldsymbol{\sigma}}}(\hat{\mathbf{h}}) \| \mathcal{P}_{\hat{\boldsymbol{\lambda}}}) \right] + \hat{\gamma} K(\hat{\boldsymbol{\lambda}}) + \eta
$$

$$
= L_{PAC}(\hat{\mathbf{h}}, \hat{\gamma}, \hat{\boldsymbol{\sigma}}, \hat{\boldsymbol{\lambda}}) + \eta + \frac{\log(n(\varepsilon))}{\hat{\gamma}m}, \tag{4}
$$

*where $\eta = (\frac{1}{\gamma_1 m} + \gamma_2)(\eta_1(\varepsilon) + \eta_2(\varepsilon))$.*

The proof is available in Appendix B.2. The theorem provides a generalization bound on the model learned as the minimizer of equation P with data-dependent priors. This bound contains two parts, the PAC-Bayes loss $L_{PAC}$ and two correction terms. Notably, these correction terms were absent in the traditional PAC-Bayes bound with fixed priors. Given that $(\hat{\mathbf{h}}, \hat{\gamma}, \hat{\boldsymbol{\sigma}}, \hat{\boldsymbol{\lambda}})$ minimizes $L_{PAC}$, evaluating $L_{PAC}$ at its own minimizer guarantees the first term as small as it can be. If the correction terms are negligible, then this PAC-Bayes bound remains low. Suppose we choose a small $\varepsilon$, then the first correction term $\eta$ would be small provided $\eta_1$ and $\eta_2$ increase with a sensible rate. Regarding the second correction term, the standard information-theoretical argument implies $\log(n(\varepsilon))$ grows linearly with the dimension $k$ of $\Lambda$. Therefore, if $k$ remains small relative to the dataset size $m$, the second correction component will also stay minimal. In the next section, we will explicitly calculate the last two correction terms when the prior and posterior are both Gaussian.

## 5 PAC-BAYES TRAINING ALGORITHM

### 5.1 GAUSSIAN PRIOR AND POSTERIOR

For the $L_{PAC}$ objective to have a closed-form formula, in the paper, we pick the simplest Gaussian distribution family as both the prior and the posterior distributions. More specifically, we set the posterior distribution to be centered around the training model $\mathbf{h}$ with trainable variance, i.e., $\mathcal{Q}_{\boldsymbol{\sigma}}(\mathbf{h}) = \mathcal{N}(\mathbf{h}, \mathrm{diag}(\boldsymbol{\sigma}))$, where $\boldsymbol{\sigma}$ contains the anisotropic variance of the weights and $\mathbf{h}$ contains the mean. By using a diagonal covariance matrix, we assume the weights to be independent of each other. We consider two types of priors, both centered around the initialization of the neural network $\mathbf{h}_0$ (as suggested by Dziugaite & Roy (2017)), but with different assumptions on the variance.

- **Scalar prior**, we use a universal scalar to encode the variance of all the weights in the prior, i.e., $\mathcal{P}_\lambda = \mathcal{N}(\mathbf{h}_0, \lambda I_d)$, where $\lambda$ is a scalar. With this prior, the KL divergence in P is:

$$
\mathrm{KL}(\mathcal{Q}_{\boldsymbol{\sigma}}(\mathbf{h}) \| \mathcal{P}_\lambda(\mathbf{h}_0)) = \frac{1}{2} \left[ -\mathbf{1}_d^\top \ln(\boldsymbol{\sigma}) + d(\ln(\lambda) - 1) + \frac{1}{\lambda}(\|\boldsymbol{\sigma}\|_1 + \|\mathbf{h} - \mathbf{h}_0\|^2) \right]. \tag{5}
$$

- **Layerwise prior**, weights in the $i$th layer share a common scalar variance $\boldsymbol{\lambda}_i$, but different layers have different variances. By setting $\boldsymbol{\lambda} = (\boldsymbol{\lambda}_1, ...., \boldsymbol{\lambda}_k)$ as the vector containing all the layerwise variances of a $k$ layers neural network, the prior is $\mathcal{P}_{\boldsymbol{\lambda}} = \mathcal{N}(\mathbf{h}_0, \mathrm{BlockDiag}(\boldsymbol{\lambda}))$, where $\mathrm{BlockDiag}(\boldsymbol{\lambda}))$ is obtained by diagonally stacking all $\boldsymbol{\lambda}_i I_{d_i}$ into a $d \times d$ matrix, where $d_i$ is the size of the $i$th layer. The KL divergence for layerwise prior is in Appendix A.1.

For shallow networks, it is enough to use the scalar prior; for deep neural networks, we found the layerwise prior is better.

By plugging in the closed-form for $\mathrm{KL}(\mathcal{Q}_{\boldsymbol{\sigma}}(\mathbf{h})||\mathcal{P}_{\boldsymbol{\lambda}})$ with the Gaussian family into the PAC-Bayes bound in Theorem 4.2, we have the following corollary that justifies the usage of PAC-Bayes bound on large neural networks. The proof is available in Appendix B.3.

**Corollary 5.0.1.** *Suppose the posterior and prior are Gaussian as defined above. Assume all parameters for the prior and posterior are bounded, i.e., we restrict the model $\mathbf{h}$, the posterior variance $\boldsymbol{\sigma}$ and the prior variance $\boldsymbol{\lambda}$, all to be searched over bounded sets, $\mathcal{H} := \{\mathbf{h} \in \mathbb{R}^d : \|\mathbf{h}\|_2 \leq \sqrt{d}M\}$, $\Sigma := \{\boldsymbol{\sigma} \in \mathbb{R}^d_+ : \|\boldsymbol{\sigma}\|_1 \leq dT\}$, $\Lambda =: \{\boldsymbol{\lambda} \in [e^{-a}, e^b]^k\}$, respectively, with fixed $M, T, a, b > 0$. Then,*

- *Assumption 4.1.1 holds with $\eta_1(x) = L_1 x$, where $L_1 = \frac{1}{2}\max\{d, e^a(2\sqrt{d}M + dT)\}$*

- *Assumption 4.1.2 holds with $\eta_2(x) = L_2 x$, where $L_2 = \frac{1}{\gamma_1^2}\left(2dM^2 e^{2a} + \frac{d(a+b)}{2}\right)$*

- *With high probability, the PAC-Bayes bound for the minimizer of equation P has the form*

$$\mathbb{E}_{\mathbf{h} \sim \mathcal{Q}_{\hat{\boldsymbol{\sigma}}}(\hat{\mathbf{h}})}\ell(\mathbf{h}; \mathcal{D}) \leq L_{PAC}(\hat{\mathbf{h}}, \hat{\gamma}, \hat{\boldsymbol{\sigma}}, \hat{\boldsymbol{\lambda}}) + \eta,$$

*where $\eta = \frac{k}{\gamma_1 m}\left(1 + \log\frac{CL(b+a)\gamma_1 m}{2k}\right)$, $C = \frac{1}{\gamma_1 m} + \gamma_2$ and $L = L_1 + L_2$.*

**Remark 5.0.1.** *In the bound, the term $L_{PAC}(\hat{\mathbf{h}}, \hat{\gamma}, \hat{\boldsymbol{\sigma}}, \hat{\boldsymbol{\lambda}})$ is inherently minimized as it evaluates the function $L_{PAC}$ at its own minimizer. If the correction term $\eta$ can be deemed insignificant, then the overall bound remains low. As explained in Remark 5.0.2 and 5.0.3, the logarithm term in the definition of $\eta$ grows very mildly with the dimension, so we can treat it (almost) as a constant. Thus, $\eta \sim \frac{k}{\gamma_1 m}$, from which we see $\eta$ (and therefore the bound) would be small if prior's degree of freedom $k$ is substantially less than the dataset size $m^3$.*

**Remark 5.0.2.** *In defining the boundedness of the domain $\mathcal{H}$ of $\mathbf{h}$ in Corollary 5.0.1, we used $\sqrt{d}M$ as the bound. Here, the factor $\sqrt{d}$ (where $d$ denotes the dimension of $\mathbf{h}$) is used to encapsulate the idea that if on average, the components of the weight are bounded by $M$, then the $\ell_2$ norm would naturally be bounded by $\sqrt{d}M$. The same idea applies to the definition of $\Sigma$.*

**Remark 5.0.3.** *Due to the above remark, $M, T, a, b$ can be treated as dimension-independent constants that do not grow with the network size $d$. As a result, the constants $L_1, L_2, L$ in Corollary 5.0.1, are dominated by $d$, and $L_1, L_2, L = O(d)$. This then implies the logarithm term in $\eta$ scales as $O(\log d)$, which grows very mildly with the size. Therefore, Corollary 5.0.1 can be used as the generalization guarantee for large neural networks.*

## 5.2 PAC-BAYES TRAINING ALGORITHM

**Estimating $K_{\min}(\boldsymbol{\lambda})$:** In practice, the function $K_{\min}(\boldsymbol{\lambda})$ must be estimated. Since we showed in Corollary 5.0.1 that $K_{\min}(\boldsymbol{\lambda})$ is Lipschtiz continuous, we can approximate it using piecewise-linear functions. Notably, since for each fixed $\boldsymbol{\lambda} \in \Lambda$, the prior is independent of the data, this procedure of estimating $K_{\min}(\boldsymbol{\lambda})$ can be carried out before training. More details are in Appendix A.2.

**Tuning-free PAC-Bayes training:** Algorithm 1 describe the PAC-Bayes training procedure with scalar prior. The one with layerwise prior can be found in Appendix A.3. Although there are several input parameters to be specified, the generalization performance is insensitive to the choice of parameters (Please see numerical results in Sec.6 for the stability and Appendix A.5 for more discussions), and we used the same choice of values across all the different settings. When everything else in the PAC-Bayes loss is fixed, $\gamma \in [\gamma_1, \gamma_2]$ has a closed-form solution,

$$\gamma^* = \min\left\{\max\left\{\gamma_1, \frac{1}{K_{\min}}\sqrt{\frac{\ln\frac{1}{\delta} + \mathrm{KL}(\mathcal{Q}_{\boldsymbol{\sigma}}(\mathbf{h})||\mathcal{P}_{\boldsymbol{\lambda}}(\mathbf{h}_0))}{m}}\right\}, \gamma_2\right\}. \quad (6)$$

---

[3]In all our experiments for various neural networks, we set $\gamma_1 = 0.5$ and $\gamma_2 = 10$. More discussions are in Appendix C.1.

---

**Algorithm 1** Tuning-free PAC-Bayes training (scalar prior)

---

**Input:** initial model $\mathbf{h}_0 \in \mathbb{R}^d$, $T_1 = 500$, $\lambda_1 = e^{-7}$, $\lambda_2 = 1$, $\gamma_1 = 0.5$, $\gamma_2 = 10$
**Output:** trained model $\hat{\mathbf{h}}$, posterior noise level $\hat{\boldsymbol{\sigma}}$
   $\mathbf{h} \leftarrow \mathbf{h}_0$, $\mathbf{v} \leftarrow \mathbf{1_d} \cdot \log(\frac{1}{d}\|\mathbf{h}_0\|_1)$, $b \leftarrow \log(\frac{1}{d}\|\mathbf{h}_0\|_1)$
   Obtain the estimated $\hat{K}(\lambda)$ with $\Lambda = [\lambda_1, \lambda_2]$ using equation 9 (Appendix Algorithm 2)
   **for** epoch = $1 : T_1$ **do**                                               ▷ **Stage 1**
      **for** sampling one batch $s$ from $\mathcal{S}$ **do**
         $\lambda \leftarrow \exp(b)$, $\boldsymbol{\sigma} \leftarrow \exp(\mathbf{v})$                        ▷ Ensure non-negative variances
         $\mathcal{P}_\lambda \leftarrow \mathcal{N}(\mathbf{h}_0; \lambda I_d)$, $\mathcal{Q}_{\boldsymbol{\sigma}}(\mathbf{h}) \leftarrow \mathbf{h} + \mathcal{N}(\mathbf{0}; \text{diag}(\boldsymbol{\sigma}))$
         Draw one $\tilde{\mathbf{h}} \sim \mathcal{Q}_{\boldsymbol{\sigma}}(\mathbf{h})$ and evaluate $\ell(\tilde{\mathbf{h}}; \mathcal{S})$,    ▷ Stochastic version of $\mathbb{E}_{\tilde{\mathbf{h}} \sim \mathcal{Q}_{\boldsymbol{\sigma}}(\mathbf{h})} \ell(\tilde{\mathbf{h}}; \mathcal{S})$
         Compute the KL divergence as equation 5
         Compute $\gamma$ as equation 6
         Compute the loss function $\mathcal{L}$ as $L_{PAC}$ in equation P
         $b \leftarrow b + \eta\frac{\partial \mathcal{L}}{\partial b}$, $\mathbf{v} \leftarrow \mathbf{v} + \eta\frac{\partial \mathcal{L}}{\partial \mathbf{v}}$, $\mathbf{h} \leftarrow \mathbf{h} + \eta\frac{\partial \mathcal{L}}{\partial \mathbf{h}}$          ▷ Update all parameters
      **end for**
   **end for**
   $\hat{\boldsymbol{\sigma}} \leftarrow \exp(\mathbf{v})$                                 ▷ Fix the noise level from now on
   **while** not converge **do**                                       ▷ **Stage 2**
      **for** sampling one batch $s$ from $\mathcal{S}$ **do**
         Draw one sample $\tilde{\mathbf{h}} \sim \mathcal{Q}_{\hat{\boldsymbol{\sigma}}}(\mathbf{h})$ and evaluate $\ell(\tilde{\mathbf{h}}; \mathcal{S})$ as $\tilde{\mathcal{L}}$,         ▷ Noise injection
         $\mathbf{h} \leftarrow \mathbf{h} + \eta\frac{\partial \tilde{\mathcal{L}}}{\partial \mathbf{h}}$                          ▷ Update model parameters
      **end for**
   **end while**
   $\hat{\mathbf{h}} \leftarrow \mathbf{h}$

---

Therefore, we only need to perform gradient updates on the other three variables, $\mathbf{h}, \boldsymbol{\sigma}, \boldsymbol{\lambda}$.

**The second stage of training:** Gastpar et al. (2023); Nagarajan & Kolter (2019) showed that achieving high accuracy on certain distributions precludes the possibility of getting a tight generalization bound in overparameterized settings. This also implies that it is less possible to use reasonable generalization bound to fully train one overparameterized model on a particular dataset. By minimizing the PAC-Bayes bound only, it is also observed in our PAC-Bayes training that the training accuracy can not be $100\%$. Therefore, we add a second stage to ensure convergence of the training loss. Specifically, in Stage 2, we continue to update the model by minimizing only $\mathbb{E}_{\mathbf{h} \sim \mathcal{Q}_{\boldsymbol{\sigma}}} \ell(\mathbf{h}; \mathcal{S})$ over $\mathbf{h}$, and keep all other variables (i.e., $\boldsymbol{\lambda}, \boldsymbol{\sigma}$) fixed to the solution found by Stage 1. This is essentially a stochastic gradient descent with noise injection, the level of which has been learned from Stage 1. At the high level, this two-stage training is similar to the idea of learning-rate scheduler (LRS). In LRS, the initial large learning rate introduces an implicit bias that guides the solution path towards a flat region (Cohen et al., 2021; Barrett & Dherin, 2020), and the later smaller learning rate ensures the convergence to a local minimizer in this region. Without the large learning rate stage, it cannot reach the flat region, and without the small learning rate, it cannot converge to a local minimizer. For the two-stage PAC-Bayes training, Stage 1 (PAC-Bayes stage) guides the solution to flat regions by minimizing the generalization bound, and Stage 2 is necessary for an actual convergence to a local minimizer. The practical impacts of each stage are also very similar to those of the LRS.

**Prediction:** After training, we use the mean of the posterior as the trained model and perform deterministic prediction on the test dataset. In Appendix A.6, we provide some mathematical intuition of why the deterministic predictor is expected to perform even better than the Bayesian predictor.

**Regularizations in the PAC-Bayes training:** By plugging the KL divergence (equation 5) into P, we can see that in the case of Gaussian priors and posteriors, the PAC-Bayes loss is nothing but the original training loss augmented by a noise injection and a weight decay, except that the weight decay term is now centered at $\mathbf{h}_0$ instead of $\mathbf{0}$, the coefficients of the weight decay change from layer to layer when using layerwise prior, and the noise injection has anisotropic variances. Since many factors in normal training, such as mini-batch and dropout, enhance generalization by some sort of noise injection, it is not surprising that they can be substituted by the well-calibrated noise injection in PAC-Bayes training. More discussions are available in Appendix A.4.

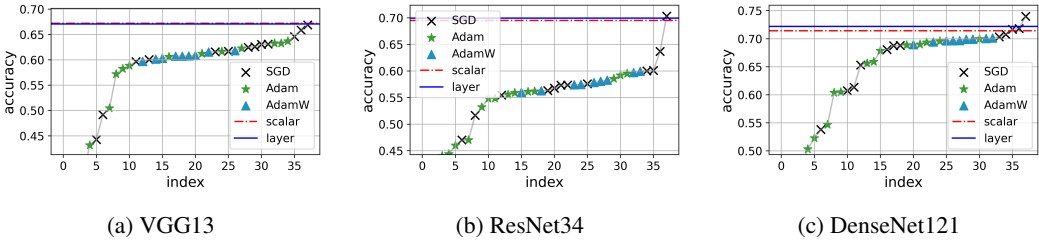

| (a) VGG13 | (b) ResNet34 | (c) DenseNet121 |

Figure 1: Sorted testing accuracy of CIFAR100. "scalar" and "layer" represent the tuning-free PAC-Bayes training with the scalar and layerwise prior.

## 6 EXPERIMENTS

**Evaluation on deep convolution neural networks:** We test the proposed method on CIFAR10 and CIFAR100 datasets with *no data augmentation*[4] on various popular deep neural networks, VGG13, VGG19 (Simonyan & Zisserman, 2014), ResNet18, ResNet34 (He et al., 2016), and DenseNet121 (Huang et al., 2017) by comparing its performance with the normal training by SGD and Adam with various regularizations (which we call baselines). The training of baselines involves a grid search of hyper-parameters, including momentum for SGD (0.3 to 0.9), learning rate (1e-3 to 2e-1), weight decay (1e-4 to 1e-2), and noise injection (5e-4 to 1e-2); thus, it is quite time-consuming. We used an extensive grid search as a baseline to ensure the best achievable test accuracy in the literature, e.g., Table 4 of Geiping et al. (2021). We compared the mean testing accuracy of the last five epochs to determine each optimal hyper-parameter. We only applied noise injection to Adam/AdamW, as it sometimes causes instability to SGD. We set batch size as 128, since it usually gives the best performance from the literature. Finally, 38 searches were conducted for each baseline model on one dataset. The testing accuracy for the optimally tuned baseline and a single run of the PAC-Bayes training are presented in Table 1. As the CIFAR10 and CIFAR100 lack a published validation dataset, we used the test dataset to adjust hyper-parameters for baselines. This might lead to a slightly inflated performance for baselines, offering them an unfair advantage over our method. Nevertheless, the testing accuracy of our method with scalar and layerwise prior matches the best testing accuracy of baselines. There is no grid search for our PAC-Bayes training, and we use Adam as the optimizer and the learning rate as $10^{-4}$ for all models. To provide more details, we have plotted all the searched results for baselines for VGG13, ResNet34, and DenseNet121 on CIFAR100 in Figure 1. The plotted results are sorted in ascending order based on their testing accuracy. The figure shows that our training algorithms are better than most searched settings.

Additionally, we find that the result of the tuning-free PAC-Bayes training is insensitive to the batch size. Table 2 shows that changing the batch size from 128 to 2048 for VGG13 and ResNet18 does not decrease the performance of the PAC-Bayes training as much as it does for the normal training. Despite not requiring exhaustive tuning, our proposed tuning-free algorithms match the best testing accuracy of baselines with a large batch size. This observation enables the use of large batch sizes for the PAC-Bayes training to accelerate the convergence. Besides the batch size, our proposed method is also insensitive to the learning rate. Please refer to Appendix C.2, C.3 C.5 for more training details and results. Training cost is also discussed in Appendix C.6. In short, PAC-Bayes training nearly doubles the parameters needing optimization, yet the required backpropagations per run are comparable to baselines.

**Evaluation on graph neural networks:** To demonstrate the robustness of the proposed PAC-Bayes training algorithm across different network architectures, we also test it on graph neural networks. Moreover, the number of training samples for node classification tasks is generally much smaller than CIFAR10/100, allowing us to examine the performance of the algorithm in the data scarcity setting. Unlike CNNs, the GNN baselines find their best performance with AdamW optimizer and with dropout turned on, while the proposed PAC-Bayes training algorithm stays the same as in the CNN setting. To ensure the best results of baselines, we searched the learning rate, weight decay, noise injection, and dropout of baselines. We follow the convention for graph datasets by randomly assigning 20 nodes per class for training, 500 for validation, and the remaining for testing. We evaluated the accuracy of the validation nodes to identify the

---

[4]Result with data augmentation can be found in Appendix C.4

Table 1: Testing accuracy of convolution neural networks on C10 (CIFAR10) and C100 (CIFAR100). "scalar" and "layer" denote the PAC-Bayes training with scalar and layerwise prior.

| (a) VGG13 | C10 | C100 |
|---|---|---|
| SGD | 90.2 | 66.9 |
| Adam | 88.5 | 63.7 |
| AdamW | 88.4 | 61.8 |
| scalar | 88.7 | 67.2 |
| layer | 89.7 | 67.1 |

| (b) VGG19 | C10 | C100 |
|---|---|---|
| SGD | 90.2 | 64.5 |
| Adam | 89.0 | 58.8 |
| AdamW | 89.0 | 62.3 |
| scalar | 89.2 | 61.3 |
| layer | 90.5 | 62.3 |

| (c) Resnet18 | C10 | C100 |
|---|---|---|
| SGD | 89.9 | 64.0 |
| Adam | 87.5 | 61.6 |
| AdamW | 87.9 | 61.4 |
| scalar | 88.0 | 68.8 |
| layer | 89.3 | 68.9 |

| (d) Resnet34 | C10 | C100 |
|---|---|---|
| SGD | 90.0 | 70.3 |
| Adam | 87.9 | 59.5 |
| AdamW | 88.3 | 59.9 |
| scalar | 89.6 | 69.5 |
| layer | 90.9 | 69.9 |

| (e) DenseNet121 | C10 | C100 |
|---|---|---|
| SGD | 91.8 | 74.0 |
| Adam | 91.2 | 70.0 |
| AdamW | 91.5 | 70.1 |
| scalar | 91.2 | 71.4 |
| layer | 91.5 | 72.2 |

Table 2: The results of large batch sizes on the testing accuracy of CNNs on C10 (CIFAR10) and C100 (CIFAR100). The number in $(\cdot)$ indicates how much the results differ from using a small batch size $(128)$. "scalar" and "layer" denote the PAC-Bayes training with scalar and layerwise prior. The best results are **highlighted**.

| (a) VGG13 | C10 | C100 |
|---|---|---|
| SGD | 87.7 (-2.5) | 60.1 (-6.8) |
| Adam | **90.7** (+2.2) | 66.2 (+2.5) |
| AdamW | 87.2 (-1.1) | 61.0 (-0.8) |
| scalar | 88.9 (0.2) | 66.0 (-1.2) |
| layer | 89.4 (-0.3) | **67.1** (-0.0) |

| (b) ResNet18 | C10 | C100 |
|---|---|---|
| SGD | 85.4 (-4.5) | 61.5 (-2.6) |
| Adam | 87.7 (+0.2) | 65.4 (+3.8) |
| AdamW | 84.9 (-2.9) | 58.9 (-2.5) |
| scalar | 88.9 (0.9) | 68.7 (-0.1) |
| layer | **89.2** (-0.1) | **69.3** (+0.3) |

| (c) ResNet34 | C10 | C100 |
|---|---|---|
| SGD | 87.0 (-3.0) | 61.5 (-8.8) |
| Adam | 89.5 (+1.6) | 67.1 (+7.6) |
| AdamW | 86.8 (-1.5) | 58.8 (-1.1) |
| scalar | 90.2 (0.6) | 67.5 (-2.0) |
| layer | **90.6** (-0.3) | **69.1** (-0.8) |

best hyper-parameters and report the corresponding testing accuracy. We tested GCN (Kipf & Welling, 2016), GAT (Veličković et al., 2017), SAGE (Hamilton et al., 2017), and APPNP (Gasteiger et al., 2018) on CoraML, Citeseer, PubMed, Cora and DBLP (Bojchevski & Günnemann, 2017).

There are only two convolution layers for GNNs, so we only test our algorithm with the scalar prior. We added one dropout layer between two graph convolution layers for baselines only, except keeping the dropout in the attention layers of GAT for both our algorithm and baselines since it essentially drops the input graph's edges. In Table 3, for each GNN architecture, the rows of AdamW and scalar, respectively, record the performance of baselines and the PAC-Bayes training with early stopping determined by the validation dataset. We see that our algorithm's results match the best

Table 3: Testing accuracy of GNNs. "AdW" is AdamW, the baseline, and "scalar" is our algorithm with a scalar prior.

| | | CoraML | Citeseer | PubMed | Cora | DBLP |
|---|---|---|---|---|---|---|
| GCN | AdW | 85.7±0.7 | 90.3±0.4 | 85.0±0.6 | 60.7±0.7 | 80.6±1.4 |
| | scalar | 86.1±0.7 | 90.0±0.4 | 84.9±0.8 | 62.0±0.4 | 80.5±0.6 |
| GAT | AdW | 85.7±1.0 | 90.8±0.3 | 84.0±0.4 | 63.5±0.4 | 81.8±0.6 |
| | scalar | 85.9±0.8 | 90.6±0.5 | 84.4±0.5 | 60.9±0.6 | 81.0±0.5 |
| SAGE | AdW | 85.7±0.5 | 90.5±0.5 | 83.5±0.4 | 60.6±0.5 | 80.7±0.6 |
| | scalar | 86.5±0.5 | 90.0±0.5 | 84.4±0.6 | 61.2±0.2 | 79.9±0.5 |
| APPNP | AdW | 86.6±0.7 | 91.0±0.4 | 85.1±0.5 | 62.5±0.4 | 80.6±2.8 |
| | scalar | 87.1±0.6 | 90.4±0.5 | 85.7±0.4 | 63.5±0.4 | 81.8±0.5 |

baseline testing accuracy. Appendix C.7 provides additional training information and evaluation outcomes. Extra analysis on few-shot text classification with transformers is in Appendix C.8.

## 7 CONCLUSION AND FUTURE WORK

This paper demonstrated that the PAC-Bayes bound can go beyond a purely theoretical bound and prove valuable during training. It not only attains a state-of-the-art performance level but also achieves auto-tuning. Moreover, an intriguing observation in this paper is that PAC-Bayes training did not seem to suffer from the curse of dimensionality, a phenomenon traditionally believed as a significant limitation of the generalization bounds. The PAC-Bayes framework can also provide the posterior variance of the weights, which can potentially be used to determine the importance of the weights. These may lead to new algorithm designs for network pruning and adversarial training.

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

## APPENDIX A  ALGORITHM DETAILS

### A.1  KL DIVERGENCE OF THE GAUSSIAN PRIOR AND POSTERIOR

For a $k$-layer network, the prior is written as $\mathcal{P}_{\boldsymbol{\lambda}}(\mathbf{h}_0)$, where $\mathbf{h}_0$ is the random initialized model and $\boldsymbol{\lambda} \in \mathbb{R}_+^k$ is the vector containing the variance for each layer. The set of all such priors is denoted by $\mathfrak{P} := \{\mathcal{P}_{\boldsymbol{\lambda}}(\mathbf{h}_0), \boldsymbol{\lambda} \in \Lambda \subseteq \mathbb{R}^k, \mathbf{h}_0 \in \mathcal{H}\}$. In the PAC-Bayes training, we select the posterior distribution to be centered around the trained model $\mathbf{h}$, with independent anisotropic variance. Specifically, for a network with $d$ trainable parameters, the posterior is $\mathcal{Q}_{\boldsymbol{\sigma}}(\mathbf{h}) := \mathcal{N}(\mathbf{h}, \mathrm{diag}(\boldsymbol{\sigma}))$, where $\mathbf{h}$ (the current model) is the mean and $\boldsymbol{\sigma} \in \mathbb{R}_+^d$ is the vector containing the variance for each trainable parameter. The set of all posteriors is $\mathfrak{Q} := \{\mathcal{Q}_{\boldsymbol{\sigma}}(\mathbf{h}), \boldsymbol{\sigma} \in \Sigma, \mathbf{h} \in \mathcal{H}\}$, and the KL divergence between all such prior and posterior in $\mathfrak{P}$ and $\mathfrak{Q}$ is:

$$\mathrm{KL}(\mathcal{Q}_{\boldsymbol{\sigma}}(\mathbf{h})||\mathcal{P}_{\boldsymbol{\lambda}}(\mathbf{h}_0)) = \frac{1}{2}\sum_{i=1}^k\left[-\mathbf{1}_{d_i}^\top \ln(\boldsymbol{\sigma}_i) + d_i(\ln(\lambda_i)-1) + \frac{\|\boldsymbol{\sigma}_i\|_1 + \|(\mathbf{h}-\mathbf{h}_0)_i\|^2)}{\lambda_i}\right], \quad (7)$$

where $\boldsymbol{\sigma}_i, (\mathbf{h}-\mathbf{h}_0)_i$ are vectors denoting the variances and weights for the $i$-th layer, respectively, and $\lambda_i$ is the scalar variance for the $i$-th layer. $d_i = \dim(\boldsymbol{\sigma}_i)$, and $\mathbf{1}_{d_i}$ denotes an all-ones vector of length $d_i$[5].

Scalar prior is a special case of the layerwise prior by setting all entries of $\boldsymbol{\lambda}$ to be equal, for which the KL divergence reduces to

$$\mathrm{KL}(\mathcal{Q}_{\boldsymbol{\sigma}}(\mathbf{h})||\mathcal{P}_{\lambda}(\mathbf{h}_0)) = \frac{1}{2}\left[-\mathbf{1}_d^\top \ln(\boldsymbol{\sigma}) + d(\ln(\lambda)-1) + \frac{1}{\lambda}(\|\boldsymbol{\sigma}\|_1 + \|\mathbf{h}-\mathbf{h}_0\|^2)\right]. \quad (8)$$

### A.2  ALGORITHMS TO ESTIMATE $K(\boldsymbol{\lambda})$

For a discrete set, $\{\boldsymbol{\lambda}_1,....,\boldsymbol{\lambda}_s\} \subseteq \Lambda$, we estimate the corresponding $K_1,....K_s$ using the empirical version of equation 2, that is for any $i = 1,...,s$, we solve

$$K_{\min}(\boldsymbol{\lambda}_i) = \arg\min_{K>0} K$$
$$\text{s.t. } \exp\left(\gamma^2 K\right) \geq \frac{1}{nm}\sum_{l=1}^n\sum_{j=1}^m \exp(\gamma(\ell(\mathbf{h}_l;\mathcal{S}) - \ell(\mathbf{h}_l;z_j))), \quad \forall\, \gamma \in [\gamma_1,\gamma_2], \qquad (9)$$

where $\mathbf{h}_l \sim \mathcal{P}_{\boldsymbol{\lambda}_i}(\mathbf{h}_0), l = 1,...,n$, are samples from the prior distribution and are fixed when solving equation 9 for $K_{\min}(\boldsymbol{\lambda}_i)$. This optimization problem can be solved by a bisection search. From the pairs $(\boldsymbol{\lambda}_i, K_{\min}(\boldsymbol{\lambda}_i))$, we construct $K_{\min}(\boldsymbol{\lambda})$ using piecewise-linear interpolation.

---

**Algorithm 2** Compute $K(\boldsymbol{\lambda})$ given a set of query priors

---

**Input:** $\gamma_1$ and $\gamma_2$, $s$ query prior variances $\mathcal{V} = \{\boldsymbol{\lambda}_i \in \Lambda \subseteq \mathbb{R}^k, i = 1,...,s\}$, the initial neural network weight $\mathbf{h}_0$, the training dataset $\mathcal{S} = \{z_i\}_{i=1}^m$, model sampling time $n = 10$
**Output:** the piece-wise linear interpolation $\tilde{K}(\boldsymbol{\lambda})$ for $K_{\min}(\boldsymbol{\lambda})$
  **for** $\boldsymbol{\lambda}_i \in \mathcal{V}$ **do**
    Set up a discrete grid $\Gamma$ for the interval $[\gamma_1,\gamma_2]$ of $\gamma$.
    **for** $l = 1 : n$ **do**
      Sampling weights from the Gaussian distribution $\mathbf{h}_l \sim \mathcal{N}(\mathbf{h}_0, \boldsymbol{\lambda}_i)$
      Use $\mathbf{h}_l$, $\Gamma$ and $\mathcal{S}$ to compute one term in the sum in equation 9
    Solve $\hat{K}_i$ using equation 9
  Fit a piece-wise linear function $\tilde{K}(\boldsymbol{\lambda})$ to the data $\{(\boldsymbol{\lambda}_i, \hat{K}_i)\}_{i=1}^s$

---

It is simple to evaluate $\boldsymbol{\lambda}$ when $k = 1$ using the algorithm above. When using the layerwise prior version of our method, the exponential moment bound $K_{\min}(\boldsymbol{\lambda})$ is a $k$-dimensional function, and

---

[5]Note that with a little ambiguity, the $\boldsymbol{\lambda}_i$ here has a different meaning from that in equation 9 and Algorithm 2, here $\boldsymbol{\lambda}_i$ means the $i$th element in $\boldsymbol{\lambda}$, whereas in equation 9 and Algorithm 2, $\boldsymbol{\lambda}_i$ means the $i$th element in the discrete set.

estimating it accurately requires a large number of samples. In this paper, we adopt a simple empirical approach to address this issue. We assume that the original function $K_{\min}(\boldsymbol{\lambda})$ can be well-approximated by a one-dimensional function $\tilde{K}$ that solely depends on the mean value of the input. That is, we assume there is a one-dimensional function $\tilde{K}$ such that $K_{\min}(\boldsymbol{\lambda}) \approx \tilde{K}(\bar{\boldsymbol{\lambda}})$, where $\bar{\boldsymbol{\lambda}}$ denotes the mean of $\boldsymbol{\lambda}$. Then we only need to estimate this one-dimensional function $\tilde{K}$ through function interpolation. We note that one can always choose to directly interpolate the $k$-dimensional function $K_{\min}(\boldsymbol{\lambda})$ directly, but at the expense of increased sampling complexity and computational time.

### A.3 PAC-BAYES TRAINING WITH LAYERWISE PRIOR

Similar to Algorithm 1, the PAC-Bayes training with layerwise prior is stated here in Algorithm 3.

---

**Algorithm 3** Tuning-free PAC-Bayes training (layerwise prior)

---

**Input:** initial model $\mathbf{h}_0 \in \mathbb{R}^d$, the number of layers $k$, $T_1$, $\lambda_1 = e^{-7}$, $\lambda_2 = 1$, $\gamma_1 = 0.5$, $\gamma_2 = 10$
**Output:** trained model $\hat{\mathbf{h}}$, posterior noise level $\hat{\boldsymbol{\sigma}}$

$\quad \mathbf{h} \leftarrow \mathbf{h}_0, \mathbf{v} \leftarrow \mathbf{1}_d \cdot \log(\frac{1}{d}\sum_{i=1}^{d} |\mathbf{h}_{0,i}|), \mathbf{b} \leftarrow \mathbf{1}_k \cdot \log(\frac{1}{d}\sum_{i=1}^{d} |\mathbf{h}_{0,i}|)$ $\qquad\qquad$ ▷ Initialization
$\quad$ Obtain the estimated $\tilde{K}(\bar{\boldsymbol{\lambda}})$ with $\Lambda = [\lambda_1, \lambda_2]^k$ using equation 9 and Appendix A.2
$\quad$ **for** epoch = $1 : T_1$ **do** $\qquad\qquad\qquad\qquad\qquad\qquad\qquad\qquad\qquad\qquad\qquad$ ▷ **Stage 1**
$\quad\quad$ **for** sampling one batch $s$ from $\mathcal{S}$ **do**
$\quad\quad\quad$ $\boldsymbol{\lambda} \leftarrow \exp(\mathbf{b}), \boldsymbol{\sigma} \leftarrow \exp(\mathbf{v})$ $\qquad\qquad\qquad\qquad$ ▷ Ensure non-negative variances
$\quad\quad\quad$ Construct the covariance of $\mathcal{P}_{\boldsymbol{\lambda}}$ from $\boldsymbol{\lambda}$ ▷ Setting the variance of the weights in layer-$i$ all
$\quad$ to the scalar $\boldsymbol{\lambda}(i)$
$\quad\quad\quad$ Draw one $\tilde{\mathbf{h}} \sim \mathcal{Q}_{\boldsymbol{\sigma}}(\mathbf{h})$ and evaluate $\ell(\tilde{\mathbf{h}}; \mathcal{S})$, $\qquad$ ▷ Stochastic version of $\mathbb{E}_{\tilde{\mathbf{h}} \sim \mathcal{Q}_{\boldsymbol{\sigma}}(\mathbf{h})} \ell(\tilde{\mathbf{h}}; \mathcal{S})$
$\quad\quad\quad$ Compute the KL-divergence as equation 7
$\quad\quad\quad$ Compute $\gamma$ as equation 6
$\quad\quad\quad$ Compute the loss function $\mathcal{L}$ as $L_{PAC}$ in equation P
$\quad\quad\quad$ $\mathbf{b} \leftarrow \mathbf{b} + \eta\frac{\partial\mathcal{L}}{\partial\mathbf{b}}, \mathbf{v} \leftarrow \mathbf{v} + \eta\frac{\partial\mathcal{L}}{\partial\mathbf{v}}, \mathbf{h} \leftarrow \mathbf{h} + \eta\frac{\partial\mathcal{L}}{\partial\mathbf{h}}$ $\qquad\qquad$ ▷ Update all parameters
$\quad\quad$ **end for**
$\quad$ **end for**
$\quad$ $\hat{\boldsymbol{\sigma}} \leftarrow \exp(\mathbf{v})$ $\qquad\qquad\qquad\qquad\qquad\qquad$ ▷ Fix the noise level from now on
$\quad$ **while** not converge **do** $\qquad\qquad\qquad\qquad\qquad\qquad\qquad\qquad\qquad$ ▷ **Stage 2**
$\quad\quad$ **for** sampling one batch $s$ from $\mathcal{S}$ **do**
$\quad\quad\quad$ Draw one sample $\tilde{\mathbf{h}} \sim \mathcal{Q}_{\hat{\boldsymbol{\sigma}}}(\mathbf{h})$ and evaluate $\ell(\tilde{\mathbf{h}}; \mathcal{S})$ as $\tilde{\mathcal{L}}$, $\qquad$ ▷ Noise injection
$\quad\quad\quad$ $\mathbf{h} \leftarrow \mathbf{h} + \eta\frac{\partial\tilde{\mathcal{L}}}{\partial\mathbf{h}}$ $\qquad\qquad\qquad\qquad\qquad$ ▷ Update model parameters
$\quad\quad$ **end for**
$\quad$ **end while**
$\quad$ $\hat{\mathbf{h}} \leftarrow \mathbf{h}$

---

### A.4 REGULARIZATIONS IN PAC-BAYES BOUND

Only noise injection and weight decay are essential from our derived PAC-Bayes bound. Like most commonly used implicit regularizations (large lr, momentum, small batch size), dropout and batch-norm are also known to penalize the loss function's sharpness indirectly. Wei et al. (2020) studies that dropout introduces an explicit regularization that penalizes *sharpness* and an implicit regularization that is analogous to the effect of stochasticity in small mini-batch stochastic gradient descent. Similarly, it is well-studied that batch-norm Luo et al. (2018) allows the use of a large learning rate by reducing the variance in the layer batches, and large allowable learning rates regularize *sharpness* through the edge of stability Cohen et al. (2020). As shown in the equation below, the first term (noise-injection) in our PAC-Bayes bound explicitly penalizes the Trace of the Hessian of the loss, which directly relates to sharpness and is quite similar to the regularization effect of batch-norm and dropout. During training, suppose the current posterior is $\mathcal{Q}_{\hat{\sigma}}(\hat{h}) = \mathcal{N}(\hat{h}, \text{diag}(\hat{\sigma}))$, then the training

loss expectation over the posterior is:

$$
\mathbb{E}_{h \sim \mathcal{Q}_{\hat{\sigma}}(\hat{h})} \ell(h; \mathcal{D}) = \mathbb{E}_{\Delta h \sim \mathcal{Q}_{\hat{\sigma}}(0)} \ell(\hat{h} + \Delta h; \mathcal{D})
$$
$$
\approx \ell(\hat{h}, \mathcal{D}) + \mathbb{E}_{\Delta h \sim \mathcal{Q}_{\hat{\sigma}}(0)} (\ell(\hat{h}; \mathcal{D}) \Delta h + \frac{1}{2} \Delta h^\top \nabla^2 \ell(\hat{h}; \mathcal{D}) \Delta h)
$$
$$
= \ell(\hat{h}; \mathcal{D}) + \frac{1}{2} \mathrm{Tr}(\mathrm{diag}(\hat{\sigma}) \nabla^2 \ell(\hat{h}; \mathcal{D})).
$$

The second regularization term (weight decay) in the bound additionally ensures that the minimizer found is close to initialization. Although the relation of this regularizer to sharpness is not very clear, empirical results suggest that weight decay may have a separate regularization effect from sharpness. So in brief, we state that the effect of sharpness regularization from dropout and batch norm can also be well emulated by noise injection with the additional effect of weight decay.

### A.5 LIMITATIONS OF THE PROPOSED PAC-BAYES TRAINING

We've showcased the significant potential of PAC-Bayes training. However, it's important to note its limitations to inspire future works:

1. In conventional training, the weights of the neural network $\mathbf{h}$ is the only parameter to be stored and updated. In PAC-Bayes training, we have four parameters $\mathbf{h}, \boldsymbol{\lambda}, \boldsymbol{\sigma}, \gamma$. Among these variables, $\gamma$ can be computed on the fly or whenever needed. We need to store $\mathbf{h}, \boldsymbol{\lambda}, \boldsymbol{\sigma}$, where $\boldsymbol{\sigma}$ has the same size as $\mathbf{h}$ and $\boldsymbol{\lambda}$ is much smaller. Hence the total storage is approximately doubled. Likewise, we now need to compute the gradient for $\mathbf{h}, \boldsymbol{\lambda}, \boldsymbol{\sigma}$. However, since we need to do only one backpropagation and use the result to compute the gradients for both $\mathbf{h}$ and $\boldsymbol{\sigma}$, the cost of automatic differentiation in each iteration is also approximately the same. In the inference stage, the complexity is the same as in conventional training. The real training complexity is discussed in Appendix C.6, showing that the PAC-Bayes training needs a much shorter time to achieve the best generalization with the comparable average number of backpropagation compared with baselines, although a single run of the baseline is faster.

2. The additional parameters to be optimized in PAC-Bayes training inevitably increase the difficulty of the optimization, and the most direct consequence is that there are more parameters to be initialized. In most of the experiments we ran, the recommended initialization of $\mathbf{v}$ and $\mathbf{b}$ work well. Rarely is it necessary to modify this setup. But if it occurs (i.e., in some settings, the recommended noise initializations are too large or too small), then the convergence of the algorithm would usually be affected immediately after the training started. So, if one observes a stall in the training accuracy in the early iterations, it usually indicates that the noise initialization requires adjustment, and a simple adjustment of multiplying the noise level by a global scalar often suffices. The tuning of initialization for PAC-Bayes training (most of the time unnecessary) can be performed much more efficiently than the hyper-parameter tuning for the baseline, as inappropriate noise initializations lead to convergence problems appearing right after the training starts, whereas for the hyper-parameter turning, one needs to wait until the completion of the entire training process to evaluate the effectiveness of the current parameter configuration, which is much more time-consuming.

### A.6 DETERMINISTIC PREDICTION

Recall that for any $\mathbf{h} \in \mathbb{R}^d$ and $\boldsymbol{\sigma} \in \mathbb{R}_+^d$, we used $\mathcal{Q}_{\boldsymbol{\sigma}}(\mathbf{h})$ to denote the multivariate normal distribution with mean $\mathbf{h}$ and covariance matrix $\mathrm{diag}(\boldsymbol{\sigma})$. If we rewrite the left-hand side of the PAC-Bayes bound by Taylor expansion, we have:

$$
\mathbb{E}_{\mathbf{h} \sim \mathcal{Q}_{\hat{\sigma}}(\hat{\mathbf{h}})} \ell(\mathbf{h}; \mathcal{D}) = \mathbb{E}_{\Delta \mathbf{h} \sim \mathcal{Q}_{\hat{\sigma}}(0)} \ell(\hat{\mathbf{h}} + \Delta \mathbf{h}; \mathcal{D})
$$
$$
\approx \ell(\hat{\mathbf{h}}, \mathcal{D}) + \mathbb{E}_{\Delta \mathbf{h} \sim \mathcal{Q}_{\hat{\sigma}}(0)} (\ell(\hat{\mathbf{h}}; \mathcal{D}) \Delta \mathbf{h} + \frac{1}{2} \Delta \mathbf{h}^\top \nabla^2 \ell(\hat{\mathbf{h}}; \mathcal{D}) \Delta \mathbf{h})
$$
$$
= \ell(\hat{\mathbf{h}}; \mathcal{D}) + \frac{1}{2} \mathrm{Tr}(\mathrm{diag}(\boldsymbol{\sigma}) \nabla^2 \ell(\hat{\mathbf{h}}; \mathcal{D})) \geq \ell(\hat{\mathbf{h}}, \mathcal{D}). \tag{10}
$$

Recall here $\hat{\mathbf{h}}$ and $\hat{\boldsymbol{\sigma}}$ are the minimizers of the PAC-Bayes loss, obtained by solving the optimization problem equation P. Equation equation 10 states that the deterministic predictor has a smaller

prediction error than the Bayesian predictor. However, note that the last inequality in equation 10 is derived under the assumption that the term $\nabla^2 \ell(\hat{\mathbf{h}}, \mathcal{D})$ is positive-semidefinite. This is a reasonable assumption as $\hat{\mathbf{h}}$ is the local minimizer of the PAC-Bayes loss and the PAC-Bayes loss is close to the population loss when the number of samples is large. Nevertheless, since this assumption does not holds for all cases, the presented argument can only serve only as an intuition that shows the potential benefits of using the deterministic predictor.

## APPENDIX B  PROOFS

### B.1  PROOFS OF THEOREM 4.1

**Theorem B.1.** *Given a prior $\mathcal{P}_{\boldsymbol{\lambda}}$ parametrized by $\boldsymbol{\lambda} \in \Lambda$ over the hypothesis set $\mathcal{H}$. Fix $\boldsymbol{\lambda} \in \Lambda$, $\delta \in (0, 1)$ and $\gamma \in [\gamma_1, \gamma_2]$. For any choice of i.i.d $m$-sized training dataset $\mathcal{S}$ according to $\mathcal{D}$, and all posterior distributions $\mathcal{Q}$ over $\mathcal{H}$, we have*

$$\mathbb{E}_{\mathbf{h}\sim\mathcal{Q}}\ell(\mathbf{h}; \mathcal{D}) \leq \mathbb{E}_{\mathbf{h}\sim\mathcal{Q}}\ell(\mathbf{h}; \mathcal{S}) + \frac{1}{\gamma m}(\ln\frac{1}{\delta} + \mathrm{KL}(\mathcal{Q}||\mathcal{P}_{\boldsymbol{\lambda}})) + \gamma K(\boldsymbol{\lambda}) \tag{11}$$

*holds with probability at least $1 - \delta$ when $\ell(\mathbf{h}, \cdot)$ satisfies Definition 2 with bound $K(\boldsymbol{\lambda})$.*

**Proof.** Firstly, in the bounded interval $\gamma \in [\gamma_1, \gamma_2]$, we bound the difference of the expected loss over the posterior distribution evaluated on the training dataset $\mathcal{S}$ and $\mathcal{D}$ with the KL divergence between the posterior distribution $\mathcal{Q}$ and prior distribution $\mathcal{P}_{\boldsymbol{\lambda}}$ evaluated over a hypothesis space $\mathcal{H}$.

For $\gamma \in [\gamma_1, \gamma_2]$,

$$\mathbb{E}_{\mathcal{S}\sim\mathcal{D}}[\exp\left(\gamma m(\mathbb{E}_{\mathbf{h}\sim\mathcal{Q}}\ell(\mathbf{h}; \mathcal{D}) - \mathbb{E}_{\mathbf{h}\sim\mathcal{Q}}\ell(\mathbf{h}; \mathcal{S})) - \mathrm{KL}(\mathcal{Q}||\mathcal{P}_{\boldsymbol{\lambda}}))\right]$$

$$=\mathbb{E}_{\mathcal{S}\sim\mathcal{D}}[\exp\left(\gamma m(\mathbb{E}_{\mathbf{h}\sim\mathcal{Q}}\ell(\mathbf{h}; \mathcal{D}) - \mathbb{E}_{\mathbf{h}\sim\mathcal{Q}}\ell(\mathbf{h}; \mathcal{S})) - \mathbb{E}_{\mathbf{h}\sim\mathcal{Q}}\log\frac{\mathrm{d}\mathcal{Q}}{\mathrm{d}\mathcal{P}_{\boldsymbol{\lambda}}}(\mathbf{h}))\right] \tag{12}$$

$$\leq\mathbb{E}_{\mathcal{S}\sim\mathcal{D}}\mathbb{E}_{\mathbf{h}\sim\mathcal{Q}}[\exp\left(\gamma m(\ell(\mathbf{h}; \mathcal{D}) - \ell(\mathbf{h}; \mathcal{S})) - \log\frac{\mathrm{d}\mathcal{Q}}{\mathrm{d}\mathcal{P}_{\boldsymbol{\lambda}}}(\mathbf{h}))\right] \tag{13}$$

$$=\mathbb{E}_{\mathcal{S}\sim\mathcal{D}}\mathbb{E}_{\mathbf{h}\sim\mathcal{Q}}[\exp(\gamma m(\ell(\mathbf{h}; \mathcal{D}) - \ell(\mathbf{h}; \mathcal{S})))\frac{\mathrm{d}\mathcal{P}_{\boldsymbol{\lambda}}}{\mathrm{d}\mathcal{Q}}(\mathbf{h})]$$

$$=\mathbb{E}_{\mathcal{S}\sim\mathcal{D}}\mathbb{E}_{\mathbf{h}\sim\mathcal{P}_{\boldsymbol{\lambda}}}[\exp(\gamma m(\ell(\mathbf{h}; \mathcal{D}) - \ell(\mathbf{h}; \mathcal{S})))\frac{\mathrm{d}\mathcal{P}_{\boldsymbol{\lambda}}}{\mathrm{d}\mathcal{Q}}(\mathbf{h})\frac{\mathrm{d}\mathcal{Q}}{\mathrm{d}\mathcal{P}_{\boldsymbol{\lambda}}}(\mathbf{h})] \tag{14}$$

$$=\mathbb{E}_{\mathbf{h}\sim\mathcal{P}_{\boldsymbol{\lambda}}}\mathbb{E}_{\mathcal{S}\sim\mathcal{D}}[\exp(\gamma m(\ell(\mathbf{h}; \mathcal{D}) - \ell(\mathbf{h}; \mathcal{S})))], \tag{15}$$

where $\mathrm{d}\mathcal{Q}/\mathrm{d}\mathcal{P}$ denotes the Radon-Nikodym derivative.

In equation 12, we use $\mathrm{KL}(\mathcal{Q}||\mathcal{P}_{\lambda}) = \mathbb{E}_{\mathbf{h}\sim\mathcal{Q}}\log\frac{\mathrm{d}\mathcal{Q}}{\mathrm{d}\mathcal{P}_{\lambda}}(\mathbf{h})$. From equation 12 to equation 13, Jensen's inequality is used over the convex exponential function. And in equation 14, the expectation is changed to the prior distribution from the posterior.

Let $X = \ell(\mathbf{h}; \mathcal{D}) - \ell(\mathbf{h}; \mathcal{S})$, then $X$ is centered with $\mathbb{E}[X] = 0$. Then, by Definition 2,

$$\exists K(\boldsymbol{\lambda}), \ \mathbb{E}_{\mathbf{h}\sim\mathcal{P}_{\lambda}}\mathbb{E}_{\mathcal{S}\sim\mathcal{D}}[\exp\left(\gamma mX\right)] \leq \exp\left(m\gamma^2 K(\boldsymbol{\lambda})\right). \tag{16}$$

Using Markov's inequality, equation 17 holds with probability at least $1 - \delta$.

$$\exp\left(\gamma mX\right) \leq \frac{\exp\left(m\gamma^2 K(\boldsymbol{\lambda})\right)}{\delta}. \tag{17}$$

Combining equation 15 and equation 17, the following inequality holds with probability at least $1 - \delta$.

$$\exp\left(\gamma m(\mathbb{E}_{\mathbf{h}\sim\mathcal{Q}}\ell(\mathbf{h}; \mathcal{D}) - \mathbb{E}_{\mathbf{h}\sim\mathcal{Q}}\ell(\mathbf{h}; \mathcal{S})) - \mathrm{KL}(\mathcal{Q}||\mathcal{P}_{\boldsymbol{\lambda}})\right) \leq \frac{\exp\left(m\gamma^2 K(\boldsymbol{\lambda})\right)}{\delta}$$

$$\Rightarrow \gamma m(\mathbb{E}_{\mathbf{h}\sim\mathcal{Q}}\ell(\mathbf{h}; \mathcal{D}) - \mathbb{E}_{\mathbf{h}\sim\mathcal{Q}}\ell(\mathbf{h}; \mathcal{S})) - \mathrm{KL}(\mathcal{Q}||\mathcal{P}_{\boldsymbol{\lambda}}) \leq \ln\frac{1}{\delta} + m\gamma^2 K(\boldsymbol{\lambda})$$

$$\Rightarrow \mathbb{E}_{\mathbf{h}\sim\mathcal{Q}}\ell(\mathbf{h}; \mathcal{D}) \leq \mathbb{E}_{\mathbf{h}\sim\mathcal{Q}}\ell(\mathbf{h}; \mathcal{S}) + \frac{1}{\gamma m}(\ln\frac{1}{\delta} + \mathrm{KL}(\mathcal{Q}||\mathcal{P}_{\boldsymbol{\lambda}})) + \gamma K(\boldsymbol{\lambda}) \tag{18}$$

The bound 18 is exactly the statement of the Theorem.

$\square$

## B.2 Proof of Theorem 4.2

**Theorem B.2.** *Let $n(\varepsilon) := \mathcal{N}(\Lambda, \|\cdot\|, \varepsilon)$ be the covering number of the set of the prior parameters $\Lambda$. Under Assumption 4.1.1 and Assumption 4.1.2, the following inequality holds for the minimizer $(\hat{\mathbf{h}}, \hat{\gamma}, \hat{\boldsymbol{\sigma}}, \hat{\boldsymbol{\lambda}})$ of upper bound in equation 3 with probability as least $1 - \epsilon$:*

$$\mathbb{E}_{\mathbf{h} \sim \mathcal{Q}_{\hat{\boldsymbol{\sigma}}}(\hat{\mathbf{h}})} \ell(\mathbf{h}; \mathcal{D}) \leq \mathbb{E}_{\mathbf{h} \sim \mathcal{Q}_{\hat{\boldsymbol{\sigma}}}(\hat{\mathbf{h}})} \ell(\mathbf{h}; \mathcal{S}) + \frac{1}{\hat{\gamma}m} \left[ \log \frac{n(\varepsilon)}{\epsilon} + \mathrm{KL}(\mathcal{Q}_{\hat{\boldsymbol{\sigma}}}(\hat{\mathbf{h}}) \| \mathcal{P}_{\hat{\boldsymbol{\lambda}}}) \right] + \hat{\gamma} K(\hat{\boldsymbol{\lambda}}) + \eta$$

$$= L_{PAC}(\hat{\mathbf{h}}, \hat{\gamma}, \hat{\boldsymbol{\sigma}}, \hat{\boldsymbol{\lambda}}) + \eta + \frac{\log(n(\varepsilon))}{\hat{\gamma}m} \tag{19}$$

*holds for any $\epsilon, \varepsilon > 0$, where $\eta = (\frac{1}{\gamma_1 m} + \gamma_2)(\eta_1(\varepsilon) + \eta_2(\varepsilon))$.*

***Proof:*** In this proof, we extend the PAC-Bayes bound 3 with data-independent priors to data-dependent ones that accommodate the error when the prior distribution is parameterized and optimized over a finite set of parameters $\mathfrak{P} = \{P_{\boldsymbol{\lambda}}, \boldsymbol{\lambda} \in \Lambda \subseteq \mathbb{R}^k\}$ with a much smaller dimension than the model itself. Let $\mathbb{T}(\Lambda, \|\cdot\|, \varepsilon)$ be an $\varepsilon$-cover of the set $\Lambda$, which states that for any $\boldsymbol{\lambda} \in \Lambda$, there exists a $\tilde{\boldsymbol{\lambda}} \in \mathbb{T}(\Lambda, \|\cdot\|, \varepsilon)$, such that $\|\boldsymbol{\lambda} - \tilde{\boldsymbol{\lambda}}\| \leq \varepsilon$.

Now we select the posterior distribution as $\mathcal{Q}_{\boldsymbol{\sigma}}(\mathbf{h}) := \mathbf{h} + \mathcal{Q}_{\boldsymbol{\sigma}}$, where $\mathbf{h}$ is the current model and $\mathcal{Q}_{\boldsymbol{\sigma}}$ is a zero mean distribution parameterized by $\boldsymbol{\sigma} \in \mathbb{R}^d$. Assuming the prior $\mathcal{P}$ is parameterized by $\boldsymbol{\lambda} \in \mathbb{R}^k$ ($k \ll m, d$).

Then the PAC-Bayes bound 3 holds already for any $(\hat{\mathbf{h}}, \hat{\gamma}, \hat{\boldsymbol{\sigma}}, \boldsymbol{\lambda})$, with fixed $\boldsymbol{\lambda} \in \Lambda$, i.e.,

$$\mathbb{E}_{\tilde{\mathbf{h}} \sim \mathcal{Q}_{\hat{\boldsymbol{\sigma}}}(\hat{\mathbf{h}})} \ell(\tilde{\mathbf{h}}; \mathcal{D}) \leq \mathbb{E}_{\tilde{\mathbf{h}} \sim \mathcal{Q}_{\hat{\boldsymbol{\sigma}}}(\hat{\mathbf{h}})} \ell(\tilde{\mathbf{h}}; \mathcal{S}) + \frac{1}{\hat{\gamma}m}(\ln \frac{1}{\delta} + \mathrm{KL}(\mathcal{Q}_{\hat{\boldsymbol{\sigma}}}(\hat{\mathbf{h}}) \| \mathcal{P}_{\boldsymbol{\lambda}})) + \hat{\gamma} K(\boldsymbol{\lambda}) \tag{20}$$

with probability over $1 - \delta$.

Now, for the collection of $\boldsymbol{\lambda}$s in the $\varepsilon$-net $\mathbb{T}(\Lambda, \|\cdot\|, \varepsilon)$, by the union bound, the PAC-Bayes bound uniformly holds with probability at least $1 - |\mathbb{T}|\delta = 1 - n\delta$. For an arbitrary $\boldsymbol{\lambda} \in \Lambda$, its distance to the $\varepsilon$-net is at most $\varepsilon$. Then under Assumption 4.1.1 and Assumption 4.1.2, we have:

$$\min_{\tilde{\boldsymbol{\lambda}} \in \mathbb{T}} |\mathrm{KL}(\mathcal{Q} \| \mathcal{P}_{\boldsymbol{\lambda}}) - \mathrm{KL}(\mathcal{Q} \| \mathcal{P}_{\tilde{\boldsymbol{\lambda}}})| \leq \eta_1(\|\boldsymbol{\lambda} - \tilde{\boldsymbol{\lambda}}\|) \leq \eta_1(\varepsilon),$$

and

$$\min_{\tilde{\boldsymbol{\lambda}} \in \mathbb{T}} |K(\boldsymbol{\lambda}) - K(\tilde{\boldsymbol{\lambda}})| \leq \eta_2(\|\boldsymbol{\lambda} - \tilde{\boldsymbol{\lambda}}\|) \leq \eta_2(\varepsilon).$$

With these two inequalities, we can control the PAC-Bayes loss at the given $\boldsymbol{\lambda}$ as follows:

$$\min_{\tilde{\boldsymbol{\lambda}} \in \mathbb{T}} |L_{PAC}(\hat{\mathbf{h}}, \hat{\gamma}, \hat{\boldsymbol{\sigma}}, \boldsymbol{\lambda}) - L_{PAC}(\hat{\mathbf{h}}, \hat{\gamma}, \hat{\boldsymbol{\sigma}}, \tilde{\boldsymbol{\lambda}})| \leq \frac{1}{\hat{\gamma}m}\eta_1(\varepsilon) + \hat{\gamma}\eta_2(\varepsilon)$$

$$\leq \frac{1}{\gamma_1 m}\eta_1(\varepsilon) + \gamma_2\eta_2(\varepsilon)$$

$$\leq C(\eta_1(\varepsilon) + \eta_2(\varepsilon))$$

where $C = \frac{1}{\gamma_1 m} + \gamma_2$ and $\gamma \in [\gamma_1, \gamma_2]$. Now, since this inequality holds for any $\boldsymbol{\lambda} \in \Lambda$, it certainly holds for the optima $\hat{\boldsymbol{\lambda}}$. Combining this with equation 20, we have

$$\mathbb{E}_{\mathbf{h} \sim \mathcal{Q}_{\hat{\boldsymbol{\sigma}}}(\hat{\mathbf{h}})} \ell(\mathbf{h}; \mathcal{D}) \leq L_{PAC}(\hat{\mathbf{h}}, \hat{\gamma}, \hat{\boldsymbol{\sigma}}, \hat{\boldsymbol{\lambda}}) + C(\eta_1(\varepsilon) + \eta_2(\varepsilon))$$

where $C := \frac{1}{\gamma_1 m} + \gamma_2$.

Now taking $\epsilon := n(\varepsilon)\delta$, we get, with probability $1 - \epsilon$, it holds that

$$\mathbb{E}_{\mathbf{h}\sim\mathcal{Q}_{\hat{\boldsymbol{\sigma}}}(\hat{\mathbf{h}})}\ell(\mathbf{h};\mathcal{D}) \leq \mathbb{E}_{\mathbf{h}\sim\mathcal{Q}_{\hat{\boldsymbol{\sigma}}}(\hat{\mathbf{h}})}\ell(\mathbf{h};\mathcal{S}) + \frac{1}{\hat{\gamma}m}\left[\ln\frac{n(\varepsilon)}{\epsilon} + \mathrm{KL}(\mathcal{Q}_{\hat{\boldsymbol{\sigma}}}(\hat{\mathbf{h}})||\mathcal{P}_{\hat{\boldsymbol{\lambda}}})\right] + \hat{\gamma}K(\hat{\boldsymbol{\lambda}}) + \eta$$

$$= L_{PAC}(\hat{\mathbf{h}}, \hat{\gamma}, \hat{\boldsymbol{\sigma}}, \hat{\boldsymbol{\lambda}}) + \eta + \frac{\ln(n(\varepsilon))}{\hat{\gamma}m} \tag{21}$$

and the proof is completed. $\qquad\square$

## B.3 Proof of Corollary 5.0.1

Recall for the training, we proposed to optimize over all four variables: $\mathbf{h}$, $\gamma$, $\boldsymbol{\sigma}$, and $\boldsymbol{\lambda}$.

$$(\hat{\mathbf{h}}, \hat{\gamma}, \hat{\boldsymbol{\sigma}}, \hat{\boldsymbol{\lambda}}) = \arg\min_{\substack{\mathbf{h},\boldsymbol{\lambda},\boldsymbol{\sigma}, \\ \gamma\in[\gamma_1,\gamma_2]}} \underbrace{\mathbb{E}_{\tilde{\mathbf{h}}\sim\mathcal{Q}_{\boldsymbol{\sigma}}(\mathbf{h})}\ell(\tilde{\mathbf{h}};\mathcal{S}) + \frac{1}{\gamma m}(\ln\frac{1}{\delta} + \mathrm{KL}(\mathcal{Q}_{\boldsymbol{\sigma}}(\mathbf{h})||\mathcal{P}_{\boldsymbol{\lambda}})) + \gamma K(\boldsymbol{\lambda})}_{\equiv L_{PAC}(\mathbf{h},\gamma,\boldsymbol{\sigma},\boldsymbol{\lambda})}. \tag{22}$$

**Corollary B.2.1.** *Assume all parameters for the prior and posterior are bounded, i.e., we restrict the model $\mathbf{h}$, the posterior variance $\boldsymbol{\sigma}$ and the prior variance $\boldsymbol{\lambda}$, all to be searched over bounded sets, $\mathcal{H} := \{\mathbf{h} \in \mathbb{R}^d : \|\mathbf{h}\|_2 \leq \sqrt{d}M\}$, $\Sigma := \{\boldsymbol{\sigma} \in \mathbb{R}_+^d : \|\boldsymbol{\sigma}\|_1 \leq dT\}$, $\Lambda =: \{\boldsymbol{\lambda} \in [e^{-a}, e^b]^k\}$, respectively, with fixed $M, T, a, b > 0$. Then,*

- *Assumption 4.1.1 holds with $\eta_1(x) = L_1 x$, where $L_1 = \frac{1}{2}\max\{d, e^a(2\sqrt{d}M + dT)\}$*

- *Assumption 4.1.2 holds with $\eta_2(x) = L_2 x$, where $L_2 = \frac{1}{\gamma_1^2}\left(2dM^2e^{2a} + \frac{d(a+b)}{2}\right)$*

- *With high probability, the PAC-Bayes bound for the minimizer of equation P has the form*

$$\mathbb{E}_{h\sim\mathcal{Q}_{\hat{\boldsymbol{\sigma}}}(\hat{\mathbf{h}})}\ell(\mathbf{h};\mathcal{D}) \leq L_{PAC}(\hat{\mathbf{h}}, \hat{\gamma}, \hat{\boldsymbol{\sigma}}, \hat{\boldsymbol{\lambda}}) + \eta,$$

*where $\eta = \frac{k}{\gamma_1 m}\left(1 + \ln\frac{CL(b+a)\gamma_1 m}{2k}\right)$, $C = \frac{1}{\gamma_1 m} + \gamma_2$ and $L = L_1 + L_2$.*

***Proof:*** We first prove the two assumptions are satisfied by the Gaussian family with bounded parameter spaces. To prove Assumption 4.1.1 is satisfied, let $v_i = \log 1/\lambda_i$, $i = 1, ..., k$ and perform a change of variable from $\lambda_i$ to $v_i$. The prior for the $i$th layer now becomes $\tilde{\mathcal{P}}_{\mathbf{v}_i} := \mathcal{P}_{\boldsymbol{\lambda}_i} = \mathcal{N}(\mathbf{0}, \lambda_i\mathbf{I}_{d_i})) = \mathcal{N}(\mathbf{0}, e^{-v_i}\mathbf{I}_{d_i}))$, where $d_i$ is the number of trainable parameters in the $i$th layer. It is straightforward to compute

$$\frac{\partial\mathrm{KL}(\mathcal{Q}_{\boldsymbol{\sigma}}||\tilde{\mathcal{P}}_{\mathbf{v}})}{\partial v_i} = \frac{1}{2}[-d_i + e^{v_i}(\|\boldsymbol{\sigma}_i\|_1 + \|\mathbf{h}_i - \mathbf{h}_{0,i}\|^2)],$$

where $\boldsymbol{\sigma}_i$, $\mathbf{h}_i$, $\mathbf{h}_{0,i}$ are the blocks of $\boldsymbol{\sigma}$, $\mathbf{h}$, $\mathbf{h}_0$, containing the parameters associated with the $i$th layer, respectively. Now, given the assumptions on the boundedness of the parameters, we have:

$$\|\nabla_{\mathbf{v}}\mathrm{KL}(\mathcal{Q}_{\boldsymbol{\sigma}}||\tilde{\mathcal{P}}_{\mathbf{v}})\|_2 \leq \|\nabla_{\mathbf{v}}\mathrm{KL}(\mathcal{Q}_{\boldsymbol{\sigma}}||\tilde{\mathcal{P}}_{\mathbf{v}})\|_1 \leq \frac{1}{2}\max\{d, e^a(2\sqrt{d}M + dT)\} \equiv L_1(d, M, T, a), \tag{23}$$

where we used the assumption $\|\boldsymbol{\sigma}\|_1 \leq dT$ and $\|\mathbf{h}_0\|_2, \|\mathbf{h}\|_2 \leq \sqrt{d}M$.

Equation 23 says $L_1(d, M, T, a)$ is a valid Lipschitz bound on the KL divergence and therefore Assumption 4.1.1 is satisfied by setting $\eta_1(x) = L_1(d, M, T, a)x$.

Next, we prove Assumption 4.1.2 is satisfied. We use $K_{\min}(\boldsymbol{\lambda})$ defined in Definition 2 as the $K(\boldsymbol{\lambda})$ in the PAC-Bayes training, and verify that it makes Assumption 4.1.2 hold.

$$|K_{\min}(\boldsymbol{\lambda}_1) - K_{\min}(\boldsymbol{\lambda}_2)|$$

$$= \left| \sup_{\gamma \in [\gamma_1, \gamma_2]} \frac{1}{\gamma^2} \log(\mathbb{E}_{\mathbf{h} \sim \mathcal{P}_{\boldsymbol{\lambda}_1}} \mathbb{E}_{z \sim \mathcal{D}}[\exp(\gamma \ell(\mathbf{h}, z))]) - \sup_{\gamma \in [\gamma_1, \gamma_2]} \frac{1}{\gamma^2} \log(\mathbb{E}_{\mathbf{h} \sim \mathcal{P}_{\boldsymbol{\lambda}_2}} \mathbb{E}_{z \sim \mathcal{D}}[\exp(\gamma \ell(\mathbf{h}, z))]) \right|$$

$$\leq \sup_{\gamma \in [\gamma_1, \gamma_2]} \frac{1}{\gamma^2} \left| \log(\mathbb{E}_{\mathbf{h} \sim \mathcal{P}_{\boldsymbol{\lambda}_1}} \mathbb{E}_{z \sim \mathcal{D}}[\exp(\gamma \ell(\mathbf{h}, z))]) - \log(\mathbb{E}_{\mathbf{h} \sim \mathcal{P}_{\boldsymbol{\lambda}_2}} \mathbb{E}_{z \sim \mathcal{D}}[\exp(\gamma \ell(\mathbf{h}, z))]) \right|$$

$$= \sup_{\gamma \in [\gamma_1, \gamma_2]} \frac{1}{\gamma^2} \left| \log(\mathbb{E}_{\mathbf{h} \sim \mathcal{P}_{\boldsymbol{\lambda}_2}} \mathbb{E}_{z \sim \mathcal{D}}[\exp(\gamma \ell(\mathbf{h}, z))] \frac{p_{\boldsymbol{\lambda}_1}(\mathbf{h})}{p_{\boldsymbol{\lambda}_2}(\mathbf{h})}) - \log(\mathbb{E}_{\mathbf{h} \sim \mathcal{P}_{\boldsymbol{\lambda}_2}} \mathbb{E}_{z \sim \mathcal{D}}[\exp(\gamma \ell(\mathbf{h}, z))]) \right|$$

$$\leq \sup_{\gamma \in [\gamma_1, \gamma_2]} \frac{1}{\gamma^2} \sup_{\mathbf{h} \in \mathcal{H}} \left| \log \frac{p_{\boldsymbol{\lambda}_1}(\mathbf{h})}{p_{\boldsymbol{\lambda}_2}(\mathbf{h})} \right|$$

$$\leq \frac{1}{\gamma_1^2} \sup_{\mathbf{h} \in \mathcal{H}} \left| \log \frac{p_{\boldsymbol{\lambda}_1}(\mathbf{h})}{p_{\boldsymbol{\lambda}_2}(\mathbf{h})} \right|$$

$$\leq \frac{1}{\gamma_1^2} \left( 2dM^2 e^{2a} + \frac{d(a+b)}{2} \right) \|\boldsymbol{\lambda}_1 - \boldsymbol{\lambda}_2\|,$$

where the first inequality used the property of the supremum, the $p_{\boldsymbol{\lambda}_1}(\mathbf{h}), p_{\boldsymbol{\lambda}_2}(\mathbf{h})$ in the fourth line denote the probability density function of Gaussian with mean $\mathbf{h}_0$ and variance parametrized by $\boldsymbol{\lambda}_1$, $\boldsymbol{\lambda}_2$ (i.e., $\boldsymbol{\lambda}_{1,i}, \boldsymbol{\lambda}_{2,i}$ are the variances for the $i$th layer), the second inequality use the fact that if $X(\mathbf{h})$ is a non-negative function of $\mathbf{h}$ and $Y(\mathbf{h})$ is a bounded function of $\mathbf{h}$, then

$$|\mathbb{E}_{\mathbf{h}}(X(\mathbf{h})Y(\mathbf{h}))| \leq (\sup_{\mathbf{h} \in \mathcal{H}} |Y(\mathbf{h})|) \cdot \mathbb{E}_{\mathbf{h}} X(\mathbf{h}).$$

The last inequality used the formula of the Gaussian density

$$p(x; \mu, \Sigma) = \frac{1}{(2\pi)^{d/2} |\Sigma|^{1/2}} \exp\left( -\frac{1}{2}(x - \mu)^T \Sigma^{-1}(x - \mu) \right)$$

and the boundedness of the parameters. Therefore, Assumption 4.1.2 is satisfied by setting $\eta_2(x) = L_2(d, M, \gamma_1, a)x$, where $L_2(d, M, \gamma_1, a) = \frac{1}{\gamma_1^2} \left( 2dM^2 e^{2a} + \frac{d(a+b)}{2} \right)$.

Let $L(d, M, T, \gamma_1, a) = L_1(d, M, T, a) + L_2(d, M, \gamma_1, a)$. Then we can apply Theorem 4.2, to get with probability $1 - \epsilon$,

$$\mathbb{E}_{\mathbf{h} \sim \mathcal{Q}_{\hat{\sigma}}(\hat{\mathbf{h}})} \ell(\mathbf{h}; \mathcal{D})$$
$$\leq \mathbb{E}_{\mathbf{h} \sim \mathcal{Q}_{\hat{\sigma}}(\hat{\mathbf{h}})} \ell(\mathbf{h}; \mathcal{S}) + \frac{1}{\hat{\gamma}m} \left[ \ln \frac{n(\varepsilon)}{\epsilon} + \mathrm{KL}(\mathcal{Q}_{\hat{\sigma}}(\hat{\mathbf{h}}) \| \mathcal{P}_{\boldsymbol{\lambda}}) \right] + \hat{\gamma} K_{\min}(\hat{\boldsymbol{\lambda}}) + CL(d, M, T, \gamma_1, a))\varepsilon.$$
(24)

Here, we used $\eta_1(x) = L_1 x$ and $\eta_2(x) = L_2 x$. Note that for the set $[-b, a]^k$, the covering number $n(\varepsilon) = \mathcal{N}([-b, a]^k, |\cdot|, \varepsilon)$ is $\left( \frac{b+a}{2\varepsilon} \right)^k$. Introducing a new variable $\rho > 0$, letting $\varepsilon = \frac{\rho}{CL(d, M, T, \gamma_1, a))}$ and inserting them to the above, we obtain with probability $1 - \epsilon$:

$$\mathbb{E}_{\mathbf{h} \sim \mathcal{Q}_{\hat{\sigma}}(\hat{\mathbf{h}})} \ell(\mathbf{h}; \mathcal{D})$$
$$\leq \mathbb{E}_{\mathbf{h} \sim \mathcal{Q}_{\hat{\sigma}}(\hat{\mathbf{h}})} \ell(\mathbf{h}; \mathcal{S}) + \frac{1}{\hat{\gamma}m} \left[ \ln \frac{1}{\epsilon} + \mathrm{KL}(\mathcal{Q}_{\hat{\sigma}}(\hat{\mathbf{h}}) \| \mathcal{P}_{\boldsymbol{\lambda}}) \right]$$
$$+ \hat{\gamma} K_{\min}(\hat{\boldsymbol{\lambda}}) + \rho + \frac{k}{\gamma_1 m} \ln \frac{C \cdot L(d, M, T, \gamma_1, a)(b+a)}{2\rho}.$$

Further making the upper bound tighter by optimizing over $\rho$, we obtain

$$\mathbb{E}_{\mathbf{h} \sim \mathcal{Q}_{\hat{\boldsymbol{\sigma}}}(\hat{\mathbf{h}})} \ell(\mathbf{h}; \mathcal{D})$$

$$\leq \mathbb{E}_{\mathbf{h} \sim \mathcal{Q}_{\hat{\boldsymbol{\sigma}}}(\hat{\mathbf{h}})} \ell(\mathbf{h}; \mathcal{S}) + \frac{1}{\hat{\gamma} m} \left[ \ln \frac{1}{\epsilon} + \mathrm{KL}(\mathcal{Q}_{\hat{\boldsymbol{\sigma}}}(\hat{\mathbf{h}}) \| \mathcal{P}_{\boldsymbol{\lambda}}) \right]$$

$$+ \hat{\gamma} K_{\min}(\hat{\boldsymbol{\lambda}}) + \frac{k}{\gamma_1 m} \left( 1 + \ln \frac{C \cdot L(d, M, T, \gamma_1, a)(b + a)\gamma_1 m}{2k} \right)$$

$$= L_{PAC}(\hat{\mathbf{h}}, \hat{\gamma}, \hat{\boldsymbol{\sigma}}, \hat{\boldsymbol{\lambda}}) + \frac{k}{\gamma_1 m} \left( 1 + \ln \frac{C \cdot L(d, M, T, \gamma_1, a)(b + a)\gamma_1 m}{2k} \right).$$

$\square$

## APPENDIX C    EXTENDED EXPERIMENTAL DETAILS

We conducted experiments using eight A5000 GPUs that were powered by four AMD EPYC 7543 32-Core Processors. To speed up the training process for posterior and prior variance, we utilized a warmup method that involved updating the noise level in the posterior of each layer as a scalar for the first 50 epochs and then proceeding with normal updates after the warmup period. This method only affects the convergence speed, not the generalization, and it was only used for large models in image classification.

### C.1    INITIALIZATION

Recall that the exponential momentum bound $K(\boldsymbol{\lambda})$ is estimated over a range $[\gamma_1, \gamma_2]$ of $\gamma$ as per Definition 2. It means that we need the inequality

$$\mathbb{E}_{\mathbf{h} \sim \mathcal{P}_{\boldsymbol{\lambda}}} \mathbb{E}[\exp(\gamma X(\mathbf{h}))] \leq \exp(\gamma^2 K(\boldsymbol{\lambda}))$$

to hold for any $\gamma$ in this range. One needs to be a little cautious when choosing the upper bound $\gamma_2$, because if it is too large, then the empirical version of $\mathbb{E}_{\mathbf{h} \sim \mathcal{P}} \mathbb{E}[\exp(\gamma X(\mathbf{h}))]$ would be a very poor approximation to the true expectation due to the fact that the variable $X$ is on the exponent, unless a large amount of samples $h$ is drawn, which would then be extremely time-consuming and unrealistic in the deep neural network setting. Therefore, we recommended $\gamma_2$ to be set to 10, or 20 at most, to avoid this issue. Even though in some cases, this means the optimal $\gamma$ that minimizes the PAC-Bayes bound is ruled out, we found that a more accurate estimation of $K$ resulting from using the recommended $\gamma_2$ is more crucial to obtaining the best performance of the PAC-Bayes training. The choice of $\gamma_1$ is not as critical as the choice of $\gamma_2$. But a smaller $\gamma_1$ usually means a larger value of $K(\boldsymbol{\lambda})$ for any given $\boldsymbol{\lambda}$, and therefore a looser PAC-Bayes bound and worse performance.

Typically, the training dataset for a graph neural network is quite small. As a result, the KL divergence term in the PAC-Bayes loss gets large, easily provided the initialization of the noise is not sufficiently good. This poses a challenge in initializing the PAC-Bayes training process. Although the proposed initialization in Algorithm 1 works well for most GNN networks and datasets, it may fail on some occasions. To address this issue, we modify the initialization by adding a clipping of the noise levels $\boldsymbol{\sigma}$ and $\lambda$ at a lower bound of $-\log(10)$. For GNN, this operation increases the noise level. Please refer to Remark C.0.1 for the theoretical reason of the clipping and to Appendix A.5 (second item) for how the value $-\log 10$ is found in practice.

**Remark C.0.1.** *For large datasets (like in MNIST or CIFAR10), we can set $M, T, a, b$ to be relatively large little increase of the bound, then during training the parameters would not exceed these bounds. Hence, no effort needs to be made to ensure the boundedness assumption in Corollary 5.0.1 holds. But When the dataset size $m$ is small compared to $\log d$ or $k$, we should choose smaller $M, T, a, b$ to make the PAC-Bayes bound in Corollary 5.0.1 small. Consequently, in the training algorithm, we need to ensure the parameters stay in the sets $\mathcal{H}, \Sigma,$ and $\Gamma$ by projecting onto these sets after each gradient update.*

### C.2    IMAGE CLASSIFICATION

There is no data augmentation in our experiments. The implementation is based on the GitHub repo Liu (2021). For the layerwise prior, we treated each parameter in the PyTorch object

model.parameters() as an independent layer, i.e., the weights and bias of one convolution/batch-norm layer were treated as two different layers. The number of training epochs of Stage 1 is 500 epochs for PAC-Bayes training. Moreover, a learning rate scheduler was added to both our method and the baseline to make the training fully converge. Specifically, the learning rate will be reduced by $0.1$ whenever the training accuracy does not increase for 20 epochs. For PAC-Bayes training, the scheduler is only activated in Stage 2. The training will be terminated when the training accuracy is above $99.9\%$ for 20 epochs or when the learning rate decreases to below $1e{-}5$.

We also add label smoothing $(0.1)$ (Szegedy et al., 2016) to all models because minimizing the loss and maximizing the accuracy are two separate though related tasks. In the image or node classification task, the goal is to increase the classification accuracy, while in PAC-Bayes training, the goal is to minimize the population loss. However, in cases when accuracy and loss are perfectly negatively correlated, these two goals coincide. With label-smoothing, we observed a larger correlation between the loss and accuracy than without, and therefore we reported results with label-smoothing turned on for both the PAC-Bayes training and baselines in image classification tasks. When the label smoothing is turned off, the performances of the two methods will decrease by comparable amounts.

The detailed searched values of hyper-parameters include momentum for SGD $(0.3, 0.6, 0.9)$, learning rates $(1e{-}3, 5e{-}3, 1e{-}2, 5e{-}2, 1e{-}1, 2e{-}1)$, weight decay $(1e{-}4, 5e{-}4, 1e{-}3, 5e{-}3, 1e{-}2)$, and noise injection $(5e{-}4, 1e{-}3, 5e{-}3, 1e{-}2)$. The best learning rate for Adam and AdamW is the same since weight decay is the only difference between the two optimizers. We adjusted one hyper-parameter at a time while keeping the others fixed to accelerate the search. To determine the optimal hyper-parameter for a variable, we compared the mean testing accuracy of the last five epochs. We then used this selected hyper-parameter to tune the next one.

The testing accuracy from all experiments with batch size $128$ with the learning rate $1e{-}4$ is shown in Figure 2 and Figure 3. We also visualize the sorted testing accuracy of baselines and our proposed PAC-Bayes training with large batch sizes and a larger learning rate $5e{-}4$ (used only to obtain faster convergence) in Figure 4 and Figure 5. These figures demonstrate that our PAC-Bayes training algorithm achieves better testing accuracy than most searched settings. For models VGG13 and ResNet18, the large batch size is $2048$, and for large models VGG19 and ResNet34, the large batch size is set to $1280$ due to the GPU memory limitation.

To best demonstrate the sensitivity of the hyper-parameter selection of baselines and motivate our PAC-Bayes training, we organized the test accuracy below for ResNet18. Considering the search efficiency, we searched the hyper-parameter one by one. For SGD, we first searched the learning rate, set the momentum and the weight decay as 0 (both are default values for SGD), and then used the best learning rate to search for the momentum. At last, the best-searched learning rate and momentum are used to search for weight decay. For Adam, we searched the learning rate, weight decay, and noise injection in an order similar to SGD. Since AdamW and Adam are the same when setting the weight decay as 0, we searched for the best weight decay based on the best learning rate obtained from searching on Adam.

The tables 4,5,6,7 show that the standard deviation when searching different hyper-parameters can be significant. We can also see the increase of the maximum test accuracy when adding a new regularization (both learning rate and momentum are implicit regularizations Smith et al.; Ghosh et al. (2022)), showing that all hyper-parameters are necessary to search for better generalization. Compared with Adam/AdamW, SGD is more sensitive to different hyper-parameters, but we need SGD to achieve the best test accuracy when using the small batch size (128). The sensitivity of the search shows the advantages of our proposed PAC-Bayes training. Note that for CIFAR10 and CIFAR100, we use the testing dataset to search hyper-parameters of baselines. PAC-Bayes training can match the test accuracy directly without any information on the test data.

We also compared PAC-Bayes training with SGD/Adam/AdamW on CIFAR10, using a batch size of 128, and allocated 10% of the training data for training and the remaining 90% for hyper-parameter searching in SGD/Adam/AdamW. With ResNet18, the test accuracy of PAC-Bayes is 67.8%, while the best test accuracies for SGD, Adam, and AdamW, after hyper-parameter searching, are 64.00%, 64.96%, and 65.59%, respectively. When training ResNet18 with all the training data using a batch size of 128, SGD typically achieves the best test accuracy among the baselines. However, AdamW

Table 4: Test accuracy for ResNet18 on CIFAR10 with the batch size 128 when tuning different hyper-parameters.

| Optimizer | Parameter | Max | Min | Mean | Std |
|---|---|---|---|---|---|
| SGD | learning rate (1e-3 to 0.2) | 84.8 | 59.8 | 75.3 | 10.2 |
| SGD | momentum (0.3 to 0.9) | 85.6 | 84.1 | 85.0 | 0.8 |
| SGD | weight decay (1e-4 to 1e-2) | 89.9 | 85.9 | 88.2 | 1.6 |
| Adam | learning rate (1e-3 to 0.2) | 87.5 | 82.5 | 84.5 | 1.7 |
| Adam | weight decay (1e-4 to 1e-2) | 86.7 | 82.8 | 85.5 | 1.6 |
| Adam | noise injection (5e-4 to 1e-2) | 87.3 | 85.6 | 86.5 | 0.9 |
| AdamW | weight decay (1e-4 to 1e-2) | 87.9 | 86.6 | 87.4 | 0.5 |
| AdamW | noise injection (5e-4 to 1e-2) | 87.2 | 86.0 | 86.8 | 0.6 |

Table 5: Test accuracy for ResNet18 on CIFAR10 with the batch size 2048 when tuning different hyper-parameters.

| Optimizer | Parameter | Max | Min | Mean | Std |
|---|---|---|---|---|---|
| SGD | learning rate (1e-3 to 0.2) | 72.0 | 56.5 | 62.8 | 6.2 |
| SGD | momentum (0.3 to 0.9) | 75.0 | 72.3 | 74.0 | 1.4 |
| SGD | weight decay (1e-4 to 1e-2) | 85.4 | 72.0 | 77.8 | 6.2 |
| Adam | learning rate (1e-3 to 0.2) | 85.1 | 70.0 | 78.4 | 6.1 |
| Adam | weight decay (1e-4 to 1e-2) | 86.4 | 77.4 | 83.7 | 3.8 |
| Adam | noise level (5e-4 to 1e-2) | 87.7 | 85.1 | 86.4 | 1.2 |
| AdamW | weight decay (1e-4 to 1e-2) | 84.8 | 84.5 | 84.7 | 0.1 |
| AdamW | noise level (5e-4 to 1e-2) | 84.9 | 82.7 | 83.9 | 0.9 |

outperforms SGD when using only 10% of the training data. This demonstrates the necessity of both hyper-parameter searching and choosing the appropriate optimizer.

### C.3 ABLATION STUDY ON IMAGE CLASSIFICATION

We conducted an ablation study to showcase some extra benefits of the proposed PAC-Bayes training algorithm besides its ability to achieve auto-tuning. Specifically, we tested the effect of different learning rates on ResNet18 and VGG13 models trained with layerwise prior. Learning rate has long been known as an important impact factor of the generalization for baseline training. Within the stability range of gradient descent, the larger the learning rate is, the better the generalization has been observed (Lewkowycz et al., 2020). In contrast, the generalization of the PAC-Bayes trained model is less sensitive to the learning rate. We do observe that due to the newly introduced noise parameters, the stability of the optimization gets worse, which in turn requires a smaller learning rate to achieve stable training. But as long as the stability is guaranteed by setting the learning rate small enough, our results, as Table 8 and 9, indicated that the testing accuracy remained stable across various learning rates for VGG13 and Resnet18. The dash in the table means that the learning rate for that particular setting might be too large to main the stability of the algorithm. For learning rates below $1e-4$, we trained the model in Stage 1 for more epochs (700) to fully update the prior and posterior variance.

We also demonstrate that the warmup iterations (as discussed at the beginning of this section) do not affect generalization. As shown in Table 10, the testing accuracy is insensitive to different numbers of warmup iterations.

### C.4 COMPATIBILITY WITH DATA AUGMENTATION

We didn't include data augmentation in most experiments for rigorousness considerations. Because with data augmentation, there is no rigorous way of choosing the sample size $m$ that appears in the

Table 6: Test accuracy for ResNet18 on CIFAR100 with the batch size 128 when tuning different hyper-parameters.

| Optimizer | Parameter | Max | Min | Mean | Std |
|---|---|---|---|---|---|
| SGD | learning rate (1e-3 to 0.2) | 58.7 | 33.7 | 48.5 | 10.3 |
| SGD | momentum (0.3 to 0.9) | 59.0 | 51.8 | 56.6 | 4.1 |
| SGD | weight decay (1e-4 to 1e-2) | 64.0 | 56.6 | 61.8 | 3.0 |
| Adam | learning rate (1e-3 to 0.2) | 61.5 | 27.1 | 44.6 | 14.4 |
| Adam | momentum (0.3 to 0.9) | 58.4 | 43.1 | 53.0 | 6.3 |
| Adam | weight decay (1e-4 to 1e-2) | 61.6 | 61.0 | 61.2 | 0.3 |
| AdamW | weight decay (1e-4 to 1e-2) | 61.1 | 57.3 | 59.3 | 1.8 |
| AdamW | noise level (5e-4 to 1e-2) | 61.4 | 58.8 | 59.8 | 1.1 |

Table 7: Test accuracy for ResNet18 on CIFAR100 with batch size 2048 when tuning different hyper-parameters.

| Optimizer | Parameter | Max | Min | Mean | Std |
|---|---|---|---|---|---|
| SGD | learning rate (1e-3 to 0.2) | 44.2 | 27.8 | 35.8 | 6.8 |
| SGD | momentum (0.3 to 0.9) | 52.2 | 45.9 | 49.2 | 3.1 |
| SGD | weight decay (1e-4 to 1e-2) | 61.5 | 54.5 | 57.3 | 3.2 |
| Adam | learning rate (1e-3 to 0.2) | 56.3 | 31.0 | 42.9 | 11.8 |
| Adam | weight decay (1e-4 to 1e-2) | 59.4 | 56.0 | 57.6 | 1.4 |
| Adam | noise level (5e-4 to 1e-2) | 65.4 | 59.0 | 62.0 | 3.3 |
| AdamW | weight decay (1e-4 to 1e-2) | 56.5 | 55.9 | 56.1 | 0.2 |
| AdamW | noise level (5e-4 to 1e-2) | 58.9 | 57.1 | 58.3 | 0.8 |

Table 8: Testing accuracy of ResNet18 trained with different learning rates.

| lr | $3e-5$ | $5e-5$ | $1e-4$ | $2e-4$ | $3e-4$ | $5e-4$ |
|---|---|---|---|---|---|---|
| CIFAR10 | 88.4 | 88.8 | 89.3 | 88.6 | 88.3 | 89.2 |
| CIFAR100 | 69.2 | 69.0 | 68.9 | 69.1 | 69.1 | 69.6 |

Table 9: Testing accuracy of VGG13 trained with different learning rates.

| lr | $3e-5$ | $5e-5$ | $1e-4$ | $2e-4$ | $3e-4$ | $5e-4$ |
|---|---|---|---|---|---|---|
| CIFAR10 | 88.6 | 88.9 | 89.7 | 89.6 | 89.6 | 89.5 |
| CIFAR100 | 67.7 | 68.0 | 67.1 | - | - | - |

Table 10: Testing accuracy of ResNet18 trained with warmup epochs of $\sigma$.

| | 10 | 20 | 50 | 80 | 100 | 150 |
|---|---|---|---|---|---|---|
| CIFAR10 | 88.5 | 88.5 | 89.3 | 89.5 | 89.5 | 88.9 |
| CIFAR100 | 69.4 | 69.6 | 68.9 | 69.1 | 69.0 | 68.1 |

PAC-Bayes bound. More specifically, for the PAC-Bayes bound to be valid, the training data has to be i.i.d. samples from some underlying distribution. However, most data augmentation techniques would break the i.i.d. assumption. As a result, if we have 10 times more samples after augmentation, the new information they bring in would be much less than those from 10 times i.i.d. samples. In this case, how to determine the effective sample size $m$ to be used in the PAC-Bayes bound is a problem.

Since knowing whether a training method can work well with data augmentation is important, we carried out the PAC-Bayes training with an ad-hoc choice of $m$, that is, we set $m$ to be the size of the augmented data. We compared the grid-search result of SGD and Adam versus PAC-Bayes training on CIFAR10 with ResNet18. The augmentation is achieved by random flipping and random cropping. The data augmentation increased the size of the training sample by 128 times. The grid search is the same as in Appendix C.2. The testing accuracy for SGD is 95.2%, it is 94.3% for Adam, it is 94.4% for AdamW, and it is 94.3% for PAC-Bayes training with the layerwise prior. In contrast, the testing accuracy without data augmentation is lower than 90% for all methods. It suggests that data augmentation does not conflict with the PAC-Bayes training in practice.

## C.5   MODEL ANALYSIS

We examined the learning process of PAC-Bayes training by analyzing the posterior variance $\sigma$ for different layers in models trained by Algorithm 3. Typically, batch norm layers have smaller $\sigma$ values than convolution layers. Additionally, shadow convolution and the last few layers have smaller $\sigma$ values than the middle layers. We also found that skip-connections in ResNet18 have smaller $\sigma$ values than nearby layers, suggesting that important layers with a greater impact on the output have smaller $\sigma$ values.

In Stage 1, the training loss is higher than the testing loss, which means the adopted PAC-Bayes bound is able to bound the generalization error throughout the PAC-Bayes training stage. Additionally, we observed that the final value of $K$ is usually very close to the minimum of the sampled function values. The average value of $\sigma$ experienced a rapid update during the initial 50 warmup epochs but later progressed slowly until Stage 2. The details can be found in Figure 11-15. Based on the figures, shadow convolution and the last few layers have smaller $\sigma$ values than the middle layers for all models. We also found that skip-connections in ResNet18 and ResNet34 have smaller $\sigma$ values than nearby layers on both datasets, suggesting that important layers with a greater impact on the output have smaller $\sigma$ values.

## C.6   TRAINING COMPLEXITY

When running ResNet18 on CIFAR10 with a GTX 3090, it takes around 16 mins to pre-compute K. The reported average searching cost is over all searched settings in Section 6, including various batch sizes, learning rates, weight decay coefficients, and noise injection levels (only for Adam/AdamW). For PAC-Bayes training, we compute the time for batch size 2048, since the test result is insensitive to the batch size. As shown in Table 11-13, our method requires a much shorter total running time than a grid search of the baseline, where "scalar" and "layer" denote the PAC-Bayes training with scalar and layerwise prior. While a single run of the baseline is currently much faster than ours, the average number of backpropagation is comparable. Since backpropagation is the computation bottleneck in each iteration, having a similar number of backpropagation suggests our algorithm has the potential to achieve a similar running time to a single run of the baseline with some code optimization. More explicitly, for SGD/Adam/AdamW, the total number of backpropagation is calculated as

$$\frac{\text{size of the training dataset}}{\text{batch size}} \times \text{number of epochs.} \tag{25}$$

Table 11: Total running time of searching (hours).

|        | C10   | C100  |
|--------|-------|-------|
| SGD    | 4.27  | 7.48  |
| Adam   | 13.91 | 12.52 |
| AdamW  | 9.50  | 3.73  |
| scalar | 2.03  | 1.94  |
| layer  | 2.22  | 2.11  |

Table 12: Average time for one search (seconds).

|        | C10  | C100 |
|--------|------|------|
| SGD    | 549  | 962  |
| Adam   | 1670 | 1503 |
| AdamW  | 1899 | 746  |
| scalar | 7293 | 6974 |
| layer  | 7980 | 7584 |

Table 13: Average backpropagation based on equation 25.

|        | C10   | C100  |
|--------|-------|-------|
| SGD    | 11501 | 17095 |
| Adam   | 28508 | 28335 |
| AdamW  | 24761 | 9560  |
| scalar | 15912 | 15216 |
| layer  | 15960 | 15168 |

When we search the hyper-parameters, the total number of backpropagations for SGD/Adam is computed as the average over all runs. For PAC-Bayes training, each iteration also only needs 1 backpropagation because we can backpropagate once and use the result to compute the gradient of both the model and the noise. So the above formula is also used to compute the total number of backpropagations for PAC-Bayes training. Although PAC-Bayes training requires more epochs to converge, each epoch is faster since it allows the use of large batch sizes. That is why the total number of iterations/backpropagations is similar to a single run of the baseline. However, the wall time of our current PAC-Bayes code in the submitted package is much larger than this estimate because we wrote the for-loops in Python instead of C/C++ for the noise injection part, and Pytorch is not optimized for the PAC-Bayes training. We expect a code optimization would bring down the wall time of PAC-Bayes training to be close to that of baselines. Moreover, further acceleration of our algorithm could be achieved by improving the optimization algorithm, e.g., optimizing the variance and weights alternatively. The memory usage of PAC-Bayes training is roughly twice of normal training, as discussed in the appendix. More concretely, when using a batch size of 128, estimating K requires 4384MB of GPU memory, and PAC-Bayes training with layerwise prior utilizes 4424MB of GPU memory. In contrast, memory usage is 2876MB when using SGD, 2950MB when using Adam without noise injection, and 3190MB with noise injection with Adam.

## C.7 Node classification by GNNs

We test the PAC-Bayes training algorithm on the following popular GNN models, tuning the learning rate $(1e{-}3, 5e{-}3, 1e{-}2)$, weight decay $(0, 1e{-}2, 1e{-}3, 1e{-}4)$, noise injection $(0, 1e{-}3, 1e{-}2, 1e{-}3)$, and dropout $(0, 0.4, 0.8)$.

- GCN (Kipf & Welling, 2016): the number of filters is 32.
- SAGE (Hamilton et al., 2017): the number of filters is 32.
- GAT (Veličković et al., 2017): the number of filters is 8, the number of heads is 8, and the dropout rate of the attention coefficient is 0.6.
- APPNP (Gasteiger et al., 2018): the number of filters is 32, $K = 10$ and $\alpha = 0.1$.

We set the number of layers to 2, which achieves the best performance for the baseline. A ReLU activation and a dropout layer are added in between the two convolution layers. Since GNNs are faster to train than convolutional neural networks, we tested all possible combinations of the above parameters for the baseline, conducting 144 searches per model on one dataset. We use Adam as the optimizer with the learning rate as $1e{-}2$ for all models using both training and validation nodes for PAC-Bayes training.

We also did a separate experiment by disabling the early stopping and using the training and validation nodes both for training. For baselines, we need first to train the model to detect the best hyper-parameters as before and then train the model again on the combined data. Our PAC-Bayes training can also match the best generalization of baselines in this setting.

All search details are visualized in Figure 6-9. The AdamW+val and scalar+val record the performances of the baseline and the PAC-Bayes training, respectively, with both training and validation datasets for training. We can see that testing accuracy after adding validation nodes increased significantly for both methods but still, the results of our algorithm match the best testing accuracy of baselines. Our proposed PAC-Bayes training with the scalar prior is better than most of the settings during searching and achieved comparable testing accuracy when adding validation nodes to training.

## C.8 Few-shot text classification with transformers

The proposed method is also observed to work on transformer networks. We conducted experiments on two text classification tasks of the GLUE benchmark as shown in Table 14. SST is the sentiment analysis task, whose performance is evaluated as the classification accuracy. Sentiment analysis is the process of analyzing the sentiment of a given text to determine if the emotional tone of the text is positive, negative, or neutral. QNLI (Question-answering Natural Language Inference) focuses on determining the logical relationship between a given question and a corresponding sentence. The

objective of QNLI is to determine whether the sentence contradicts, entails, or is neutral with respect to the question.

We use classification accuracy as the evaluation metric. The baseline method uses grid search over the hyper-parameter choices of the learning rate ($1e{-}1$, $1e{-}2$, $1e{-}3$), batch size $(2, 8, 16, 32, 80)$, dropout ratio $(0, 0.5)$, optimization algorithms (SGD, AdamW), noise injection $(0, 1e{-}5, 1e{-}4, 1e{-}3, 1e{-}2, 1e{-}1)$, and weight decay $(0, 1e{-}1, 1e{-}2, 1e{-}3, 1e{-}4)$. The learning rate and batch size of our method are set to $1e{-}3$ and $100$ (i.e., full-batch), respectively. In this task, the number of training samples is small (80). As a result, the preset $\gamma_2 = 10$ is a bit large and thus prevents the model from achieving the best performance with PAC-Bayes training. We use the refined procedure described in Appendix C.1[6].

We adopt BERT (Devlin et al., 2018) as our backbone and added one fully-connect layer to be the classification layer. Only the added classification layer is trainable, and the pre-trained model is frozen without gradient update. To simulate a few-shot learning scenario, we randomly sample 100 instances from the original training set and take the whole development set to evaluate the classification performance. We split the training set into 5 splits, taking one split as the validation data and the rest as the training set. Each experiment was conducted five times, and we report the average performance. We used the PAC-Bayes training with the scalar prior in this experiment. According to Table 14, our method is competitive to the baseline method on the SST task, the performance gap is only $0.4$ points. On the QNLI task, our method outperforms the baseline by a large margin, and the variance of our proposed method is less than that of the baseline method.

Table 14: Testing accuracy on the development sets of 2 GLUE benchmarks.

|  | SST | QNLI |
| --- | --- | --- |
| baseline | **72.9±0.99** | 62.6±0.10 |
| scalar | 72.5±0.99 | **64.2±0.02** |

---

[6]The refined procedure can be also applied to the CNN and GNN experiments but with smaller improvements than the transformers.

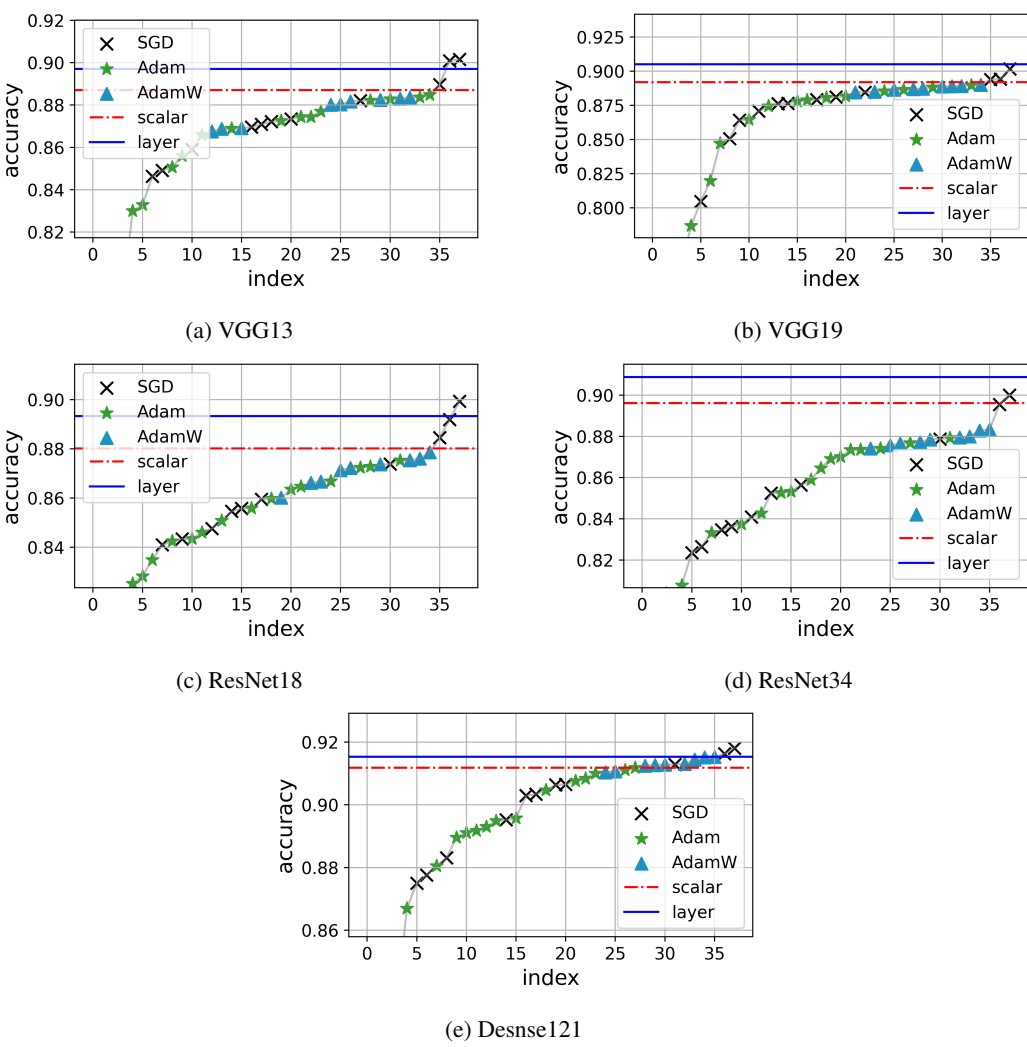

Figure 2: Sorted testing accuracy of CIFAR10. The x-axis represents the experiment index.

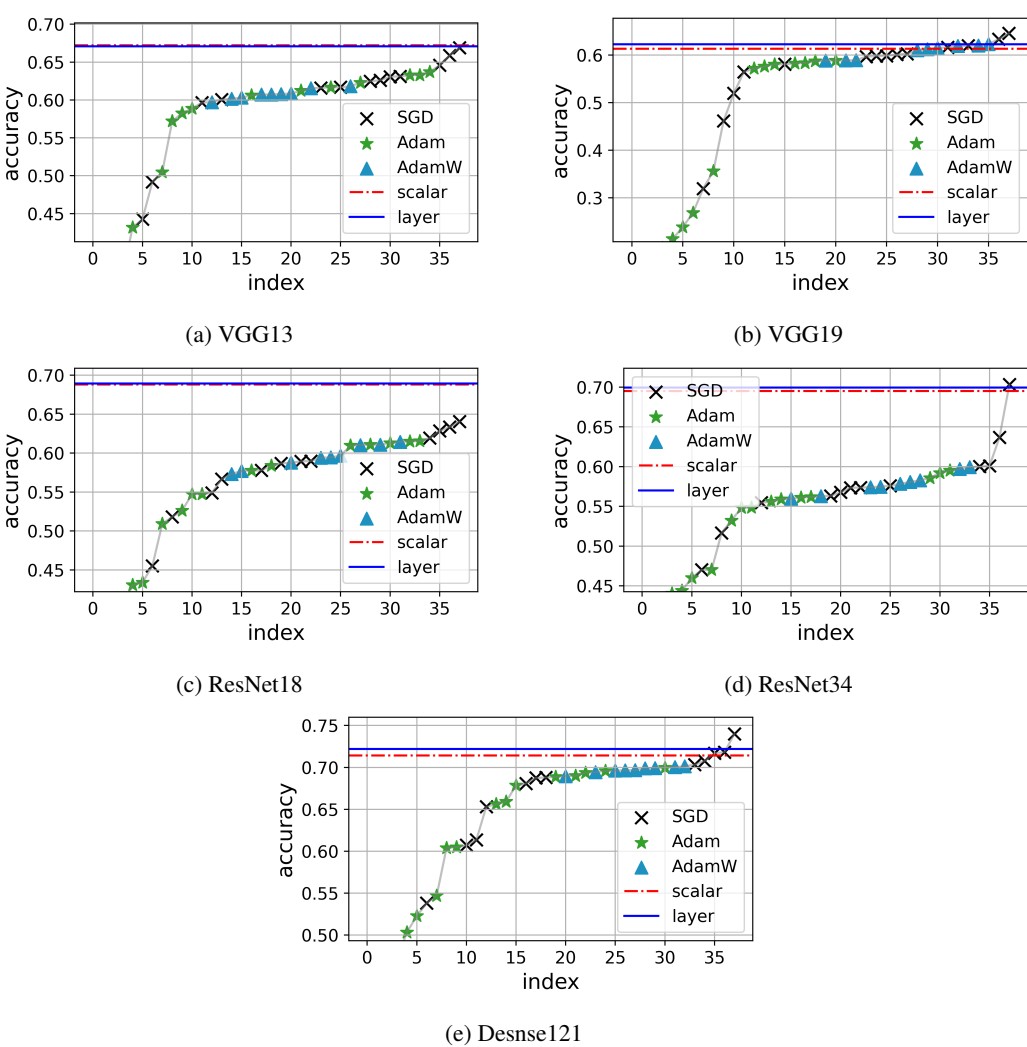

(a) VGG13

(b) VGG19

(c) ResNet18

(d) ResNet34

(e) Desnse121

Figure 3: Sorted testing accuracy of CIFAR100. The x-axis represents the experiment index.

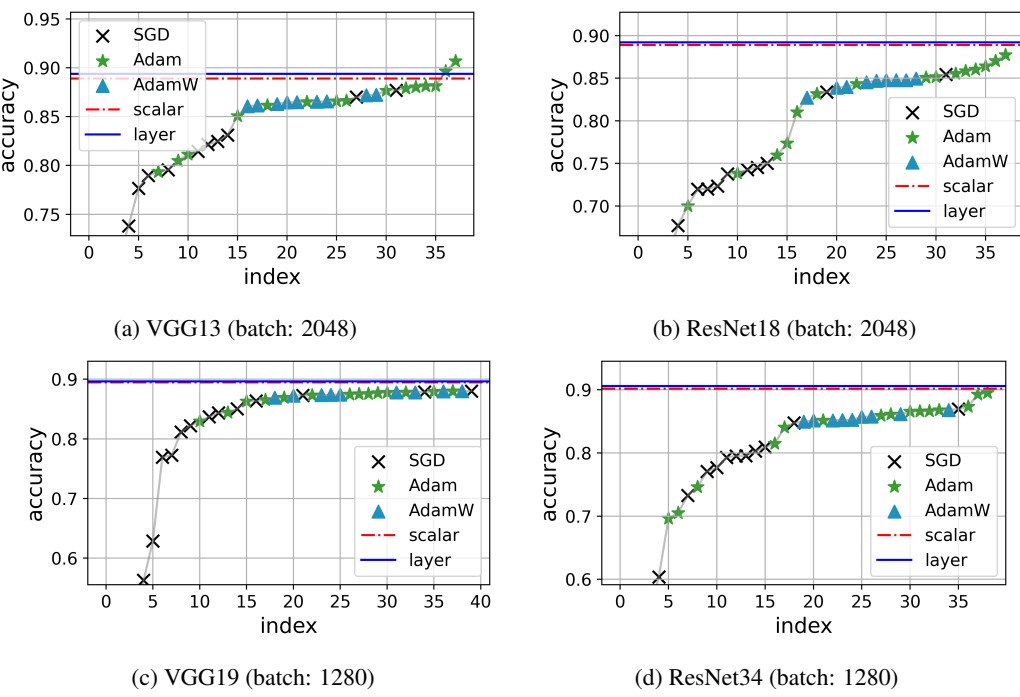

(a) VGG13 (batch: 2048)

(b) ResNet18 (batch: 2048)

(c) VGG19 (batch: 1280)

(d) ResNet34 (batch: 1280)

Figure 4: Sorted testing accuracy of CIFAR10 with large batch sizes. The x-axis represents the experiment index.

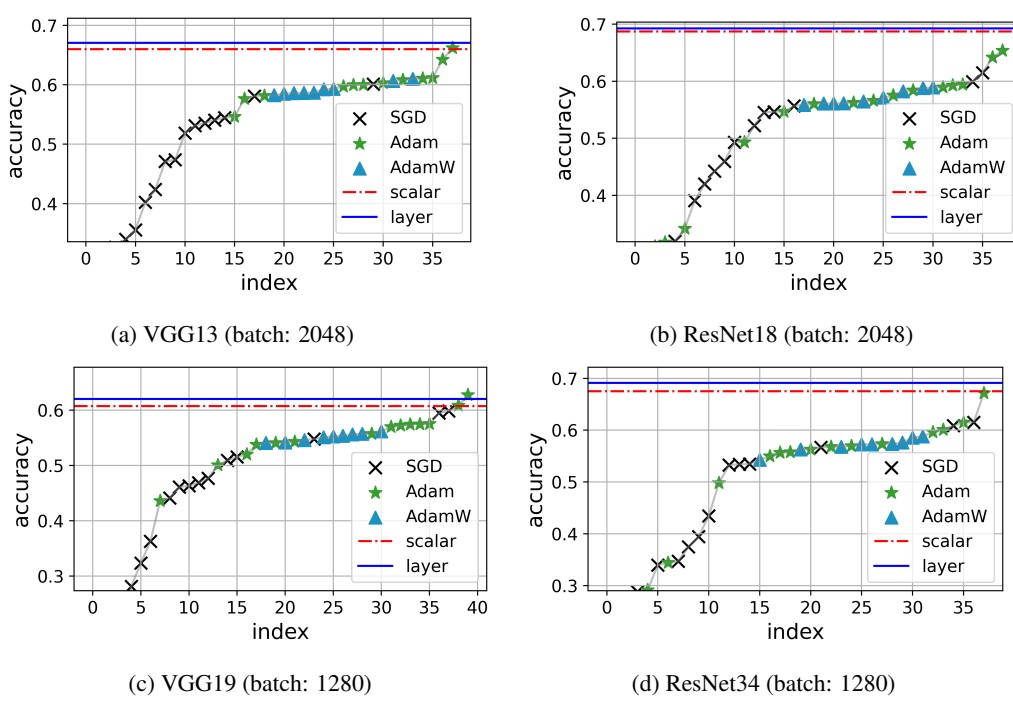

(a) VGG13 (batch: 2048)

(b) ResNet18 (batch: 2048)

(c) VGG19 (batch: 1280)

(d) ResNet34 (batch: 1280)

Figure 5: Sorted testing accuracy of CIFAR100 with large batch sizes. The x-axis represents the experiment index.

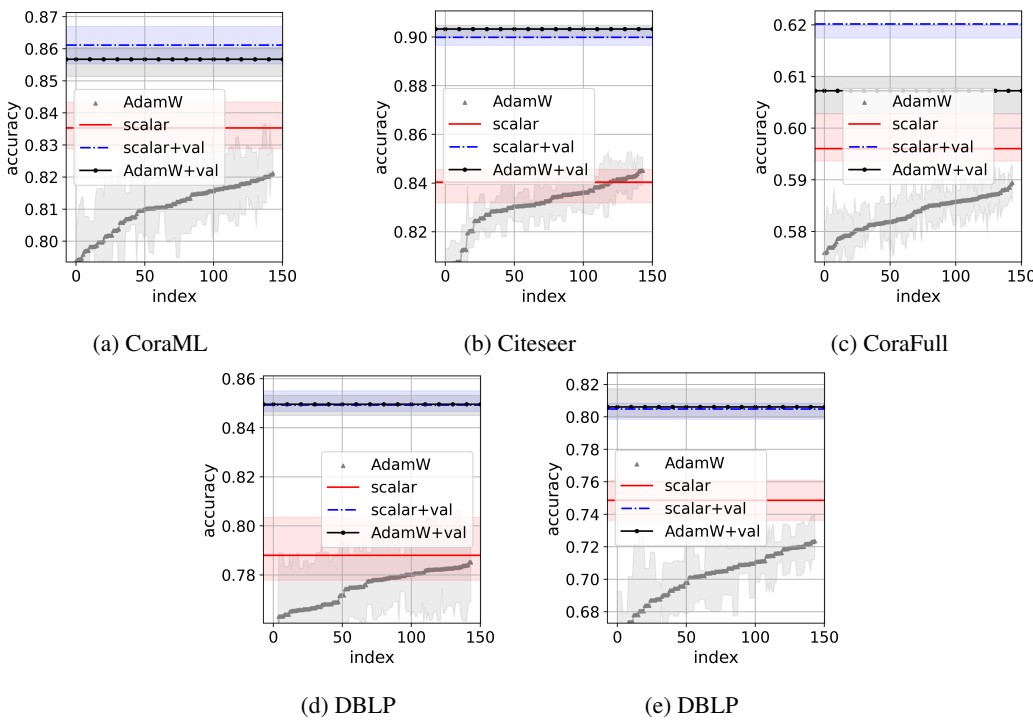

Figure 6: Testing accuracy of GCN. The interval is constructed by the first and third quartiles over the ten random splits. {+val} denotes the performance with both training and validation dataset for training.

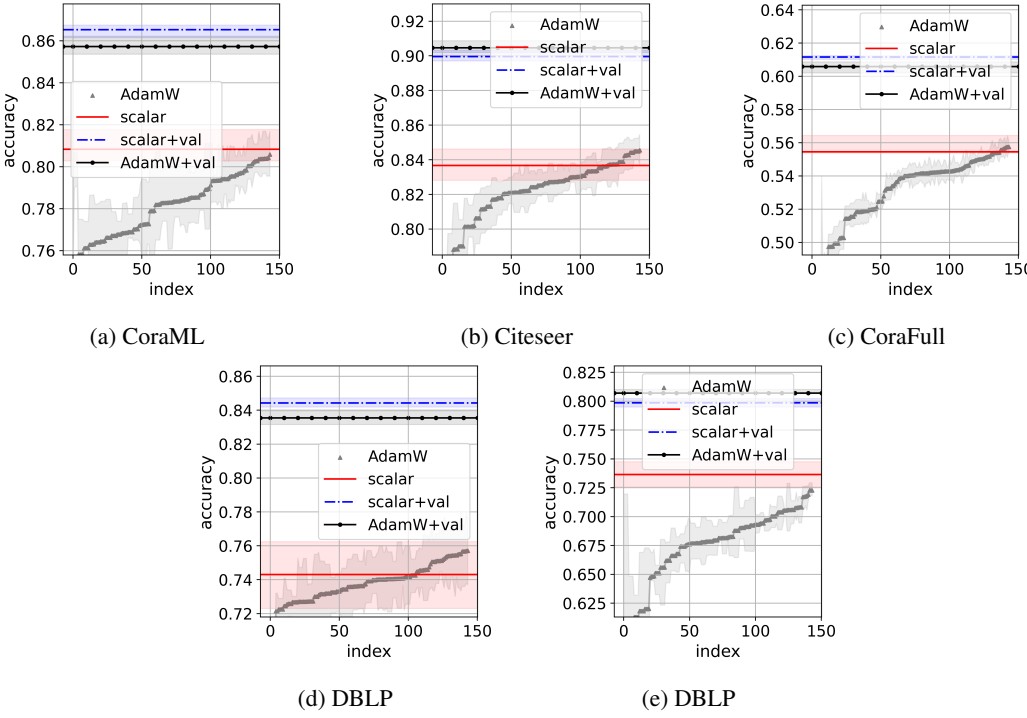

Figure 7: Testing accuracy of SAGE. The interval is constructed by the first and third quartiles over the ten random splits. {+val} denotes the performance with both training and validation dataset for training.

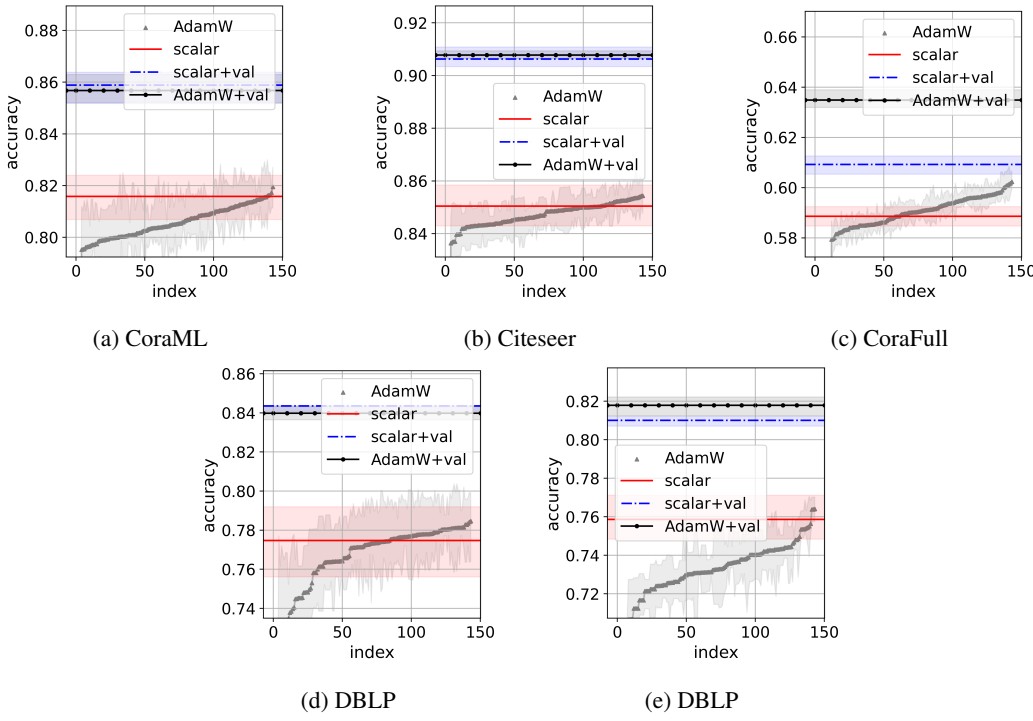

Figure 8: Testing accuracy of GAT. The interval is constructed by the first and third quartiles over the ten random splits. {+val} denotes the performance with both training and validation dataset for training.

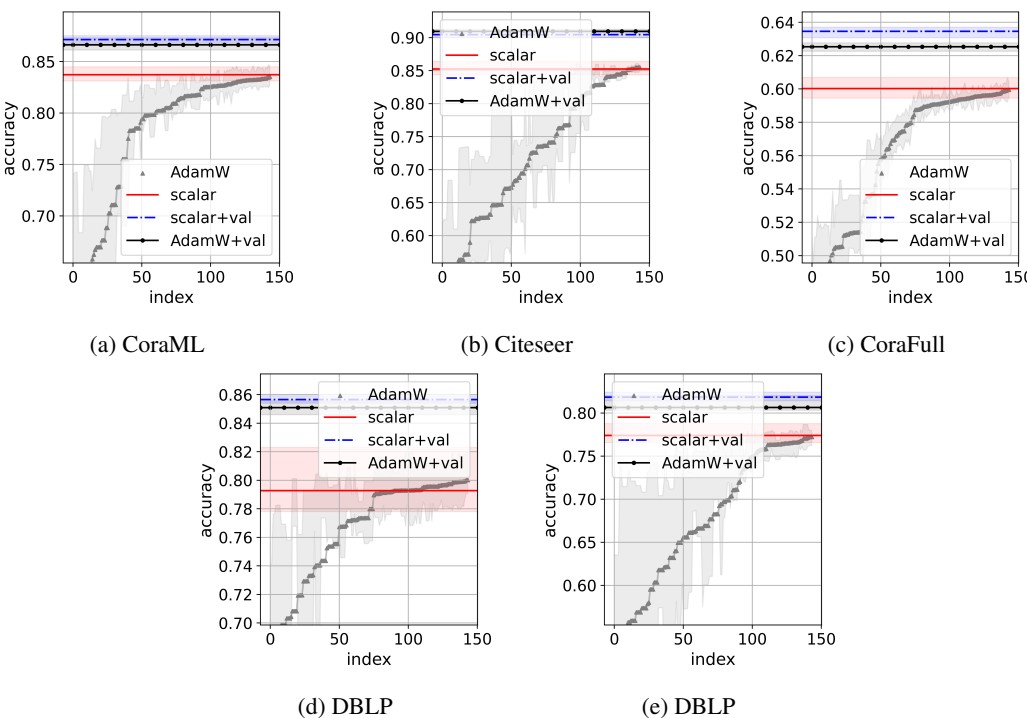

Figure 9: Testing accuracy of APPNP. The interval is constructed by the first and third quartiles over the ten random splits. {+val} denotes the performance with both training and validation dataset for training.

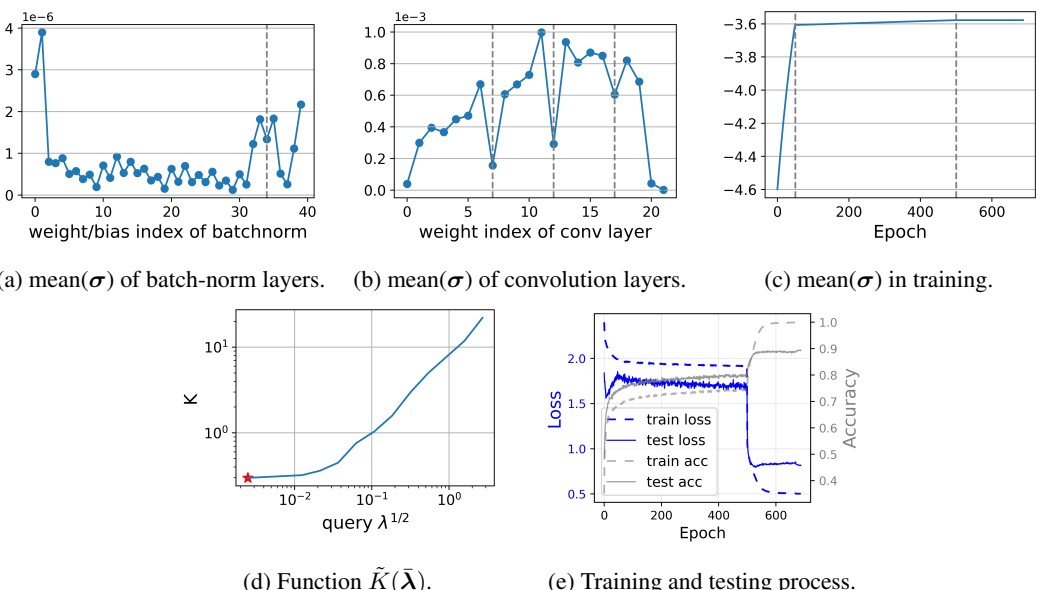

(a) mean($\boldsymbol{\sigma}$) of batch-norm layers.    (b) mean($\boldsymbol{\sigma}$) of convolution layers.    (c) mean($\boldsymbol{\sigma}$) in training.

(d) Function $\tilde{K}(\bar{\boldsymbol{\lambda}})$.    (e) Training and testing process.

Figure 10: Training details of ResNet18 on CIFAR10. The red star denotes the final $K$.

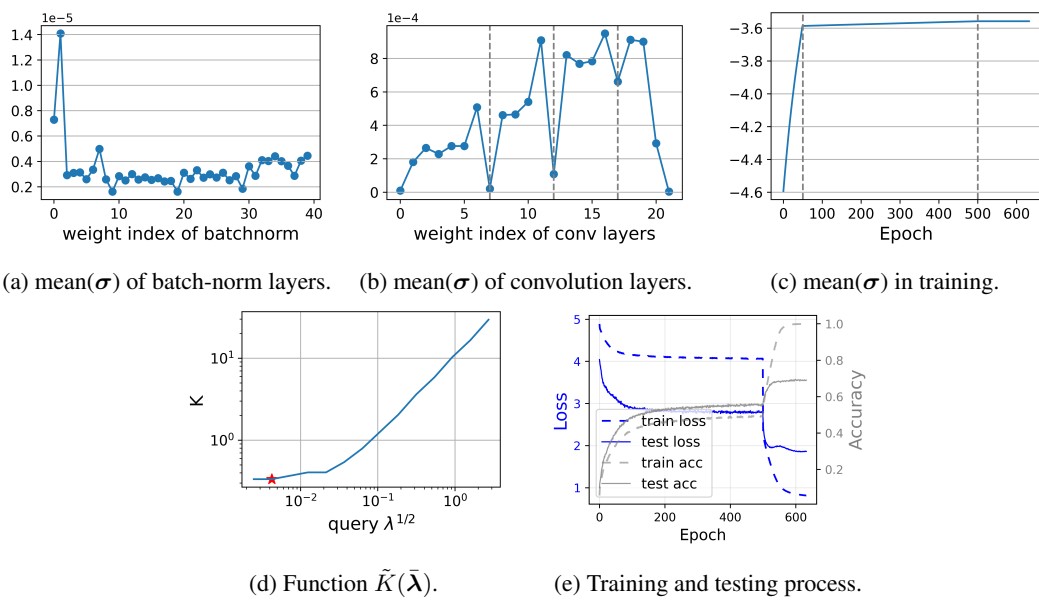

(a) mean($\boldsymbol{\sigma}$) of batch-norm layers.    (b) mean($\boldsymbol{\sigma}$) of convolution layers.    (c) mean($\boldsymbol{\sigma}$) in training.

(d) Function $\tilde{K}(\bar{\boldsymbol{\lambda}})$.    (e) Training and testing process.

Figure 11: Training details of ResNet18 on CIFAR100. The red star denotes the final $K$.

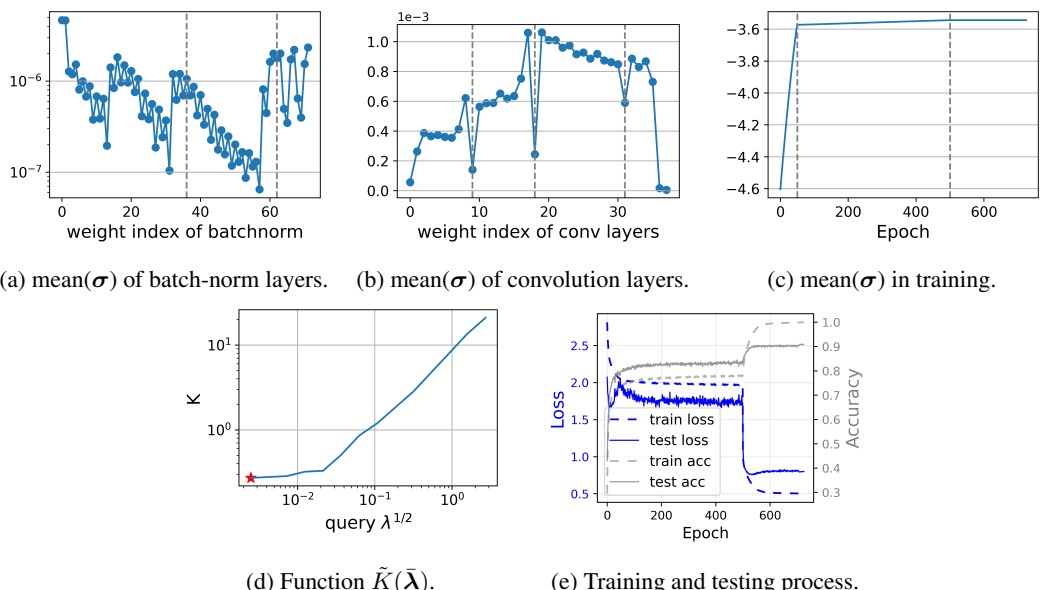

(a) mean($\boldsymbol{\sigma}$) of batch-norm layers.  (b) mean($\boldsymbol{\sigma}$) of convolution layers.  (c) mean($\boldsymbol{\sigma}$) in training.

(d) Function $\tilde{K}(\bar{\boldsymbol{\lambda}})$.  (e) Training and testing process.

Figure 12: Training details of ResNet34 on CIFAR10. The red star denotes the final $K$.

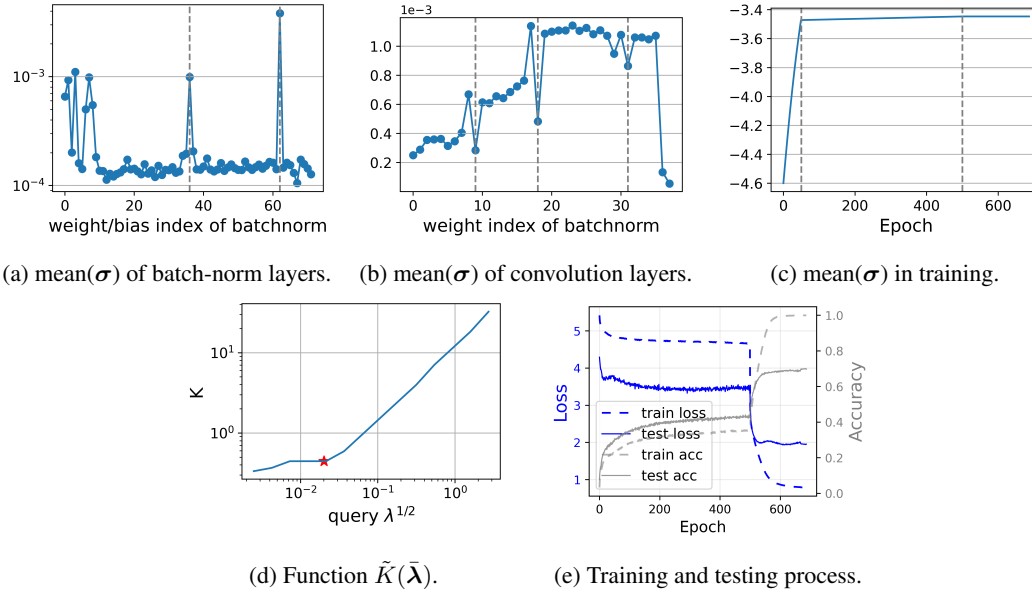

(a) mean($\boldsymbol{\sigma}$) of batch-norm layers.  (b) mean($\boldsymbol{\sigma}$) of convolution layers.  (c) mean($\boldsymbol{\sigma}$) in training.

(d) Function $\tilde{K}(\bar{\boldsymbol{\lambda}})$.  (e) Training and testing process.

Figure 13: Training details of ResNet34 on CIFAR100. The red star denotes the final $K$.

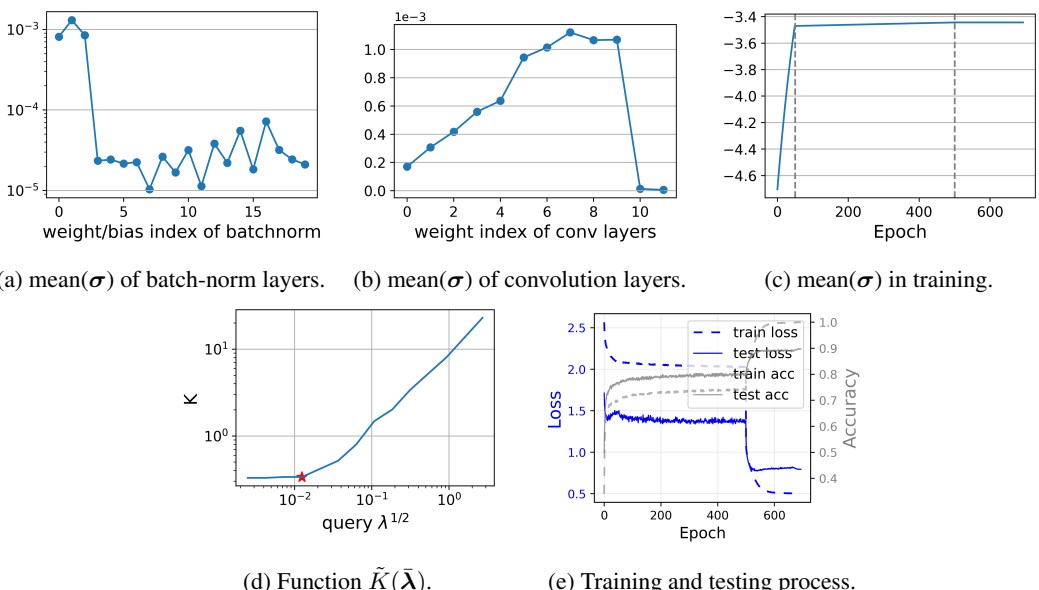

(a) mean($\boldsymbol{\sigma}$) of batch-norm layers.    (b) mean($\boldsymbol{\sigma}$) of convolution layers.    (c) mean($\boldsymbol{\sigma}$) in training.

(d) Function $\tilde{K}(\bar{\boldsymbol{\lambda}})$.    (e) Training and testing process.

Figure 14: Training details of VGG13 on CIFAR10. The red star denotes the final $K$.

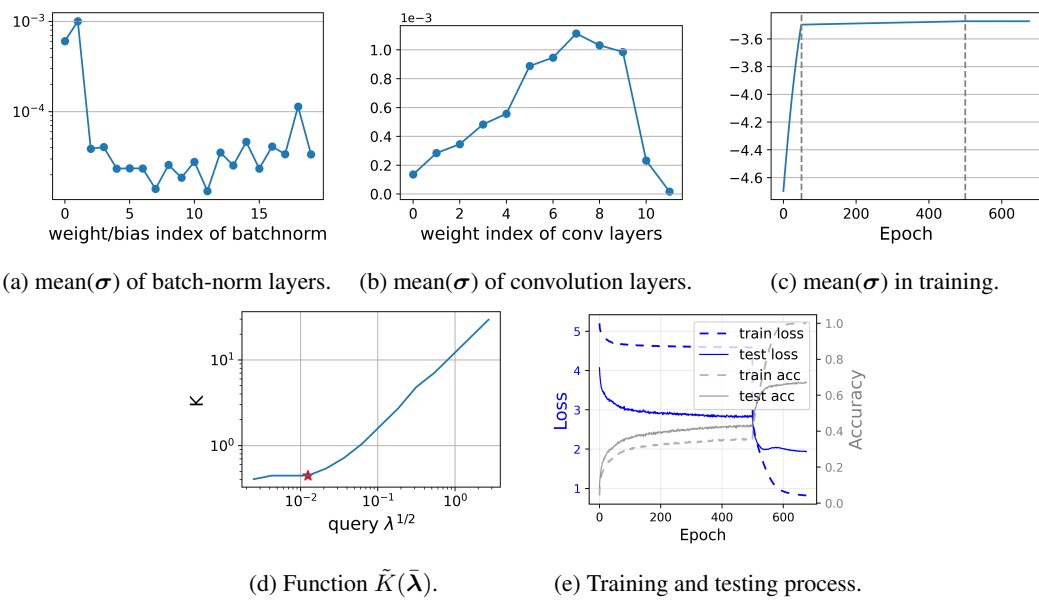

(a) mean($\boldsymbol{\sigma}$) of batch-norm layers.    (b) mean($\boldsymbol{\sigma}$) of convolution layers.    (c) mean($\boldsymbol{\sigma}$) in training.

(d) Function $\tilde{K}(\bar{\boldsymbol{\lambda}})$.    (e) Training and testing process.

Figure 15: Training details of VGG13 on CIFAR100. The red star denotes the final $K$.

