# OpenReview forum: "Unlocking Tuning-free Generalization: Minimizing the PAC-Bayes Bound with Trainable Priors"
_ICLR.cc/2024/Conference — Submitted to ICLR 2024_

### Official Review · Reviewer_Htid · 2023-10-29

**Soundness:** 4 excellent
**Presentation:** 3 good
**Contribution:** 2 fair
**Rating:** 6
**Confidence:** 3

**Summary:**

The paper proposes a principled PAC-Bayes framework for optimizing neural networks in classification. Specifically, the paper first extends the current PAC-Bayes bound for bounded loss functions to unbounded loss functions and introduces trainable priors. The end result is a practical AC-Bayes bound consisting of learnable parameters: model weights, prior parameters, posterior parameters, and a moment parameter. The algorithm jointly optimizes over all learnable weights. Compared to the existing optimization framework, the PAC-Bayes framework does not require extensive learning rate tuning and comes with built-in learnable regularization capabilities. Experimentally, the proposed framework achieves similar performance with extensively tuned existing pipelines.

**Strengths:**

* **Principled approach**: most of the proposed framework stems from rigorous derivation and extension to the PAC-Bayes learning theory. Specifically, the paper derives the PAC-Bayes bounds for both preset and learnable priors and discusses why it is reasonable to learn a prior in a data-driven way.

* **Practical novelty**: from a practical perspective, the proposed framework can potentially automatically learn its regularization strength during optimization without turning. Specifically, the KL divergence term $KL(Q|P)$ between the parametrized posterior and prior, using Gaussian distributions, reduces to learnable weight decays and noise injection. The regularization strength is controlled by the variance parameters in the prior and posterior respectively.

* **Distributional output**: Conventional optimization pipeline only yields a MLE point estimation, the proposed method returns a posterior distribution over model weights, which could be utilized for Bayesian inference with improved robustness and better uncertainty quantification.

**Weaknesses:**

* **Poor compatibility with data augmentation**: a major downside of the method is its poor compatibility with data augmentation techniques, which are important and beneficial to boost a model's performance. The reason is that the algorithm needs the exact number of training data sizes and data augmentation breaks this assumption. This is discussed by the authors in the appendix. Nevertheless, this is an important limitation of the approach.

* **Lack experiments**:  On CIFAR10 and CIFAR100 experiments, the proposed method is on par with existing optimization methods, which need extensive hyper-parameter tuning, especially for regularization heuristics. Given the theory-oriented nature of this paper, this is reasonable. However, it would be great to scale up experiments to medium/large datasets. More importantly, given the claim that the proposed framework automatically learns regularization, it is *necessary* to test its performance under a low-data regime for example using only 10% of CIFAR10 and CIFAR100. This will make the claim of being *nearly tuning-free* much more convincing.

* **Need more clarity on $K(\lambda)$**: The loss function derived from the PAC-Bayes bound has three important terms, the empirical data-driven loss, the KL divergence between the posterior and the prior, and $K(\lambda)$. It's not clear what role $K(\lambda)$ plays in the overall algorithm. Since it is estimated before training, it is also not clear how it affects the learning in terms of gradient.

**Questions:**

My major concern is over the experiments. While it is reasonable to have small-scale experiments, I would really like to see how the proposed method works for low-label situations since a major claim is its tunning-free capability. This means that it can adjust its built-in regularization strength to different dataset sizes.

It would make the claim much more convincing if the algorithm showed promise in adapting to varying numbers of training data. I would suggest the authors run a study on varying the number of training data, e.g., 10%, 50%, and report the performance and the learned variances for the prior and posterior. What do you expect the prior and posterior variances to be when the dataset size is small and how do the results support this?

I will be happy to raise my score if this concern is addressed convincingly.

---

> ### Author Response · Authors · 2023-11-14
>
> `1. Poor compatibility with data augmentation.`
>
> From our experiment in the appendix, the method works well with data augmentation, even though we don't have the theoretical explanation. Since data augmentation violates the i.i.d. assumption, having $10$ times augmented data does not mean having $10$ times new independent data. So, in the future, it would be good to explore if we can compute some sort of "effective" dimension of the data augmentation, which characterizes how much new information the data augmentation introduces, or what is the equivalent number of i.i.d. data that brings the same amount of information as the given data augmentation.
>
> `2. Lack experiments for the low-data regime.`
>
> We strongly agree that the small data and large model regimes are the hardest for PAC-Bayes training because the KL term would be large in this case. Running new experiments with our method is relatively fast, but performing the baseline search is quite time-consuming. We tried fine-tuning the pre-trained large language models to validate our method in the low-data regime. Unlike the experiments we did in the appendix, where the large pre-trained part of the model is fixed, and only the small classification layer is to be trained on the small fine-tuning dataset, we now allow both the pre-trained model and the classification layer to be trained using the small fine-tuning dataset. This is what people do in practice because its performance is much better than freezing the pre-trained model. However, this will make PAC-Bayes training challenging because the number of training samples is around 100 while the total number of training parameters is as large as $110$ million. Table 1 contains the result. It shows that PAC-Bayes training works better than all baselines for fine-tuning.
>
> Table1. Experimental Results for BERT-base-uncased Backbone Model.
> ||CoLA |SST |MNLI-m| QNLI| RTE|
> |---|---|---|---|---|--|
> |metrics|MCC|Accuracy|Accuracy|Accuracy|Accuracy|
> | | | | | |
> |Vanilla-tuning| .235| .773 |.369 |.702 |.589 |
> |Data Augmentation[1]|.171 |.817 |**.395**| .705 |.594|
> |Noise Injection[2]|.233 |.783 |.371 |.706 |.588|
> |LoRA[3]| .298 |.792| .385 |.669 |.592 |
> |Prefix-training[4]| .191 |.704 |.375 |.649 |.565|
> |BitFit[5]| .267 |.768 |.376| .647 |.588|
> |Ours| **.335** |**.834** |.387 |**.709** |**.601**|
>
> [1]. Sennrich, Rico, et al. "Improving neural machine translation models with monolingual data."
>
> [2]. Orvieto, Antonio, et al. "Explicit regularization in overparametrized models via noise injection."
>
> [3]. Hu, Edward J., et al. "Lora: Low-rank adaptation of large language models."
>
> [4]. Liu, Xiao, et al. "P-tuning: Prompt tuning can be comparable to fine-tuning across scales and tasks."
>
> [5]. Zaken, Elad Ben, et al. "Bitfit: Simple parameter-efficient fine-tuning for transformer-based masked language-models."
>
>
>
> `3. Need more clarity on $K(\lambda)$`
>
> $K(\lambda)$ is a function of $\lambda$, and $\lambda$ is the trainable parameters of the prior representing the variance. So $K(\lambda)$ affects what $\lambda$ can be achieved in training, and the prior variance $\lambda$, in turn, affects the posterior variance and the model parameters through their interaction in the bound.
>
> `4. Experiments with medium/large datasets.`
>
> We have obtained additional numerical results for image classification tasks. Our comparison of PAC-Bayes training with SGD, using a batch size of 128 on the TinyImageNet dataset, involved 100,000 images across 200 classes. Using ResNet18, the test accuracy of PAC-Bayes training is 53.2%, while the best test accuracy achieved with SGD/Adam/AdamW after hyperparameter searching is 46.4%/46.7\%/46.8\%. This further demonstrates the advantages of our proposed PAC-Bayes training.

---

> ### Author Response · Authors · 2023-11-17
>
> `5. adapting to varying numbers of training data`
>
> We compared PAC-Bayes training with SGD/Adam/AdamW on CIFAR10, using a batch size of 128, and allocated 10% of the training data for training and the remaining 90% for hyper-parameter searching in SGD/Adam/AdamW. With ResNet18, the test accuracy of PAC-Bayes is 67.8%, while the best test accuracies for SGD, Adam, and AdamW, after hyper-parameter searching, are 64.00%, 64.96%, and 65.59%, respectively. When training ResNet18 with all the training data using a batch size of 128, SGD typically achieves the best test accuracy among the baselines. However, AdamW outperforms SGD when using only 10% of the training data. This demonstrates the necessity of both hyper-parameter searching and choosing the appropriate optimizer for baselines. With the same setting and model on CIFAR100, the accuracies are as follows: SGD - 22.16\%, Adam - 24.95\%, AdamW - 25.68\%, and PAC-Bayes - 26.22\%.

---

> > ### Author Response · Authors · 2023-11-18
> >
> > Dear reviewer, we are ready to provide more details and respond to any questions you may have. Thank you very much!

---

> > > ### Comment · Reviewer_Htid · 2023-11-20
> > > **Question**
> > >
> > > Thank you for the additional experiments. I have increased my score to reflect this. However, I still have one question. As I asked in the initial review, could the authors also provide the statistics on the learned posterior and prior with varying numbers of training data? because they directly demonstrate how the proposed Bayesian framework adapts to varying degrees of evidence, e.g., training data.

---

> ### Author Response · Authors · 2023-11-21
>
> Thank you very much for raising the score!
>
> As suggested, we conducted additional experiments and observed that the variances of the learned posterior and prior change due to the different numbers of training samples.
>
> With batch size 128, by training with full/10\% of CIFAR10 on VGG13, the final standard deviations of posteriors are 0.032/0.010, and the mean of the standard deviations of priors are 0.042/0.0158. Training with full/10\% of CIFAR10 on ResNet18, the mean of the final variances of posteriors are 0.031/0.010, and the final variances of priors are 0.044/0.008.

---

> > ### Comment · Reviewer_Htid · 2023-11-21
> > **Re: questions**
> >
> > Does this make sense from a Bayesian perspective though? Shouldn't we expect a smaller variance of the posterior (i.e., uncertainty) given more data because we can be more certain about which weight/function generated the data?

---

> ### Author Response · Authors · 2023-11-21
>
> Thank you for the intriguing question! We feel it makes sense based on the following explanation. In order to fit more training data, the (final) model needs to be able to go further away from the random initial model,  which means smaller confidence should be given to the prior (i.e., larger prior variance) to allow this to happen (otherwise if we're very certain about the prior, then the model will just stay near the initial model and won't be able to fit the data).
> Since the prior and posterior variances are positively correlated, the posterior variance will also be larger. This is a possible explanation of why larger data leads to larger variance.
>
> However, we feel that there is another scenario, that is, when we have a very large amount of data compared to the model (i.e., the under-parametrized regime). Then, adding more data wouldn't pull the model much further from the initial model than it already is. In this case, more data should reduce the uncertainty and, therefore, lead to a smaller variance. We're currently doing more experiments on various amounts of training data of CIFAR10 and will report the result soon. However, we feel since CIFAR10 on ResNet/VGG is in the over-parametrized regime, we probably are not going to see this latter scenario.

---

> > ### Author Response · Authors · 2023-11-22
> >
> > We obtained the results for various data levels when training VGG13 with 10\%, 20\%, 40\%, and 100\% of the CIFAR10 samples, respectively. The corresponding standard deviation averaged over all the weights is 0.0092/0.0095/0.0103/0.0126. This is slightly different from the result we reported yesterday because yesterday, we accidentally used a small batch size (batch size = 128). Since it is known that if the batch size is small, then the variation among different batches introduces noise to the model, which would interfere with the posterior noise. Therefore, we now shift to use large batch size, and the result should be more accurate. Nevertheless, we still see the same tendency that the variance increases as data size grows.
> >
> > As mentioned in the previous post, we conjectured that this phenomenon is observed for the following reason: in order to fit more training data, the final model needs to deviate further from the randomly initialized model, therefore we have to be less certain about the prior to allow the model to be updated more.  To confirm this conjecture, we computed the distances between the initial and the final model for the different data sizes, i.e.,  we computed $||h-h_0||^2$ when using different sample sizes 10\%, 20\%, 40\%, and 100\%, the value of this term is 1.50/3.18/7.46/29.96. As we see, the more data we have, the further away the final trained model $h$ has deviated from $h_0$, and the growth of this quantity surpasses the growth of $m$ (recall in the PAC-Bayes bound with Gaussian prior/posterior, we have a term $\frac{||h-h_0||^2}{m}$, so this ratio matters). This necessitates assigning low confidence to the prior, meaning a larger prior variance is required to allow such deviations. If the prior has high certainty, the model tends to remain close to the initial model and fails to fit the data effectively. As the prior and posterior variances are positively correlated to optimize the KL-divergence in the PAC-Bayes bound, an increase in the prior variance leads to a larger posterior variance. This phenomenon could explain why a larger dataset results in increased variance.

---

### Official Review · Reviewer_YdhE · 2023-10-31

**Soundness:** 3 good
**Presentation:** 2 fair
**Contribution:** 2 fair
**Rating:** 5
**Confidence:** 3

**Summary:**

This paper examines training of deep neural networks via directly optimizing their PAC-Bayes bounds. This is done with the secondary goals of reducing reliance on hyperparameter searches and investigation of which regularization tricks/implicit biases can be omitted without compromising generalization performance. In doing so, the authors extend earlier results of Dziugaite & Roy to relax the assumption of bounded loss, by appealing to an exponential moment bound. While this assumption is not typically considered to be an issue, its relaxation may be of interest on its own.

**Strengths:**

The paper identifies that a weakness of many PAC-Bayes type bounds is the assumption of a bounded loss. Conditions that allow for this assumption to be relaxed are identified, and connections to sub-Gaussian bounds are highlighted.

**Weaknesses:**

Although the terminology surrounding prior and posterior distributions is consistent with Maurer 2004 and parts of the PAC-Bayes literature, the more recent prominence of Bayesian methods in the machine learning literature and their more specific use of these terms leaves room for confusion. Distinguishing between the distributions appearing in the bounds and the specific choice of the Gibbs posterior, for example, would be helpful (see e.g. https://arxiv.org/pdf/1605.08636.pdf, https://arxiv.org/pdf/2110.11216.pdf).

The overall contribution of this work seems minimal, since there is previous literature loosening the bounded loss assumption, and there is little additional information provided in the current work.

The posterior distribution given by $\mathcal{Q}_{\sigma}(h)$ does not appear to be a valid probability distribution, as defined in the paper.

**Questions:**

Can you please address the issues raised in the weaknesses section.

**Details Of Ethics Concerns:**

-

---

> ### Author Response · Authors · 2023-11-14
>
> `1. The overall contribution of this work seems minimal, since there is previous literature loosening the bounded loss assumption, and there is little additional information provided in the current work.`
>
> Thank you very much for the question. **We want to emphasize the gap between the theoretical bounds and applying them to training deep neural networks in practice.**
>
> Even though there are many existing PAC-Bayes bounds in the literature, to the best of our knowledge and according to our experiment, none of them works in this difficult setting (training deep CNN networks on Cifar10 with cross-entropy loss). More explicitly, when putting these bounds into practice, they are either non-applicable or too vacuous that the model does not even train (i.e., the training accuracy stays at $10\\%$ and does not increase). Therefore, we did not find a method that gives OK performance to compare with our method.
>
> For example, in the paper https://arxiv.org/pdf/1605.08636.pdf, the reviewer mentioned that Corollary 4 has a PAC-Bayes bound for unbounded loss. However, the $s^2$ is too large in practice. If we train the model with this bound, then the training accuracy stays at $10\\%$ and does not increase.
>
> Another example mentioned in the paper, "Efron-stein pac-bayesian inequalities," has a PAC-Bayes bound for unbounded loss. $E_{h\sim Q}\ell(h;D)\leq E_{h\sim Q}\ell(h;S)
>         + \sqrt{\frac{1}{m} E_{h\sim Q} \left[\ell_1(h;S) + E_{z'\sim D} \ell^2(h;z')  \right]\mathrm{KL}(Q||P)} +\frac{1}{m}$.
>
> $\ell_1(h;S):=\frac{1}{m} \sum_{i=1}^m \ell^2(h;z_i)$, and $z'\sim D$ is a test sample drawn from the data distribution. This bound holds for any unbounded loss with a finite second-order moment. However, this bound is semi-empirical, meaning that the bound contains a term, $E_{h\sim Q}E_{z'\sim D} \ell^2(h;z')$, that is hard to estimate from data.
>
> Another example is in the paper [1], where the bound requires the loss to be bounded, and works for two-class classification, rendering it unsuitable for direct use with CIFAR10/CIFAR100.
>
> [1] Dziugaite, Karolina, et al.. "Computing Nonvacuous Generalization Bounds for Deep (Stochastic) Neural Networks with Many More Parameters than Training Data."
>
> **To the best of our knowledge, we did not find any other available bounds that are suitable to train deep neural networks directly with unbounded loss and trainable priors.**
>
> `2. The posterior distribution given by $Q_\sigma(h)$ does not appear to be a valid probability distribution, as defined in the paper.`
>
> Does the reviewer refer to the definition here?  $Q_\sigma(h):=h+Q_\sigma(0)$. We apologize for the confusion and have changed the notation in the revision (please see the highlighted part in red on page 4). Here, by this notation, we meant $Q_\sigma(h)$ is the distribution of $Y = X+h$, with $X\sim Q_\sigma(0)$, and $Q_\sigma(0)$ is an arbitrary 0 mean distribution parametrized by $\sigma$. So $Q(h)$ is a valid distribution. In the case of Gaussian posteriors in later sections of the paper, this definition leads to $Q_\sigma(h):=N(h, diag(\sigma))$, where $N(h, diag(\sigma))$ refers to the multivariate Gaussian with mean 0 and covariance $diag(\sigma)$

---

> ### Author Response · Authors · 2023-11-18
>
> `3. Although the terminology surrounding prior and posterior distributions is consistent with Maurer 2004 and parts of the PAC-Bayes literature, the more recent prominence of Bayesian methods in the machine learning literature and their more specific use of these terms leaves room for confusion. Distinguishing between the distributions appearing in the bounds and the specific choice of the Gibbs posterior, for example, would be helpful (see e.g. https://arxiv.org/pdf/1605.08636.pdf, https://arxiv.org/pdf/2110.11216.pdf).`
>
> We're not sure if we have understood your question correctly. We believe our notation is consistent with the literature you mentioned https://arxiv.org/pdf/2110.11216.pdf. More explicitly, our Theorem 4.1 is of a similar type to Theorem 2.1 in the referenced paper, which deals with general posteriors. The referenced paper also includes several theorems for Gibbs posterior, but our work does not. This is because, according to our method, the final posterior we used for prediction is computed as the minimizer of the PAC-Bayes bound,  which does not admit a closed-form expression like Gibbs posterior.
>
> In addition, because we also optimize during the optimization over the prior, there is no existing bound in the literature that can be used to bound the population error of this posterior. Therefore, we build Theorem 4.2 to characterize the algorithm's performance. We believe this is the first bound of this type which allows data-dependent prior (previous work all require fixed prior).  Please let us know if this response addresses your question, or if we have completely missed the point you were making.

---

> > ### Author Response · Authors · 2023-11-18
> >
> > Dear reviewer, we kindly ask for your feedback on our response. We are eager to offer any additional information and address your queries. Many thanks.

---

> > > ### Comment · Reviewer_YdhE · 2023-11-22
> > >
> > > Thanks for your response, I have adjusted my score accordingly.

---

> > > > ### Author Response · Authors · 2023-11-22
> > > >
> > > > `Tightness of the generalization bound. (numerical comparison)`
> > > >
> > > > To further alleviate concerns about tightness compared to other bounds, we have numerically compared our proposed PAC-Bayes bound with the existing one referenced in the literature (Corollary 4, https://arxiv.org/pdf/1605.08636.pdf) mentioned by the reviewer, which is also for unbounded loss.
> > > >
> > > > Figure (https://tinyurl.com/ycksnjvb) displays the result. Even though the forms of the bounds look similar, their actual numerical values are significantly different. As shown in this figure, our proposed bound is far tighter than this baseline one. The sub-Gaussian norm term ($s^2$) in this baseline bound alone is larger than $100$, making it not appealing in practice.

---

### Official Review · Reviewer_geV1 · 2023-10-31

**Soundness:** 3 good
**Presentation:** 3 good
**Contribution:** 2 fair
**Rating:** 5
**Confidence:** 4

**Summary:**

This work proposes minimising a novel PAC-Bayes bound as a new objective instead of the usual cross entropy loss. This novel PAC-Bayes bound applies for unbounded losses and further allows for (weakly) training the prior. Besides the benefits of minimising an upper bound to the generalisation error, the authors further observe that such an objective is significantly more stable with respect to the choice of various training hyper-parameters such as learning rate, batch size, weight decay etc. This thus alleviates a tedious hyper-parameter search, leading to speed-ups of a single training cycle.

**Strengths:**

1. Improving upon the optimisation objectives in deep learning is a very under-studied avenue. Especially PAC-Bayes bounds moreover are very principled theoretically as they serve as upper bounds to the generalisation error. Making such techniques more practical and showing benefits over standard training would be very important and potentially impactful.
2. The paper is very well-written and gives a very accessible introduction to PAC-Bayes bounds, how they have been used and what weaknesses of prior work the authors aim to address. I really enjoyed reading this part of the paper! The extension to trainable priors also seems non-trivial (although I have some questions on this later in the review) and the authors manage to match performance of baselines without too much tuning of hyper-parameters.

**Weaknesses:**

1. The authors develop a novel and arguably tighter PAC-Bayes bound but never put it to test (at least from what I could see). I don’t see any generalisation bounds reported in the paper, so it is hard to gauge how much of a contribution the novel bound is in terms of tightness, compared to the previous works cited in this paper. I understand that this is not the main focus of the paper but it would definitely strengthen its technical contribution, i.e. the bound itself. I would have also liked to see a discussion regarding the trade-offs of having a learnable prior. What if the set of priors is so large that it includes the posterior, i.e. the KL term could be set to zero perfectly. How large would the resulting penalisation term be? Is there ever a scenario where this could lead to a non-vacuous bound?  Also, how does this learnable prior compare to methods that perform a (discrete) grid search over the prior and then perform a union bound, resulting in an additional  penalty term (e.g. [1])?
2. The paper focuses on the efficiency of their proposed method, which arises due to the absence of hyper-parameter tuning. I am not convinced by these claims based on the empirical experiments performed in this work:
\
\
    a) There seems to be a certain arbitrariness as to what hyper-parameters are simply chosen ad-hoc and turn out to work well. The $\gamma_1$ and $\gamma_2$ values are for instance set to values that work well for the vision tasks but they actually need adaptation for the graph and text tasks, suggesting that some tuning is actually needed. I also could not find a stability analysis for the default choices, were those just the first values tried or was an initial grid search actually needed to find those values? There is also a warmup period only detailed in the Appendix to ensure faster convergence, how was the duration of this set?  I also would like to see a comparison to SGD/Adam/AdamW, where the default hyper-parameter settings are used. For instance, I very rarely see anyone changing the default momentum value for simple tasks such as CIFAR10/CIFAR100. I would also like to see a plot depicting how sensitive the baselines are to individual parameters such as the learning rate, momentum etc, I think the results are already in Figure 1 but need to be re-grouped accordingly.
\
\
    b) The method is strongly more involved conceptually than the standard objectives. First, K_min needs to be estimated before-hand and it’s not clear how the quality of such an approximation affects results. The optimisation has to be split into two phases, where the objective is reduced in the second part to ensure better convergence properties. The objective itself is naturally also more complicated although this might be fine as it also comes with theoretical guarantees. The hurdle for practitioners to use this framework might be rather high and I’m not convinced at this point if it is worth the effort.
\
\
    c) My biggest concern however is due to the training time required for the proposed method. A single run on CIFAR10, according to Table 8 in the Appendix takes roughly 7000 seconds, while for SGD the same takes roughly 600 seconds. This means that one can roughly perform 12 SGD runs in the time it takes to perform a single run with the PAC-Bayes technique. I would argue first that (1) it does not take 12 runs of SGD to find an acceptable hyper-parameter setting, especially not on simple tasks such as CIFAR10/CIFAR100. (2) Even if 12 runs are required, I would argue that one has obtained a better understanding of the task in the sense of what techniques work etc, and subsequent re-runs of the same method will be very cheap. Moreover, one could even make use of those 12 runs and build an ensemble, perform uncertainty quantification etc. I also believe that this should be discussed more transparently in the main text, instead of the Appendix.


[1] Dziugaite and Roy, Computing Nonvacuous Generalization Bounds for Deep (Stochastic) Neural Networks with Many More Parameters than Training Data

**Questions:**

See above section

---

> ### Author Response · Authors · 2023-11-14
>
> `1. I don’t see any generalisation bounds reported in the paper, so it is hard to gauge how much of a contribution the novel bound is.`
>
> Thank you very much for the question. The short answer is that despite many existing PAC-Bayes bounds in the literature, none of them works in our settings (training deep CNN networks on Cifar10 with cross-entropy loss). More explicitly, when putting these bounds into practice, they are too vacuous that the model does not even train (i.e., the training accuracy stays at $10\%$ and does not increase). Therefore, we did not find a method that gives OK performance to compare with.
>
> **We want to emphasize that there is a gap between the theoretical bounds and applying them in training deep neural networks in practice.**
>
> For example, in the paper "Efron-stein pac-bayesian inequalities," the bound is provided as:
>
> $E_{h\sim Q}\ell(h;D)\leq E_{h\sim Q}\ell(h;S)
>     + \sqrt{\frac{1}{m} E_{h\sim Q} \left[\ell_1(h;S) + E_{z'\sim D} \ell(h;z')^2  \right]\mathrm{KL}(Q||P)} +\frac{1}{m}$.
>
> $\ell_1(h;S):=\frac{1}{m} \sum_{i=1}^m \ell(h;z_i)^2$, and $z'\sim D$ is a test sample drawn from the data distribution. This bound holds for any unbounded loss with a finite second-order moment. However, the term
> $E_{h \sim Q} E_{z' \sim D} \ell(h;z')^2$ is almost as difficult to estimate as the generalization error itself.
>
> **To the best of our knowledge, we did not find any other available bounds that are suitable to train deep neural networks directly with unbounded loss and trainable priors.**
>
> In the paper (Dziugaite and Roy) mentioned by the reviewer, the bound requires that the loss be bounded for two-class classification, which can not be used for CIFAR10/CIFAR100 directly.
>
> `2. I would have also liked to see a discussion regarding the trade-offs of having a learnable prior. `
>
>  Thanks for bringing up this critical point. In the theoretical guarantee (Theorem 4.2), it says our PAC-Bayes training is valid if the correction terms $\eta +\frac{\log(n(\epsilon)}{\hat \gamma m}$ is small. Note that the $\log(n(\epsilon))$ in the correction term represents how rich the set of priors is. So, if the set of priors we're searching over is too rich/large, then this $\log(n(\epsilon)$ would make the correction term large, and the PAC-Bayes training will no longer be reliable. In practice, we observe that for deep network on cifar10, setting the prior to having 1 degree of freedom per layer is fine, but setting the prior to having 1 degree of freedom per weight is too much. This is the motivation of the proposed layerwise prior.
>
> `3.The $\gamma_1$ and $\gamma_2$ values are, for instance set to values that work well for the vision tasks, but they actually need adaptation for the graph and text tasks, suggesting that some tuning is actually needed.`
>
> We used the same range of $\gamma$ $[0.5,10]$ across different tasks.  We can add a sensitivity analysis of the upper and lower bounds of this range to the paper. In short, the result is quite insensitive to the lower bound $\gamma_1$ and is relatively more sensitive to the upper bound $\gamma_2$. So, we ran the CIFAR10 experiment several times to obtain a good value of $\gamma_2$ and fixed it to this value in all other settings.

---

> ### Author Response · Authors · 2023-11-14
>
> `4. There is also a warmup period only detailed in the Appendix to ensure faster convergence, how was the duration of this set?`
>
> The warmup is not necessary and is only used to accelerate convergence. Please refer to the answer to the next question for more explanation of the warmup.
>
> `5. My biggest concern however is due to the training time required for the proposed method.`
>
> The long wall time of our method is due to the coding issue.  Pytorch
> is not optimized for the PAC-Bayes training. Currently, we use for-loops in Python to inject the posterior noise in each iteration, and that makes the algorithm slow. The conclusion will change if we look at the actual complexity of the algorithms. For both the normal SGD/Adam training and our method, the computation bottleneck in each iteration is the backpropagation, so we may use the total number of backpropagations to represent the complexity.  Although our method needs the gradients for both the model and the variance, they can be computed from the same quantity obtained by a common backpropagation. Therefore, our method also only requires 1 backpropagation per iteration.  The total number of backpropagations in a single run is thus equal to the total number of iterations, that is:
>
> the number of epoch $* \frac{\textrm{training data size}}{\textrm{batch size}}$
>
> The table below (Table 9 in the appendix) provides this number for CIFAR10/CIFAR100 with ResNet18.
>
> ||CIFAR10	| CIFAR100|
> |---|---|--|
> |SGD	|11501	|17095|
> |Adam	|28508	|28335|
> |AdamW	|24761	|9560|
> |scalar	|15912	|15216|
> |layer	|15960	|15168|
>
> It can be seen that the complexity of the methods is similar. We can expect with code optimization, the wall-time issue can be fixed.
>
> The "trick" like the warm-up iteration mentioned in the supplementary, is used to accelerate the convergence without affecting the final result. Once the code optimization is done, this trick can be removed.

---

> ### Author Response · Authors · 2023-11-16
>
> `7. Performance of default settings of baselines.`
>
> Since for SGD, Pytorch has 0 as the default momentum, we guess by default, you mean the setting people commonly use, i.e., momentum $= 0.9$ (please correct us if we're wrong). We are not sure about the common choice of weight decay, so we found the default configuration for ResNet from [1], where it sets the learning rate to 0.1, weight decay to 1e-4, and momentum to 0.9, and uses a learning rate scheduler (reducing by a factor of 10 when the error plateaus). But with this parameter setting, the test accuracy of ResNet18 on CIFAR10 and CIFAR100 is only 85.21\% and 60.66\%, respectively, which is approximately 4\% to 5\% lower than that achieved with the best-tuned settings. This suggests when changing network type or dataset, in order to achieve the best performance, hyper-parameter tuning is still needed.
>
> The default learning rate for Adam is 1e-3, and the default weight decay is 0. Using Adam's default settings, the test accuracies for VGG13 and ResNet18 on CIFAR10 and CIFAR100 are displayed in Tables 1 and 2. Although the best test accuracy is similar to that obtained with Adam's default settings at a batch size of 128, the discrepancy is more pronounced with a batch size of 2048. Optimal test accuracy for VGG13 is attainable only with larger batch sizes, where the default settings are ineffective. Our PAC-Bayes training achieves the best test accuracy across various settings.
>
> Table1. The test accuracy when using the default setting of Adam with a batch size of 128. The accuracy inside the $()$ denotes the best test accuracy with a batch size of 128.
> ||CIFAR10|CIFAR100|
> |---|---|---|
> |VGG13|88.3 (88.5)|63.3 (63.7)|
> |ResNet18|87.5 (87.5)|61.5 (61.6)|
>
> Table2. The test accuracy when using the default setting of Adam with batch size as 2048. The accuracy inside the $()$ denotes the best test accuracy with a batch size of 2048.
> ||CIFAR10|CIFAR100|
> |---|---|---|
> |VGG13|86.6 (90.7)|58.2 (66.2)|
> |ResNet18|84.3 (87.7)|56.3 (65.4)|
>
> The default learning rate for AdamW is 1e-3, and the default weight decay is 1e-2. Under AdamW's default settings, the test accuracies for VGG13 and ResNet18 on CIFAR10 and CIFAR100 are presented in Tables 3 and 4. Although these default settings generally yield the best test accuracy for AdamW when using small batch size, they are not as effective as Adam/SGD, making optimizer selection essential. Our PAC-Bayes training achieves better test accuracy than AdamW.
>
> Table3. The test accuracy when using the default setting of AdamW with a batch size of 128. The accuracy inside the $()$ denotes the best test accuracy with a batch size of 128.
> ||CIFAR10|CIFAR100|
> |---|---|---|
> |VGG13|88.4 (88.4)|61.8 (61.8)|
> |ResNet18|87.9 (87.9)|60.2 (61.4)|
>
> Table4. The test accuracy when using the default setting of AdamW with batch size as 2048. The accuracy inside the $()$ denotes the best test accuracy with a batch size of 2048.
> ||CIFAR10|CIFAR100|
> |---|---|---|
> |VGG13|86.7 (87.2)|57.7 (61.0)|
> |ResNet18|84.5 (84.9)|56.1 (58.9)|
>
> [1] He, Kaiming, et al. "Deep residual learning for image recognition."

---

> > ### Author Response · Authors · 2023-11-16
> >
> > `8. Sensitivity analysis of baselines. (Part 1)`
> >
> > To best demonstrate the sensitivity of the hyper-parameter selection, we organized the test accuracy below for ResNet18. Considering the search efficiency, we searched the hyper-parameter one by one. For SGD, we first searched the learning rate, set the momentum and the weight decay as 0 (both are default values for SGD), and then used the best learning rate to search for the momentum. At last, the best-searched learning rate and momentum are used to search for weight decay. For Adam, we searched the learning rate, weight decay, and noise injection in an order similar to SGD. Since AdamW and Adam are the same when setting the weight decay as 0, we searched for the best weight decay based on the best learning rate obtained from searching on Adam.
> >
> > The tables below show that the standard deviation when searching different hyper-parameters can be significant. We can also see the increase of the maximum test accuracy when adding a new regularization (both learning rate and momentum are implicit regularization as [1][2]), showing that even momentum is necessary to search in CIFAR10 and CIFAR100. Compared with Adam/AdamW, SGD is more sensitive to different hyper-parameters, but we need SGD to achieve the best test accuracy when using the small batch size (128). The sensitivity of the search shows the advantages of our proposed PAC-Bayes training. Note that for CIFAR10 and CIFAR100, we use the testing dataset to search hyper-parameters of baselines. PAC-Bayes training can match the test accuracy directly without any information on the test data.
> >
> > Table 1. Test accuracy for ResNet18 on CIFAR10 with batch size 128 when tuning different hyper-parameters.
> > |optimizer|parameter|Max|Min|Mean|Std|
> > |---|---|---|---|---|---|
> > | SGD | learning rate (1e-3 to 0.2) | 84.8 | 59.8 | 75.3 | 10.2 |
> > | SGD | momentum (0.3 to 0.9) | 85.6 | 84.1 | 85.0 | 0.8 |
> > | SGD | weight decay (1e-4 to 1e-2) | 89.9 | 85.9 | 88.2 | 1.6 |
> > | | | | | | |
> > | Adam | learning rate (1e-3 to 0.2) | 87.5 | 82.5 | 84.5 | 1.7 |
> > | Adam | weight decay (1e-4 to 1e-2) | 86.7 | 82.8 | 85.5 | 1.6 |
> > | Adam  | noise injection (5e-4 to 1e-2) | 87.3 | 85.6 | 86.5 | 0.9 |
> > | | | | | | |
> > | AdamW | weight decay (1e-4 to 1e-2) | 87.9 | 86.6 | 87.4 | 0.5 |
> > | AdamW | noise injection (5e-4 to 1e-2) | 87.2 | 86.0 | 86.8 | 0.6 |
> >
> > Table 2. Test accuracy for ResNet18 on CIFAR10 with batch size 2048 when tuning different hyper-parameters.
> > |optimizer|parameter|Max|Min|Mean|Std|
> > |---|---|---|---|---|---|
> > | SGD | learning rate (1e-3 to 0.2) | 72.0 | 56.5 | 62.8 | 6.2 |
> > | SGD | momentum (0.3 to 0.9) | 75.0 | 72.3 | 74.0 | 1.4 |
> > | SGD | weight decay (1e-4 to 1e-2) | 85.4 | 72.0 | 77.8 | 6.2 |
> > | | | | | | |
> > | Adam | learning rate (1e-3 to 0.2) | 85.1 | 70.0 | 78.4 | 6.1 |
> > | Adam | weight decay (1e-4 to 1e-2) | 86.4 | 77.4 | 83.7 | 3.8 |
> > | Adam | noise level (5e-4 to 1e-2) | 87.7 | 85.1 | 86.4 | 1.2 |
> > | | | | | | |
> > | AdamW | weight decay (1e-4 to 1e-2) | 84.8 | 84.5 | 84.7 | 0.1 |
> > | AdamW | noise level (5e-4 to 1e-2) | 84.9 | 82.7 | 83.9 | 0.9 |
> >
> > Table 3. Test accuracy for ResNet18 on CIFAR100 with batch size 128 when tuning different hyper-parameters.
> > |optimizer|parameter|Max|Min|Mean|Std|
> > |---|---|---|---|---|---|
> > | SGD | learning rate (1e-3 to 0.2) | 58.7 | 33.7 | 48.5 | 10.3 |
> > | SGD | momentum (0.3 to 0.9) | 59.0 | 51.8 | 56.6 | 4.1 |
> > | SGD | weight decay (1e-4 to 1e-2) | 64.0 | 56.6 | 61.8 | 3.0 |
> > | | | | | | |
> > | Adam | learning rate (1e-3 to 0.2) | 61.5 | 27.1 | 44.6 | 14.4 |
> > | Adam | momentum (0.3 to 0.9) | 58.4 | 43.1 | 53.0 | 6.3 |
> > | Adam | weight decay (1e-4 to 1e-2) | 61.6 | 61.0 | 61.2 | 0.3 |
> > | | | | | | |
> > | AdamW | weight decay (1e-4 to 1e-2) | 61.1 | 57.3 | 59.3 | 1.8 |
> > | AdamW | noise level (5e-4 to 1e-2) | 61.4 | 58.8 | 59.8 | 1.1 |
> >
> > Table 4. Test accuracy for ResNet18 on CIFAR100 with batch size 2048 when tuning different hyper-parameters.
> > |optimizer|parameter|Max|Min|Mean|Std|
> > |---|---|---|---|---|---|
> > | SGD | learning rate (1e-3 to 0.2) | 44.2 | 27.8 | 35.8 | 6.8 |
> > | SGD | momentum (0.3 to 0.9) | 52.2 | 45.9 | 49.2 | 3.1 |
> > | SGD | weight decay (1e-4 to 1e-2) | 61.5 | 54.5 | 57.3 | 3.2 |
> > | | | | | | |
> > | Adam | learning rate (1e-3 to 0.2) | 56.3 | 31.0 | 42.9 | 11.8 |
> > | Adam | weight decay (1e-4 to 1e-2) | 59.4 | 56.0 | 57.6 | 1.4 |
> > | Adam | noise level (5e-4 to 1e-2) | 65.4 | 59.0 | 62.0 | 3.3 |
> > | | | | | | |
> > | AdamW | weight decay (1e-4 to 1e-2) | 56.5 | 55.9 | 56.1 | 0.2 |
> > | AdamW | noise level (5e-4 to 1e-2) | 58.9 | 57.1 | 58.3 | 0.8 |
> >
> > [1] Smith, Samuel L., et al. "On the origin of implicit regularization in stochastic gradient descent."
> >
> > [2] Ghosh, Avrajit, et al. "Implicit regularization in Heavy-ball momentum accelerated stochastic gradient descent."

---

> ### Author Response · Authors · 2023-11-16
>
> `8. Sensitivity analysis of baselines. (Part 2)`
>
> We also compared PAC-Bayes training with SGD/Adam/AdamW on CIFAR10, using a batch size of 128, and allocated 10\% of the training data for training and the remaining 90\% for hyper-parameter searching in SGD/Adam/AdamW. With ResNet18, the test accuracy of PAC-Bayes is 67.8\%, while the best test accuracies for SGD, Adam, and AdamW, after hyper-parameter searching, are 64.00\%, 64.96\%, and 65.59\%, respectively. When training ResNet18 with all the training data using a batch size of 128, SGD typically achieves the best test accuracy among the baselines. However, AdamW outperforms SGD when using only 10\% of the training data. This demonstrates the necessity of both hyper-parameter searching and choosing the appropriate optimizer for baselines.

---

> > ### Author Response · Authors · 2023-11-18
> >
> > Dear reviewer, we politely request your feedback on our response. We would love to provide additional information and answer your questions. Thank you very much.

---

> > > ### Comment · Reviewer_geV1 · 2023-11-20
> > > **Reply**
> > >
> > > I thank the authors for their very extensive feedback!
> > >
> > > **Generalisation bound**: Thank you for the clarification. (1) Even if there is no baseline to compare against, the generalisation bounds themselves would still be interesting (e.g. how tight are they etc). (2) If I recall correctly, (Dziugaite and Roy) did evaluate their algorithm on a binary version of MNIST, which authors could compare against if I don't miss anything?
> > >
> > > **$\gamma_1$ and $\gamma_2$**: So if I understand correctly, $\gamma_2$ did indeed need some tuning? If yes, how many attempts were needed on CIFAR10 to find a stable value?
> > >
> > > **Training time**: I understand that if properly optimised, the approach might be more competitive but what is stopping the authors from doing so? Right now, the method is roughly 12 times slower, which is really not attractive for practitioners which might not even have the compute to perform such a long run. The fact that the authors did not implement an optimised version seems to again highlight that the suggested approach is not straight-forward and introduces some additional overhead on the user's side. Of course there is a trade-off here between complexity and computational gains obtained due to less hyper-parameter optimisation, but in its current shape I remain unconvinced that the proposed method is useful to practitioners right now.
> > >
> > > **Stability of hyper-parameters:** I thank the authors for running this experiment, this indeed helps to strengthen the contribution. Given this analysis, how many runs of SGD/ADAM(W) are needed to obtain reasonable performance? Are their any other works/heuristics that could guide hyper-parameter search that one should compare against? I'm unsure how the baselines should be compared against, i.e. how many hyper-parameter searches should be done. What about randomly sampling from potential hyper-parameter settings without replacement and test how many draws are needed to reach a certain performance? Right now I find it very hard to assess how much could be potentially saved by the suggested method (given that it can be implemented more efficiently).
> > >
> > >
> > > All in all I'm still reluctant to increase my score. The theoretical contribution seems nice but it is not really evaluated empirically, i.e. the advantages of the PAC-Bayes bound itself over prior work (or even whether it's vacuous or not) cannot be assessed. The authors focus more on the practical relevance regarding hyper-parameter search, but the current implementation is very slow, allowing for 10 runs of the same experiment using SGD. While theoretically, the implementation can be improved, I find this a bit a weak argument as the focus of this paper is on providing a faster and more stable way to avoid hyper-parameter search but currently this is not the case.

---

> ### Author Response · Authors · 2023-11-21
>
> **Thank you very much for thinking our theoretical contribution is nice!**
>
> Here, we want to further address the reviewer's questions.
>
> `1. Tightness of the generalization bound.`
>
> (1) As shown in the first 500 epochs (Stage 1) of Figures 10e-15e in the Appendix, the test loss decreases along with the training loss (our PAC-Bayes bound), and the training loss (our PAC-Bayes bound) is always higher than the test loss. Therefore, our PAC-Bayes bound is tight because: 1. it is a valid bound, and 2. the network can be optimized effectively with this bound.
>
> (2) In Dziugaite and Roy, authors used 1 and -1 to represent classes $\{0,...,4\}$ and $\{5,...,9\}$. So, it is a binary classification problem with bounded loss. We target multi-class classification tasks with unbounded loss. So, our PAC-Bayes bound is essentially different from the one in the paper (Dziugaite and Roy).
>
> `2.` $\gamma_1$ `and` $\gamma_2$
>
> We only tried $10$, $20$ for $\gamma_2$ on ResNet18 with CIFAR10. After checking the numerical results, **we selected $10$ as the default and fixed it for all experiments and all models (including NLP models such as GPT, GNN models for citation graphs, and CNN models for image classification)**. So, there is no searching on $\gamma$.
>
> `3. Training time`
>
> There are currently two challenges for us to achieve the theoretically predicted speed and make the PAC-Bayes training as fast as SGD: software and hardware limitations.
>
> a. Limited access to the source code of PyTorch: Implementing noise injection in PyTorch can be challenging and time-consuming. Adding noise to algorithms like SGD or Adam slows each iteration down to the same speed as PAC-Bayes training, which is around three times slower than a standard SGD iteration. The training time for one epoch needed for noise injection and our PAC-Bayes training are both 27s for ResNet18 on CIFAR10 with one RTX3090 GPU. For pure Adam without noise injection, it only takes 10s for one epoch. This is despite the fact that the complexity of these methods is the same. Noise injection in Julia does not suffer from this issue, but we're less familiar with Julia.
>
> b. GPU memory limitations impact our training speed. While our PAC training benefits from using large batch sizes without performance loss, and larger batch sizes could speed up the algorithm by processing more data simultaneously, our GPU memory limits us. For example, we use a batch size of 2048 for PAC-Bayes training and 128 for SGD. Ideally, training with a batch size of 2048 should be 16 times faster than with 128, but due to GPU memory constraints, it's only twice as fast.
>
> These are soft constraints to the community but are hard constraints to us.
>
> `4. Stability of hyper-parameters
>
> **The current tuning seems efficient thanks to existing knowledge, which allows people to follow established hyper-parameters. However, this is not the case if we lack such knowledge about which hyper-parameters are effective.** In the next two sets of experiments, we used the same set of hyper-parameters for searching, as mentioned in the paper for CIFAR10/CIFAR100, showing sensitivity when lacking the knowledge for searching.
>
> On CIFAR10 with a batch size of 128, we allocated 10\% of the training data for actual training and the remaining 90\% for hyper-parameter searching in baselines. With ResNet18, the test accuracy of PAC-Bayes is 67.8\%, while the best test accuracies for SGD/Adam/AdamW are 64.00\%/64.96\%/65.59\%. *When training ResNet18 with all the training data and a batch size of 128, SGD typically achieves the best test accuracy among the baselines. However, with only 10\% of the training data, AdamW outperforms SGD.*
>
> We also trained ResNet18 with a batch size of 128 on the TinyImageNet. The test accuracy of PAC-Bayes training is 53.2\%, while the best test accuracy achieved with SGD/Adam/AdamW after hyperparameter searching is 46.4\%/46.7\%/46.8\%. *This further demonstrates the advantages of our proposed PAC-Bayes training.*
>
> While SGD is faster on image tasks, it doesn't consistently deliver the best performance. Even with some knowledge about hyper-parameter searching, it might still be inadequate. Moreover, there are 4 parameters to tune for SGD and 5 for Adam, making it relatively easy to conduct over 12 tuning iterations in total, even if we ignore Adam needs longer training time than SGD.

---

> ### Author Response · Authors · 2023-11-22
>
> `Tightness of the generalization bound. (numerical comparison)`
>
> It is difficult to compare our bound with those for bounded loss (e.g., Dziugaite and Roy) because if the loss is bounded, then our bound has similar tightness as theirs, and if the loss is unbounded, their bound (for bounded loss) cannot be applied. But to partially address your concern, we have numerically compared our bounds with some recent bounds in the literature for unbounded loss, as mentioned by Reviewer YdhE. Those bounds are crude and do not enable PAC-Bayes training because the model does not see an update by minimizing those bounds.
>
> Nevertheless, as you correctly mentioned, we can still compare them as pure bounds instead of training algorithms. Figure (https://tinyurl.com/ycksnjvb) displays the result. More explicitly,  we numerically compared our new PAC-Bayes bound with the existing one in the reference (Corollary 4, https://arxiv.org/pdf/1605.08636.pdf) mentioned by the reviewer, which is also for unbounded loss. Even though the forms of the bounds look similar, their actual numerical values are significantly different. As shown in this figure, our proposed bound is far tighter than this baseline one. The sub-Gaussian norm term ($s^2$) in this baseline bound alone is larger than $100$, making it not appealing in practice.

---

### Official Review · Reviewer_DC8a · 2023-10-31

**Soundness:** 4 excellent
**Presentation:** 3 good
**Contribution:** 4 excellent
**Rating:** 8
**Confidence:** 4

**Summary:**

The paper proposes an approach for PAC Bayesian approach for training deep neural nets that automates the determination of the model hyperparameters based on a learning-theoretic bound instead of performing combinatorial search. The developed PAC Bayesian bound has novel aspects such as being applicable to unbounded losses by making more plausible assumptions than the prior art. The paper evaluates the practical benefit of the developed bound on diverse and challenging use cases such as training neural nets with a depth of 10+ layers on image classification and graph neural nets on five different graph prediction benchmarks. The proposed method appears to reach state-of-the-art performance or above in most of these use cases.

**Strengths:**

The paper does a really good job at identifying the bottlenecks of the existing approaches, such as how the assumed range of exponential moment inequality leads to vacuous bounds. There are many other such to-the-point statements clearly justifying and motivating the proposed solution.

The proposed way of developing a PAC Bayes bound for unbounded loss is novel and very interesting. The way it moves from bounded loss to bounded moment generating functions is truly creative and elegant.

The algorithm derived from the bound addresses a fundamental problem of machine learning: tuning hyperparameters of large-scale predictors. The enterprise is very ambitious and the reported results are very promising.

**Weaknesses:**

There is ample room to improve the clarity of Section 6. The current version is missing a good amount of essential information. As far as I understand, a key message of the paper is “do not do grid search, do PAC Bayes training instead”. Then grid search appears as the main baseline to improve on. Wouldn’t it then make senses to reserve some space in Section 6 to describe how they build the grid, why it is a strong alternative to PAC Bayes (i.e. how do we know that it contains competitive hyperparam values), and how much computation overhead it brings?

**Questions:**

It looks to me possible to use an existing PAC Bayes bound, such as one from Dziugaite et al. or Haddouche et al. for the same purpose: hyperparameter tuning. How does the proposed bound compare to them on the same experiment setup? I believe that I see what is novel with the bound but I do not immediately see why it should be a better bound, better in the sense of both being tighter and being a training objective that gives improved generalization accuracy. How do the current experiment results help us make this comparison? If they cannot, could the authors report additional results during the rebuttal phase to make this comparison possible?


I also have difficulties interpreting Figure 1. What does the x axis correspond to? What does “index” stand for in this context?

The conclusion section makes the claim that the proposed method also mitigates the curse of dimensionality. How do we conclude this from the rest of the paper, especially from the experiment results?

Solid work overall.

---

> ### Author Response · Authors · 2023-11-14
>
> Thank you very much for the constructive suggestions. We will take your comments into consideration in the revision.
>
> `1. How does the proposed bound compare to Dziugaite et al. or Haddouche et al. on the same experiment setup?`
>
> Thanks for the question, which also highlights our contribution: we bridged the gap between theory and practice while using PAC-Bayes bound. For example, Haddouche et al. (PAC-Bayes Generalisation Bounds for Heavy-Tailed Losses through Supermartingales) have data-free prior to searching the hyper-parameter, which is time-consuming and needs extra validation data to enable the selection.
> Dziugaite et al. (On the role of data in pac-bayes bounds) need extra data to train the prior, which could also be expensive in practice, e.g., collecting medical data. We have also tried to use existing PAC-Bayes bounds derived for unbounded losses, e.g., [1][2]. However,  the cross entropy loss does not satisfy the bound in [1] without putting extra assumptions on the data, while [2] is too difficult to estimate in practice since it requires estimating the expectation of the second-order moment of the loss by drawing a test sample.
>
> **In summary, we did not find an existing PAC-Bayes bound that can be directly applied to the classification task without hyper-parameter searching or extra data for the prior.** The primary purpose of this paper is to bridge the gap between theory and practice.
> For our method, we proposed a data-dependent prior that can be trained with the training dataset without requiring extra data to learn the prior or extra tuning to find a good prior.
>
> [1].  Haddouche, Maxime, et al. "PAC-Bayes unleashed: Generalisation bounds with unbounded losses."
>
> [2]. Kuzborskij, Ilja, et al. "Efron-stein pac-bayesian inequalities."
>
> `2. What does “index” stand for in this context?`
>
> We sorted the test accuracy of different searched hyper-parameters. The x-axis denotes the sorted index of the experiment.
>
> `3. The conclusion section makes the claim that the proposed method also mitigates the curse of dimensionality. How do we conclude this from the rest of the paper?`
>
> The curse of dimensionality refers to the fact that when the model gets large, the KL term (and therefore the PAC-Bayes bound) could potentially get very large. Experiments show our method can achieve state-of-the-art results, indicating this curse of dimensionality is mitigated. We also tried other bounds, which cannot handle models as large as resnet18 (on cifar10), as the bound gets too large to be practical.

---

> > ### Author Response · Authors · 2023-11-18
> >
> > Dear reviewer, we would greatly appreciate your insights and feedback on our response. Should you have any questions or require further information, we are more than willing to assist. Thank you!

---

> > > ### Comment · Reviewer_DC8a · 2023-11-23
> > > **Thanks**
> > >
> > > Thanks, your answers address my concerns satisfactorily. I keep my grade as an accept.

---

### Official Review · Reviewer_R7cX · 2023-11-02

**Soundness:** 3 good
**Presentation:** 3 good
**Contribution:** 3 good
**Rating:** 6
**Confidence:** 3

**Summary:**

The paper introduces a training framework that improves the generalization ability of neural networks without extensive hyper-parameter tuning and regularization. By minimizing the PAC-Bayes bound with trainable priors, the framework achieves comparable performance to traditional methods like SGD/Adam, even without additional regularization. It eliminates the need for hyper-parameter search and reduces reliance on validation data. The paper highlights the importance of weight decay and noise injections as essential techniques. The approach shows promise for enhancing generalization in deep neural networks.

**Strengths:**

1. This paper generally is well-written and easy to follow.
2. The idea of tuning-free generalizaton with trainable prior is both interesting and theoretically grounded, and seems to be promising in training neural networks, especially when it can be extended to large-scale neural networks, e.g., transfermers.
3. This paper has provided solid theoretical analysis, which can be inspiring for the follow-up works.

**Weaknesses:**

1. The paper suggests that only weight decay and noise injections are essential for PAC-Bayes training. However, this conclusion seems premature and lacks comprehensive analysis. It would be beneficial to investigate and compare the impact of other regularization techniques commonly used in deep learning, such as dropout or batch normalization, within the proposed framework. This would provide a more comprehensive understanding of the interplay between different regularization methods and their contribution to generalization performance.
2. The method proposed in this paper may require i.i.d. data and may not be able to deal with out-of-distribution tasks.

**Questions:**

1. Could the authors elaborate more on why it is so important to give a generalization bound on unbounded loss? Existing bounded one can not work well in practice? An empirical compare with them when apply those bounds for training? From my understanding, the bounded part has now been shifted to the bounded $\gamma$ with $\gamma_1$ and $\gamma_2$ in the unbounded bounds.
2. Could the authors elaborate more on why optimizing prior on training dataset will be helpful for the generalization performance. Normally, we should fix prior or choose a good one using validation dataset.

---

> ### Author Response · Authors · 2023-11-14
>
> `1. The paper suggests that only weight decay and noise injections are essential for PAC-Bayes training. However, this conclusion seems premature and lacks comprehensive analysis.`
>
> We agree that studying the effect of the interplay between different kinds of regularization is still an important open problem, and we have mentioned it in the introduction. Here, we state that only noise injection and weight decay are essential from our derived PAC-Bayes bound.
>
> Like most commonly used implicit regularizations (large lr, momentum, small batch size), dropout and batch-norm are also known to penalize the loss function's sharpness indirectly. [1] studies that dropout introduces an explicit regularization that penalizes **sharpness** and an implicit regularization that is analogous to the effect of stochasticity in small mini-batch stochastic gradient descent. Similarly, it is well-studied that batch-norm [2] allows the use of a large learning rate by reducing the variance in the layer batches, and large allowable learning rates regularize **sharpness** through the edge of stability [3].
>
> As shown in the equation below, the first term (noise-injection) in our PAC-Bayes bound explicitly penalizes the Trace of the Hessian of the loss, which directly relates to sharpness and is quite similar to the regularization effect of batch-norm and dropout. During training, suppose the current posterior is $\mathcal{Q}_{\hat{\sigma}}(\hat h) = \mathcal{N}(\hat h,\textrm{diag}(\hat \sigma))$,
>
> Let $h \sim \mathcal{Q}_{\hat{\sigma}}(\hat h)$,
>
> and $\Delta{h} \sim \mathcal{Q}_{\hat{\sigma}}(0)$.
>
> The training loss expectation over the posterior is:
>
> $\mathbb{E}_{h} \ell(h;\mathcal{D})$
>
> $= \mathbb{E}_{ \Delta{h} } \ell(\hat{h}+\Delta {h};\mathcal{D})$
>
> $\approx \ell(\hat h;\mathcal{D})+\frac{1}{2} \mathrm{Tr} (\textrm{diag}(\hat\sigma) \nabla^2 \ell(\hat{h};\mathcal{D}))$
>
> The second regularization term (weight decay) in the bound additionally ensures that the minimizer found is close to initialization. Although the relation of this regularizer to sharpness is not very clear, empirical results suggest that weight decay may have a separate regularization effect from sharpness. So, in brief, we state that the effect of sharpness regularization from dropout and batch norm can also be well emulated by noise injection with the additional effect of weight decay.
>
>   1) Wei, Colin, et al. "The implicit and explicit regularization effects of dropout."
>   2) Luo, Ping, et al. "Towards Understanding Regularization in Batch Normalization."
>   3) Cohen, Jeremy, et al. "Gradient Descent on Neural Networks Typically Occurs at the Edge of Stability."
>
> `2. The method proposed in this paper may require i.i.d. data.`
>
> Yes, that is correct. The PAC-Bayes analysis to study generalization is limited to in-distribution tasks. We do not think of it as a drawback but rather out of the scope of this paper. Analyzing out-of-distribution generalization is still a developing field and will need additional statistical tools to derive generalization bounds for OOD tasks. However, there are papers discussing using the PAC-Bayes bound to detect the out-of-distribution samples [4], which could be the future work.
>
> `3. Why is it so important to give a generalization bound on unbounded loss? Existing bounded one can not work well in practice? Any empirical comparison with them when applying those bounds for training?`
>
> Since the prevalent cross-entropy loss (CE) is unbounded, it is important to derive a PAC-bayes bound to deal with it. Existing ones for bounded loss (such as Theorem 2.1) do not work well on the CE loss because the $C$ in Theorem 2.1 has to be set to infinity for CE loss, which makes the bound meaningless.  We also tried to approximate the CE loss by a bounded loss first and then used the PAC-Bayes bound for bounded loss, which again failed to work when we explicitly tried to truncate/clip the cross-entropy loss to make the loss bounded. In the experiment, we observe that this will cause the training and test accuracy to plateau at very low levels around $10\\%$ to $20\\%$.
>
> `4. The bounded part has now been shifted to the bounded $\gamma$ .`
>
> The bounded part has been shifted to the $K(\lambda)$ in Theorem 4.1. More explicitly, the $K(\lambda)$ in Theorem 4.1 (our theorem) corresponds to the $C$ (the upper bound of the loss) in Theorem 2.1 (the one for bounded loss). So, we can understand $K(\lambda)$ as an effective bound of the unbounded loss. Limiting the range of $\gamma$ to $[\gamma_1,$ $\gamma_2]$ reduces the value of $K(\lambda)$, which makes the bound tighter. For instance, if we don't limit this range, the $K(\lambda)$ we found for cifar10 on resnet18 would be $4$ versus $0.3$ when we limit the range.

---

> > ### Author Response · Authors · 2023-11-14
> >
> > `5. Why optimizing prior to training dataset will be helpful for the generalization performance?`
> >
> > Thank you for highlighting this important central question on PAC-Bayes bound for neural networks. In applications of PAC-Bayes bounds to generalization error, the contribution of the KL divergence often dominates the bound: To have a small KL, the posterior must be close to the prior. This means it is critical to choose a good prior. However, this is difficult without knowledge of (or access to) the data distribution. Indeed, using data-independent priors (choosing the prior that is independent of data) often leads to vacuous bounds due to ignoring important, favorable properties of the data distribution. Alternatively, one can search for good priors based on validation data, but as the searching grid gets denser and denser, this is almost equivalent to optimizing over the priors, except that the latter is more efficient.
> >
> > Moreover, some tasks lack training data samples, e.g., node classification based on graph neural networks only has around 100 training data points, and splitting more nodes to select the prior will compromise the test performance in this scenario. At last, hyper-parameter tuning could be time-consuming, which we intend to avoid in this paper.

---

> > > ### Author Response · Authors · 2023-11-18
> > >
> > > Dear reviewer, we respectfully seek your thoughts and comments on our response. Please let us know if there is any further information you need or questions we can answer. Thank you in advance.

---

> > > > ### Comment · Reviewer_R7cX · 2023-11-22
> > > > **Thank the authors for their detailed response**
> > > >
> > > > I would like to thank the authors for their clear and detailed response. I truly encourage the authors to include these important discussion into their revised paper.
> > > >
> > > > However, I still have the concern about the validity of introducing a bounded $\lambda$ to make unbounded loss able to be bounded. It sounds like giving the theoretical results in a smaller domain. I suppose that the range of $\lambda$ may significantly impact the theoretical results and in certain cases the bound will be degenerated into the unbounded one? Overall, I would like to retain my score.

---

> > > > > ### Author Response · Authors · 2023-11-22
> > > > >
> > > > > Do you mean the boundedness of $\gamma$ instead of $\lambda$? Since $\lambda$ is a prior parameter optimized with the data, it is not bounded.
> > > > >
> > > > > The boundedness of $\gamma$ is not essential to get a finite PAC-Bayes bound for unbounded loss. Even if we do not require $\gamma$ to be bounded (i.e., allow $\gamma$ to take values in $[0,\infty)$),  our PAC-Bayes bound is still finite.
> > > > >
> > > > > More specifically, when $\gamma$ is unbounded, the $K(\lambda)$ in our bound will reduce to the sub-Gaussian norm of the output of the network, which is still bounded. Therefore, the ability to derive a finite PAC-Bayes bound for unbounded loss is not due to the restriction of $\gamma$ but due to the use of a sub-Gaussian type of bound. Since the sub-Gaussian norm can be understood as some sort of variance, it could be finite even if the loss is unbounded.
> > > > >
> > > > > The reason we further confine the range of $\gamma$ is that we want tighter numerical values for our bound to be useful in practice. We observed that $\gamma$ never gets very large in practice, so we think, why not exclude the useless $\gamma$ from the range when computing the $K$. This indeed helped a lot to get us the first trainable PAC-Bayes bound for large networks with cross-entropy loss, as presented in the paper.

---

> > > > > > ### Author Response · Authors · 2023-11-22
> > > > > >
> > > > > > Or do you mean to ask whether an unbounded loss should not have an upper bound?
> > > > > >
> > > > > > Given that the test error is defined as the expected loss of the posterior over the data distribution, we can establish an upper bound for this expectation despite the possibility of the loss being extremely large for certain input samples. It is improbable that the posterior, optimized from a random initialization, will perform poorly for all samples, especially considering that the loss at random initialization is not exceptionally large.

---

### Meta-Review · Area_Chair_Waot · 2023-12-06

**Metareview:**

This paper proposes a training framework based on PAC-Bayes bounds that offers similar performance to existing training approaches with less need for hyperparameter tuning. After an active discussion between authors and reviewers, the paper is very much borderline, with three reviewers leaning towards acceptance and two towards rejection. While the reviewers praised the theoretical analysis, and the novelty of the approach, they were critical of the limited set of studied regularization techniques, the clarity of the experiments section, the runtime of the method, and general question of practicality in typical training settings (e.g., using data augmentation). Unfortunately, the criticism seems to outweigh the praise at the time being. I believe that this could be a really interesting contribution and would encourage the authors to take the reviewer feedback seriously and resubmit a revised version of the paper in the future.

**Justification For Why Not Higher Score:**

the reviewers believe that the paper could be significantly improved for a resubmission

**Justification For Why Not Lower Score:**

N/A

---

### Decision · Program_Chairs · 2024-01-16

Reject